



# Application of fuzzy *c*-means clustering for analysis of chemical ionization mass spectra: insights into the gas-phase chemistry of NO₃-initiated oxidation of isoprene

Rongrong Wu[1], Sören R. Zorn[1], Sungah Kang[1], Astrid Kiendler-Scharr[1†], Andreas Wahner[1]
and Thomas F. Mentel[1*]

[1]Institute of Energy and Climate Research, Troposphere (IEK-8), Forschungszentrum Jülich GmbH, 52428 Jülich, Germany

† Deceased February-06th, 2023

*Correspondence to*: Thomas F. Mentel (t.mentel@fz-juelich.de)

## Abstract

Oxidation of volatile organic compounds (VOCs) can lead to the formation of secondary organic aerosol, a significant component of atmospheric fine particles, which can affect air quality, human health, and climate change. However, current understanding of the formation mechanism of SOA is still incomplete, which is not only due to the complexity of the chemistry, but also relates to analytical challenges in SOA precursor detection and quantification. Recent instrumental advances, especially the developments of high-resolution time-of-flight chemical ionization mass spectrometry (CIMS), greatly enhanced the capability to detect low- and extremely low-volatility organic molecules (L/ELVOCs). Although detection and characterization of low volatility vapors largely improved our understanding of SOA formation, analyzing and interpreting complex mass spectrometric data remains a challenging task. This necessitates the use of dimension-reduction techniques to simplify mass spectrometric data with the purpose of extracting chemical and kinetic information of the investigated system. Here we present an approach by using fuzzy *c*-means clustering (FCM) to analyze CIMS data from chamber experiments aiming to investigate the gas-phase chemistry of nitrate radical initiated oxidation of isoprene.

The performance of FCM was evaluated and validated. By applying FCM various oxidation products were classified into different groups according to their chemical and kinetic properties, and the common patterns of their time series were identified, which gave insights into the chemistry of the system investigated. The chemical properties are characterized by





elemental ratios and average carbon oxidation state, and the kinetic behaviors are parameterized with generation number and effective rate coefficient (describing the average

reactivity of a species) by using the gamma kinetic parameterization model. In addition, the fuzziness of FCM algorithm provides a possibility to separate isomers or different chemical processes species are involved in, which could be useful for mechanism development. Overall FCM is a well applicable technique to simplify complex mass spectrometric data, and the chemical and kinetic properties derived from clustering can be utilized to understand the

reaction system of interest.



## 1. Introduction

Volatile organic compounds (VOCs) in the atmosphere are oxidized by reactions with hydroxyl radicals (OH), ozone ($O_3$), nitrate radicals ($NO_3$), or Cl atoms, and converted to condensable vapors such as low- and extremely low-volatility organic compounds (LVOCs/ ELVOCs) that subsequently can condense onto existing particles or even form new particles and thereby form secondary organic aerosol (SOA) (Donahue et al., 2012; Hallquist et al., 2009; Ziemann and Atkinson, 2012). Secondary organic aerosol comprises a major fraction of the atmospheric submicron particulate matter and can have an adverse impact on air quality, human health, and climate (Hallquist et al., 2009; Jimenez et al., 2009; Pöschl, 2005; Spracklen et al., 2011; Zhang et al., 2007). Despite extensive studies on characterization of the products and mechanisms involved in VOC oxidation and SOA formation, how VOCs contribute to SOA formation is not yet fully understood. This is not only hampered by the complexity of the chemistry itself, but also by the remaining analytical challenges in detection of organic precursors with low volatility (Bianchi et al., 2019; Shrivastava et al., 2017).

Recent instrumental developments, especially the propagation of high-resolution time-of-flight chemical ionization mass spectrometry (CIMS), made the direct detection of low-volatility vapors possible (Ehn et al., 2012; Ehn et al., 2014; Jokinen et al., 2015). Benefitting from this it has been discovered that the highly oxygenated organic molecules (HOM), which are formed through a rapid gas-phase process called autooxidation and generally have very low volatilities, significantly contribute to SOA and even new particle formation (Crounse et al., 2013; Ehn et al., 2012; Ehn et al., 2014; Kirkby et al., 2016; Praske et al., 2018).

While advanced mass spectrometers greatly enhance our capability to investigate the chemical composition and evolution of HOM, the highly complex mass spectrometric data consisting of hundreds to thousands of variables (i.e., detected ions) over thousands of points in time makes the data processing and interpretation challenging. In addition, the molecular structure information of detected ions can only be extrapolated from their chemical composition, notwithstanding the modern apparatus of high resolution (e.g., over 10,000 m/Δm) (Breitenlechner et al., 2017; Krechmer et al., 2018), which significantly hinders the understanding of the involved chemical processes. Furthermore, it is difficult to refine and extract kinetic and mechanistic information directly from the mass spectrometric data.

To reduce this complexity dimension-reduction techniques are necessary, which compress the information in a dataset into a few to a dozen factors or clusters based on the underlying physical or chemical properties of the different variables, and thus simplify the chemistry of



investigated systems (Äijälä et al., 2017; Buchholz et al., 2019; Koss et al., 2020; Yan et al.,
2016; Zhang et al., 2019).

Factorization is one of the major data dimension-reduction techniques, within which
positive matrix factorization (PMF) (Paatero, 1997; Paatero and Tapper, 1994) is the best-
known approach in atmospheric science, especially for ambient measurements of particulate
matter by aerosol mass spectrometer (Canonaco et al., 2013; Lanz et al., 2007; Lanz et al., 2008;
Zhang et al., 2005; Zhang et al., 2011), as well as for VOC measurements in both field and
laboratory studies (Brown et al., 2007; Lanz et al., 2009; Li et al., 2021; Rosati et al., 2019;
Vlasenko et al., 2009; Yuan et al., 2012). Principal component analysis (PCA) (Wold et al.,
1987) is also a frequently used multivariate factor analysis technique for deconvolution and
interpretation of gas-phase and particle-phase composition data (Sofowote et al., 2008; Wyche
et al., 2015; Zhang et al., 2005). Additionally, non-negative matrix factorization (NMF), which
is very similar to the PMF approach, has been widely used in interdisciplinary fields (Devarajan,
2008; Fu et al., 2019; Lee and Seung, 1999) as well as in atmospheric science (Chen et al.,
2013; Karl et al., 2018; Malley et al., 2014; Song et al., 2021). Despite the similarities of
mathematical formulation and constraints to PMF, the NMF algorithm does not require an error
matrix as input, largely reducing the uncertainties that might be introduced by inappropriate
error estimation methods (Buchholz et al., 2019). In addition to factorization methods, recently
increasing number of studies adopted clustering techniques to mass spectra data (Äijälä et al.,
2017; Koss et al., 2020; Li et al., 2020; Priestley et al., 2021). For example, Äijälä et al. (2017)
combined a clustering algorithm, $k$-means ++, with PMF to classify and characterize the
organic component of air pollution plumes detected by AMS. Li et al. (2020) developed a
clustering algorithm named noise-sorted scanning clustering, based on the traditional density-
based special clustering of applications combined with a noise algorithm, and thereafter applied
this method to distinguish different types of thermal properties of different biogenic SOA. Koss
et al. (2020) compared the performance of hierarchical clustering analysis (HCA) with PMF
and gamma kinetics parameterization for analyzing complex mass spectrometric data. Their
results demonstrate the ability of HCA to identify major types of ions and patterns of time
behavior and draw out bulk chemical properties of the system that can be useful for modeling.
In addition, in a recent work Priestley et al. (2021) applied HCA techniques to infer CHON
functionality of products formed from benzene oxidation.

In this work, we choose the fuzzy $c$-means clustering algorithm (FCM) as the major
technique to analyze CIMS data collected from a chamber experiment, aiming to investigate
the gas-phase chemistry of the isoprene-NO$_3$ oxidation system. Fuzzy $c$-means clustering is the



most widely used fuzzy clustering algorithm and is adopted in this study considering the
following two aspects. Firstly, FCM allows variables to be affiliated with more than one

clusters, as PMF does, whereas hard clustering methods like the most popular *k*-means
clustering forces each variable into one cluster exclusively. In atmospheric chemistry, one
compound can originate from several different sources, or a species detected may consist of
isomers produced from different chemical processes. Therefore, assigning a variable into
multiple clusters with a quantified membership degree is more rational in this case than

assigning variables to mutually exclusive clusters. Secondly, FCM only needs a data matrix as
input, whereas for some factor analysis methods additional information is required, such as the
error matrix needed in PMF, which is usually estimated by user-defined error estimation
schemes and could result in perceptibly different outcomes accordingly (Buchholz et al., 2019;
Paatero et al., 2014; Paatero and Tapper, 1994; Ulbrich et al., 2009).

By using FCM, variables with similar time behaviors will be grouped into the same cluster,
and the centroid of this cluster (cluster center) can be used as a surrogate of these variables.
Therefore, the numerous species detected in a chemical system can be simplified and
characterized by a much smaller number of clusters, each of which represents a typical
chemical process with unique kinetic behavior. The significant reduction of the complexity of

the chemical system and the chemical and kinetic information derived from this method can
help to better understand the chemical system of interest (Koss et al., 2020). In addition, we
applied FCM to a synthetic dataset derived from a box model with explicit mechanism to
evaluate the performance of FCM clustering. By exemplifying the functionality of such a
clustering method in analyzing CIMS data, we propose that FCM is a useful method that offers

a new way to analyze mass spectrometric data and derives useful information on chemical and
kinetic properties of products that can help decipher the underlying reaction mechanism.

## 2. Methods

### 2.1 Data collection and processing

The experimental data used in this work were collected in the atmospheric simulation chamber

SAPHIR at the Forschungszentrum Jülich, Germany, during the ISOPNO$_3$ campaign in 2018.
The SAPHIR chamber is a double-walled Teflon (PEP) cylinder with an approximate volume
of 270 m$^3$ (5m in diameter, 20m in length). It is fixed by an aluminum frame with movable
shutters that can be opened or closed to simulate daytime or nighttime chemistry. Trace gases



in the chamber can be well mixed within 2 minutes with the help of two continuously operated
fans. During an experiment the chamber is filled with synthetic air and kept slightly over
pressured (~ 35 Pa) to prevent permeation of outside air into the chamber. Due to small leakages
and instrument sampling consumption, there is a replenishing flow into the chamber which
leads to a dilution rate of 4% – 7% h$^{-1}$. More details about the chamber setup and its
performance can be found elsewhere (Rohrer et al., 2005).

The experiment selected here was conducted to characterize the gas-phase chemistry of
$NO_3$-initiated oxidation of isoprene. Nitrate radicals were produced in situ by the reaction of
$NO_2$ with $O_3$, followed by the addition of ~10 ppbv of isoprene to initiate the reaction. The
injection was repeated four times (only $NO_2$ and $O_3$ were added during the last injection) to
build up products and facilitate later-generation oxidation. The mixing ratios of $O_3$ and $NO_2$ in
the chamber were approximately 100 and 25 ppbv, respectively, after the first injection, but
this was not uniform every time, as shown in Fig. S1. Detailed description of the experimental
procedure can be found elsewhere (Wu et al., 2021).

During the campaign a comprehensive set of instruments was deployed for the
measurements of radicals and products in both gas- and particle-phase, as described by Wu et
al. (2021). In this work, however, we focus on the measurements acquired by a high-resolution
time-of-flight chemical ionization mass spectrometer (Aerodyne Research Inc.) using Br$^-$ as
reagent ion, which detected the $HO_2$ radical and the gas-phase products generated by the
reaction of isoprene and $NO_3$. The mass spectrometer was operated in "*V*" mode with a mass
resolution of 3000 – 4000 ($m/\Delta m$). A customized inlet was designed to connect the CIMS
directly to the chamber to reduce losses of the $HO_2$ radical and HOM in the sampling line
(Albrecht et al., 2019). More information about settings and performance of the instrument can
be found in our previous study (Wu et al., 2021).

The raw mass spectrometric data were processed using the Tofware toolkit (v. 2.5.11,
Tofwerk AG/ Aerodyne Research Inc.) in Igor Pro (v.7.0.8, WaveMetrics) following the
routines described by Stark et al. (2015). Overall, more than 500 peaks were detected above
the background in the mass spectra obtained by the Br$^-$-CIMS. The background signal of each
peak was determined from measurements prior to precursor injection and was subtracted from
the signal measured in the chamber. These peaks consist of ions related to real isoprene
oxidation products, as well as other signals related to ion source, internal standard, and
interferences from chamber and tubing.

The product ions are those produced by isoprene oxidation, and they should have
pronounced changes when the chemistry is initiated or modified. Therefore, a simple way to



screen out the product ions from other chemically irrelevant signals is to examine the time evolution of each ion. By comparing the signals before and after each injection we can easily

distinguish the product ions from others. However, this cannot exclude variabilities unrelated to oxidation chemistry during the experiment itself. Therefore, high-resolution analysis was conducted in the mass range of *m/z* 60 – 600 to identify the chemical composition of detected ions. For high-resolution peak assignment, we fitted the observed peak using predefined instrument functions (including peak shape, peak width as a function of *m/z*, and baseline). If

necessary, contributions of more than one component were considered for the fit to reduce the residuals of the fitting. Once the peak numbers and peak positions were fixed, the chemical formula (consisting of C, H, O, and N atoms) of each peak was assigned manually by selecting from a formula list generated by the software. During the peak fitting isotopes were constrained, and only formulas within an accuracy tolerance of 10 ppm and with reasonable chemical

meanings were considered. In addition, only molecule formulas with a time behavior commensurable with expectations for the specific chemical system were assigned (Pullinen et al., 2020). For example, it is illogical if large amounts of organonitrates are observed under low $NO_x$ conditions. Since we intend to investigate the underlying chemical relationships of different products through their time behavior, not the absolute concentration, normalized

signals were finally used for further analysis. Calibration procedures are described in more detail elsewhere (Wu et al., 2021).

In addition to abovementioned chamber data, we use a synthetic dataset from a box model with the default gas-phase reaction schemes of isoprene-$NO_3$ taken from the Master Chemical Mechanism (MCM) version 3.3.1 (Jenkin et al., 2015). For the modelling, temperature, relative

humidity, and dilution rate were constrained by using measured data. The initial concentrations of $O_3$, $NO_2$ and isoprene were added into the model according to the experimental schedule. Overall, the modelled concentrations of $O_3$, $NO_2$, $NO_3$ and isoprene match the measurements well (Fig. S2). Here we only use the synthetic data to learn about principal behavior of time series in complex chemical systems using an established complex mechanism. Detailed

description of isoprene-$NO_3$ chemistry and evaluation of the model performance are outside the scope of this work. A recent updated mechanism for isoprene + $NO_3$ can be found in Carlsson et al. (2022).





**2.2 Fuzzy *c*-means clustering (FCM)**

Clustering is one of the major dimension-reduction techniques besides factorization, which

groups a set of objects into a certain number of clusters according to their (dis)similarities (generally measured by a distance metric) such that objects within each cluster are much closer to each other than to those pertaining to other clusters (Hastie et al., 2009). The notion of a fuzzy set was firstly proposed by Zadeh (1965), which gave an idea how to deal with data with indistinct boundaries of clusters. Based on this concept Bezdek et al. (1984) developed the

fuzzy *c*-means clustering algorithm. In contrast to the hard clustering counterparts such as *k*-means and *k*-medoids clustering, FCM allows each object to belong to multiple clusters with the membership degree measured by a value varying from 0 to 1 (Bezdek et al., 1984). Consequently, fuzzy clustering can better deal with non-discrete data, and thus is adopted here to analyze our CIMS data for the example of isoprene-$NO_3$ oxidation.

Fuzzy *c*-means clustering is one of the best-known fuzzy clustering algorithms by virtue of its simplicity, quick convergence, and wide applicability (Ghosh and Dubey, 2013; Ren et al., 2016; Yang, 1993;). It is a distance-based cluster assignment method, and its working principle is very similar to that of the *k*-means algorithm. FCM is conducted through an iterative process which attempts to group all objects within a dataset into a predefined number of clusters

(*c*) with a degree of membership, and meanwhile minimize the sum of squared distance between the member objects and the cluster centroid of each cluster, as defined in Eq. 1:

$$J_m(U,V) = \sum_{i=1}^{c} \sum_{j=1}^{n} u_{ij}^{m} d_{ij}^{2} \qquad (1)$$

$$v_i = \frac{\sum_{j=1}^{n} u_{ij}^{m} \cdot x_j}{\sum_{j=1}^{n} u_{ij}^{m}} \qquad (2)$$

$$u_{ij} = \left\{ \sum_{k=1}^{c} \left( \frac{d_{ij}}{d_{kj}} \right)^{\frac{2}{(m-1)}} \right\}^{-1} \qquad (3)$$

where $x_j$ represents the $j^{th}$ object in the dataset, $u_{ij}$ is the membership degree of $x_j$ to the $i^{th}$ cluster, which is enforced to satisfy $u_{ij} \in [0,1]$ and $\sum_{i=1}^{c} u_{ij} = 1$, $d_{ij} = \|x_j - v_i\|$ denotes the distance between object $x_j$ and the $i^{th}$ cluster center $v_i$, and *m* is the fuzzifier ($m \in [1, \infty)$) that controls the fuzziness level of clustering.

The clustering procedure of FCM is executed through an iterative strategy to minimize

the objective function $J_m(U,V)$. By initializing the fuzzy partition matrix $U$ randomly, one can compute the cluster centers (*V*) according to Eq. 2 with the constraint of the sum of the





membership degrees of an object to all clusters being unity. In the consecutive iteration, new membership degrees are calculated following Eq. 3. The calculation proceeds by repeating above process, and every iteration generates two new sets of *V* and *U*. The iteration ends when

the algorithm converges (no significant change with further iteration, namely $\left\| U^{(t+1)} - U^{(t)} \right\| = max_{i,j}\left\{ \left| u_{ij}^{(t+1)} - u_{ij}^{t} \right| \right\} < \varepsilon$), or the predefined maximum number of iterations is reached.

**2.3 Clustering parameters**

As noted in Sect. 2.2, several parameters need to be specified ahead of executing FCM

including the number of clusters, the distance metric to measure (dis)similarity of objects, the value of the fuzzifier, the initial fuzzy partition matrix, the maximum number of iterations, and the stopping criterion. All these parameters can affect the partition outcome, but among them the most important ones are the selection of cluster number, the distance metric, and the fuzziness index. A brief introduction to these parameters and methods how to determine their

optimal values will be discussed in detail in the following sections.

**2.3.1 Number of clusters (*c*)**

Finding the optimal number of clusters (*c*) is one of the challenges in cluster analysis. The optimal number of clusters is related to the structure of the investigated dataset and has a critical impact on clustering outcomes. To our knowledge none of the existing methods has been

proven to be able to determine the perfect cluster number in all possible cases and applications.

The frequently used method to address this problem is to set the search range of *c*, run clustering to generate solutions according to the predefined number of clusters, and then choose one or more clustering validity indices (CVIs) to evaluate the clustering outcomes. By comparing the values of CVI(s) of alternative clustering solutions obtained with different

number of clusters, the appropriate *c* is determined accordingly.

In this case, a validity index is used as a fitness function to evaluate the quality of the obtained clustering solutions in terms of intra-cluster compactness and inter-cluster separation. In addition, CVIs play an extremely important role in automatically determining the appropriate number of clusters. Plenty of CVIs have been proposed in the past. Generally, these

CVIs can be divided into three categories. The first category only uses the property of membership degree in the calculation, such as the partition coefficient (Bezdek and Pal, 1998) and partition entropy (Simovici and Jaroszewicz, 2002), which are also the earliest validity





indices for fuzzy clustering. The main disadvantage of such CVIs is that they lack direct connection to the geometry structure of the data. Therefore, with improvements, the CVIs in
the second category consider both membership degree and the geometry structure of the data in the calculation. Fukuyama-Sugeno index (Fukuyama, 1989), Xie-Beni index (Xie and Beni, 1991), Kwon index (Kwon, 1998) and Bouguessa-Wang-Sun index (Bouguessa et al., 2006) are some well-known examples of the second category. Given their advantages over those in the first category, we only chose CVIs belonging to the second category in this study. Different
from those in the first two categories, CVIs in the third category make use of the concept of hypervolume and density. The fuzzy hypervolume and the average partition density (Gath and Geva, 1989) are the most popular two indices within this category.

Although there're various of CVIs, no CVI can always outperform others due to their own limitations and complexity of different datasets (Kryszczuk and Hurley, 2010; Wang et al.,
2021). Generally, each CVI only attaches importance to a specific aspect or limited aspects of a clustering solution, while other aspects can be inadequately represented or even overlooked (Kryszczuk and Hurley, 2010). Consequently, we adopt multiple CVIs in this study, and among all the alternatives following six CVIs were chosen, namely the sum of within-cluster variance($V_{SWCV}$, Elbow method), Fukuyama-Sugeno index ($V_{FS}$), Xie-Beni index ($V_{XB}$), Kwon
index ($V_{Kwon}$), Bouguessa-Wang-Sun index ($V_{BWS}$) and fuzzy Silhouette ($FS$, Campello and Hruschka, 2006). This selection of CVIs is dictated by the fact that they are most frequently referred to in literature and are reported to perform well (Bouguessa and Wang, 2004; Campello and Hruschka, 2006; Rawashdeh and Ralescu, 2012; Zhou et al., 2014). More information about these CVIs can be found in the Supplement S1.

With respect to the search range of the number of clusters, a rule of thumb for the maximum number of clusters suggests that it should not exceed $\sqrt{n}$ ($n$ here is the number of elements in a dataset) (Ren et al., 2016; Yu and Cheng, 2002). Therefore, the search range of $c$ is set to be constant in $[2, \sqrt{n} + 1]$. For each $c$ in this range, the FCM algorithm will be performed 50 times with the default settings ($m = 2$, metric = Euclidean distance, $\varepsilon = 1 \times 10^{-5}$)
and the selected CVIs will be calculated for each repetition. By evaluating the variations in CVIs with different $c$, what we believe to the optimal number of clusters is determined.

### 2.3.2 Distance metric

The selection of an appropriate distance or (dis)similarity metric for clustering is also challenging since it not only relates to the inherent structure of the data investigated, but also
depends on the analysis purpose. Various distance metrics have been proposed for measuring



the (dis)similarity between each pair of objects, of which the Euclidean distance is most frequently used for clustering. As defined by Eq. 4, the Euclidean distance corresponds to the true geometrical distance between two objects, and many studies selected this metric by default in FCM (Haqiqi and Kurniawan, 2015; Nishom, 2019; Singh et al., 2013). However, Euclidean

distance is not always the right choice. The Euclidean distance assumes that each attribute is equally important during clustering, namely the data being spherically distributed, so it is very sensitive to outliers (Arora et al., 2019; Dik et al., 2014). If the distribution of investigated data is non-spherical in shape, using Euclidean distance may degrade the performance of clustering (Arora et al., 2019; Gueorguieva et al., 2017; Vélez-Falconí et al., 2020).

In addition to Euclidean distance some other metrics such as the Manhattan distance, the Eisen cosine distance, and the Pearson correlation distance are used to measure (dis)similarity (Äijälä et al., 2017; Koss et al., 2020). The Manhattan distance is also named city block distance or taxicab distance. It computes the sum of the absolute differences between all sets of coordinates of pairwise objects following Eq. 5, and is less sensitive to noise (Dik et al., 2014).

When the attributes are discrete or binary, the Manhattan distance is more effective than other metrics. One disadvantage of the Manhattan distance is that it depends on the rotation of the coordinate system (Vélez-Falconí et al., 2020). The Eisen cosine and the Pearson correlation distance are correlation-based metrics. The Pearson correlation distance measures the linear dependence of two objects, and the cosine distance uses the cosine angle of two objects to

measure their (dis)similarity. Both are calculated by subtracting the correlation coefficient from 1, as defined by Eq. 6 and Eq. 7, and therefore they are invariant to the magnitudes of number of variables. Two objects are considered similar if they are highly correlated in terms of correlation-based distances, even though they may be far away from each other in Euclidean space. This is particularly beneficial when dealing with mass spectrometric data. Thus, the

cosine distance is commonly used to measure the (dis)similarity of aerosol source profiles (Äijälä et al., 2017; Bozzetti et al., 2017; Heikkinen et al., 2021; Ulbrich et al., 2009). It should be noted that even though correlation-based metrics are called as "distance", strictly speaking they are (dis)similarity metrics rather than distance metrics because they do not satisfy the triangle inequality anymore (Kaufman and Rousseeuw, 2009).

$$d(x,y) = \sqrt{\sum_{i=1}^{n}(x_i - y_i)^2} \tag{4}$$

$$d(x,y) = \sum_{i=1}^{n}|x_i - y_i| \tag{5}$$





$$d(x,y) = 1 - \frac{|\sum_{i=1}^{n} x_i y_i|}{\sqrt{\sum_{i=1}^{n} x_i^2 \sum_{i=1}^{n} y_i^2}} \tag{6}$$

$$d(x,y) = 1 - \frac{\sum_{i=1}^{n}(x_i - \overline{x})(y_i - \overline{y})}{\sqrt{(\sum_{i=1}^{n}(x_i - \overline{x})^2)}\sqrt{(\sum_{i=1}^{n}(y_i - \overline{y})^2)}} \tag{7}$$

where $x$ and $y$ are n-dimensional objects, $x_i$ and $y_i$ denote the $i^{th}$ dimension of $x$ and $y$, and $\overline{x}$

and $\overline{y}$ are the means of $x$ and $y$ in all dimensions, respectively.

Since it is difficult to know the inherent structure of high-dimensional data we make use of CVIs to find the suitable distance metric of FCM for our dataset. By running FCM with all four different distance metrics mentioned above and calculated the six CVIs accordingly while retaining all other parameters, we get four parallel results for each CVI, and the optimal

distance metric is determined by comparing the outcomes.

As mentioned above, the Euclidean distance can be severely affected by the scale of objects, which means that the (dis)similarity between objects measured by Euclidean distance might get skewed if input variables are in different scales or units. Therefore, it is highly recommended to normalize the data before clustering. We also want to scale the data to directly

compare the time behavior of different variables regardless of their differences in absolute intensity or detection sensitivity. In this study, we normalize the time-series data using the Euclidean norm to eliminate the effects of different branching ratios and sensitivity of species, and to make their time patterns easily comparable.

### 2.3.3 Value of fuzzifier

The fuzzifier ($m$, $m \in [1, \infty)$) defines the fuzziness degree of the clustering. A proper value of $m$ can suppress the noise and smooth the membership function (Huang et al., 2012). When $m$ = 1, FCM is equivalent to the $k$-means algorithm. The closer $m$ is to 1, the crisper the resulting solution becomes. On the contrary, as $m$ becomes larger, the clustering outcomes become fuzzier. When $m$ approaches infinity, different cluster centers and the centroid of all objects

will coincide, and thereby all objects have the identical membership degree to each cluster, namely $u_{ij} = 1/c$. Theoretically, the larger $m$ the fuzzier the clustering outcomes would be (Hammah and Curran, 1998). Therefore, $m$ should be selected to fulfill the request of maximum recognition of a partition with a fuzziness as small as possible.

According to previous studies, the optimal value of $m$ varies in the range of 1 to 5

(Hathaway and Bezdek, 2001; Huang et al., 2012; Ozkan and Turksen, 2007; Pal and Bezdek, 1995; Wu, 2012), and is often set to be 2 as default value as recommended by Pal and Bezdek



(1995). However, it is reported that in many cases the true value of $m$ deviates from this recommended value, which is believed to be biased by the data structure of interest (Huang et al., 2012; Hwang and Rhee, 2007; Schwämmle and Jensen, 2010; Yu et al., 2004; Zhou et al., 2014). A few methods have been proposed to determine the optimal value or range of the fuzzifier (Gao et al., 2000; Huang et al., 2012; Ozkan and Turksen, 2007; Schwämmle and Jensen, 2010). However, they are either empirical or only applicable for limited cases, and it is still an open problem to determine the appropriate fuzzifier value in FCM.

In this study we adopted the method proposed by Gao et al. (2000) to determine the optimal fuzzifier value $m^*$, which is based on the fuzzy decision-making theory. By constructing the fuzzy objective function ($G$) and the fuzzy constraint ($C$), the determination of $m^*$ is transformed into a constrained non-linear optimization problem, and the intersection of $G$ and $C$ is supposed to be the optimal solution according to the decision-making theory (Eq. 10). Since a better partition comes with a smaller sum of within-cluster variation and a larger between-cluster separation, the fuzzy objective $G$ is defined as minimizing the objective function $J_m(U,V)$ as given by Eq. 8, while the fuzzy constraint $C$ is defined as minimizing the fuzzy partition entropy $H_m(U,c)$ as given by Eq. 9. The intersection of $G$ and $C$ is taken as $m^*$, as shown in Fig. 3a, which satisfies minimizing $J_m(U,V)$ and $H_m(U,c)$ simultaneously with maximum membership degree (Gao et al., 2000).

$$\mu_G(m) = exp\left\{-\alpha \times \frac{J_m(U,V)}{\max\limits_{\forall m}(J_m(U,V))}\right\} \tag{8}$$

where $\alpha$ is a constant larger than 1, and generally set to be 1.5 in practice.

$$\mu_C(m) = \left\{1 + \beta \times \left(\frac{H_m(U,c)}{\max\limits_{\forall m}(H_m(U,c))}\right)\right\}^{-1} \tag{9}$$

where $\beta$ a constant that is usually set to be 10 in practice.

$$m^* = \arg\limits_{\forall m}\left\{max\{min\{\mu_G(m), \mu_C(m)\}\}\right\} \tag{10}$$

Based on the fuzzy decision-making method, we search for $m^*$ in the range of [1.1, 9] with an increment of 0.1. The number of clusters varies between 2 and 10, and the initial fuzzy partition matrix ($U^0$) is randomly created. Other parameters are fixed. For each setting, the algorithm will run 100 times. By evaluating the variations of $m^*$ with $c$ and the initial values of membership degree the optimal value of $m$ is determined.





### 2.3.4 Other parameters


We find that when using a small number of iterations FCM does not always return the same result for each run, and sometimes not even a valid solution. In the first case, it seems that the algorithm converges on one local minimum with several local minima existing, while in the second case the limit of iterations is reached before the algorithm converges. To avoid these

two situations, the maximum number of iterations was set to be 10000 in this study. In our case, however, hundreds of iterations can already ensure a valid solution and reproducible results for our data.

The clustering results of FCM is not as clear as that of $k$-means clustering, in which each object is forced to one cluster exclusively. Consequently, it is important to distinguish an

invalid cluster and thereby to identify an invalid solution. According to the definition of the fuzzy clustering algorithm ($\sum_{i=1}^{c} u_{ij} = 1$), each object can only belong to one cluster with a membership degree larger than 0.5. Therefore, we define a cluster with at least one object having the membership degree larger than 0.5 as a valid cluster, and a solution without any invalid clusters as a valid solution. In this work, only valid solutions were considered for further

analysis.

The initial fuzzy partition matrix was randomly created by the algorithm and 50 repetitions were used to evaluate the influence of $U^0$ on clustering outcomes. As for the stop criterion, the algorithm can offer reproducible results when this value is set to $1\times10^{-3}$ or smaller. For the calculation of results selected for analysis in this study, the stop criterion was set to $1\times10^{-5}$.

### 2.4 Gamma kinetics parameterization (GKP)

The mass spectrometric data from chamber oxidation experiments not only contain chemical composition information of the products but also a great deal of kinetic clues. The kinetic information, mainly the reaction rate and the generation number (the oxidation steps needed to produce target compound) underlying in the time series of each species are helpful for

mechanism development. However, it is challenging to extract kinetic information from time-series data, and there is only a limited number of studies which involve determination of kinetic parameters based on gas-phase measurements (Koss et al., 2020; Zaytsev et al., 2019). In this study, we try to determine the kinetic parameters based on time-series data using the gamma kinetics parameterization (GKP), which describes the multistep reaction system as a linear

system with first-order reactions and was originally used in biological and chemical fields (Zhou and Zhuang, 2007). The model returns the so-called effective rate constant (overall rate




of reactions in the pathway) and the generation number that are implied by the time behaviors of individual species (Koss et al., 2020; Zhou and Zhuang, 2007). The GKP model was introduced for atmospheric chemistry studies by Koss et al. (2020) and has been successfully

applied to parameterize the kinetics of gas-phase products formed from toluene and 1,2,4-trimethylbenzene oxidation in chamber studies (Koss et al., 2020; Zaytsev et al., 2019).

According to the GKP method the NO$_3$-initiated isoprene oxidation system can be described by Eq. 11:

$$C_5H_8 \xrightarrow{k_0 \cdot [NO_3]} P_1 \xrightarrow{k_1 \cdot [NO_3]} P_2 \xrightarrow{k_2 \cdot [NO_3]} \cdots P_m \xrightarrow{k_m \cdot [NO_3]} P_{m+1} \xrightarrow{k_{m+1} \cdot [NO_3]} \cdots \qquad (11)$$

where $k_m$ is the rate constant of product $P_m$ reacting with the NO$_3$ radical and the subscript $m$ denoting the number of oxidation steps (by NO$_3$) needed to form product $P_m$.

Typically, the rate constants for different reaction steps are disparate, and the differential equations that describe Eq. 11 are mathematically unsolvable. By assuming a single rate coefficient for all steps in a sequence the differential equations in Eq. 11 become

mathematically solvable. Additionally, the bimolecular reactions between $P_m$ and NO$_3$ must be reduced to pseudo-first-order reactions by replacing the reaction time $t$ with the integrated NO$_3$ exposure $\int_0^t [NO_3] dt$. The time series of $P_m$ can then be described by Eq. 12 (Koss et al., 2020):

$$[X_m](t) = a(k[NO_3]\Delta t)^m e^{-k[NO_3]\Delta t} \qquad (12)$$

where *a* is a scaling factor that relates to the product yield as well as to the instrument sensitivity

(Koss et al., 2020), *k* is a second-order rate constant (cm$^3$ molecule$^{-1}$ s$^{-1}$), and *m* is the generation number.

## 3. Results and discussion

### 3.1 Evaluation of clustering parameters

As already noted above, one of the biggest challenges of using FCM is that several parameters

need to be predefined, and that inadequate selection of parameters can result in unreasonable clustering outcomes. The number of clusters, the distance metric and the fuzziness value are the most important ones among all the parameters that affect the partition. Therefore, in this section we will have a close look at these three parameters and evaluate their effects on the quality of clustering based on the methods introduced in Sect. 2.3, and finally determine the

optimal values of these parameters for the analysis of our data.



### 3.1.1 Number of clusters (*c*)

To explore the effect of cluster number on partition results, we ran the FCM algorithm 50 times for each *c* in the search range and calculated the corresponding CVIs. Despite small variations in some CVIs among different repetitions, the tendency of CVIs with changing cluster number

and the optimal number of clusters indicated by each CVI are always the same for each repetition. Therefore, we only choose the results from one of the 50 repetitions for further evaluation.

Figure 1 depicts different CVIs as a function of number of clusters based on FCM results from one of the repetitions. For the sum of within-cluster variance ($V_{SWCV}$) the point of

inflection in the curve (so-called "elbow" point) indicates the best value of *c*, which is in our case 5 (Fig. 1a). The Fakuyama-Sugeno index ($V_{FS}$) uses the discrepancy between compactness and separation of clusters to measure the quality of a clustering solution (as defined by Eq. S2). A smaller value of $V_{FS}$ indicates a better partition (Fukuyama, 1989). In our case, the 7-cluster solution is the best option suggested by $V_{FS}$ (Fig. 1b), while the 5-cluster solution seems to be

a local optimum. Xie-Beni index ($V_{XB}$) is defined as the ratio of compactness and separation by Eq. S3, where the within-cluster compactness is measured by the sum of the within-cluster variance, while the between-cluster separation is measured by the minimum squared distance between cluster centers. Generally, the smaller $V_{XB}$ the better a clustering solution can be since under such conditions objects within one cluster are much closer to each other but farther away

to those in other clusters (Xie and Beni, 1991). According to Fig. 1c, the local optimal cluster number is also 5. The Kwon index ($V_{kwon}$) is a modification of $V_{XB}$ which introduces a punishing function additionally to measure the cluster compactness together with the sum of within-cluster variance. As defined by Eq. S4, the punishing function measures the average squared distance between cluster centers and the overall mean of the dataset. By introducing

this factor, $V_{kwon}$ eliminates the monotonous decreasing tendency when *c* approaches the number of objects in the dataset (Kwon et al., 2021). Like $V_{XB}$, a smaller $V_{kwon}$ indicates a better partition, and the results in Fig. 2d show that the local optimal value of *c* is as well 5.





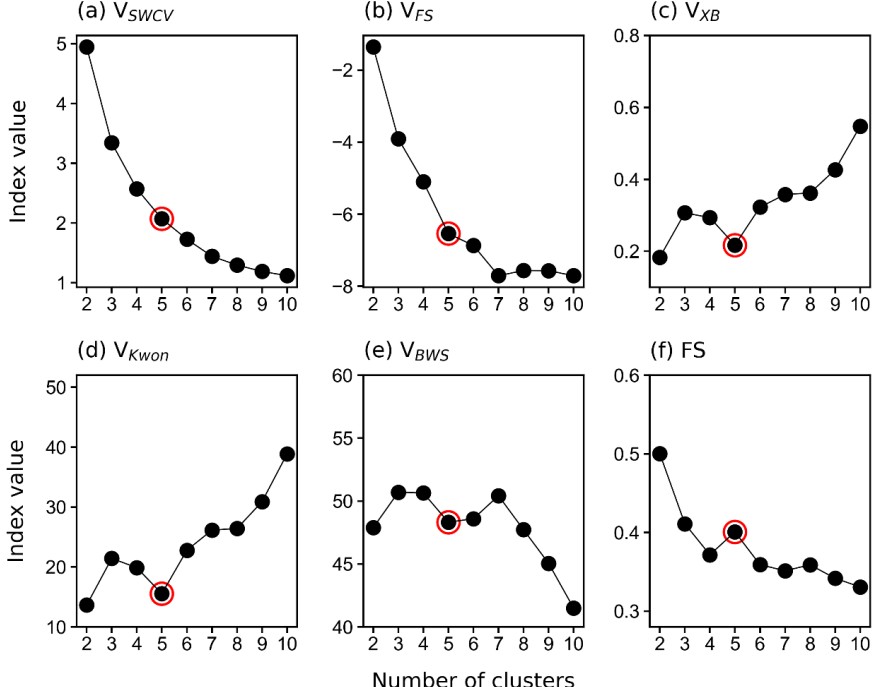

**Figure 1.** Values of selected clustering validity indices $V_{SWCV}$ (a), $V_{FS}$ (b), $V_{XB}$ (c), $V_{Kwon}$ (d),

$V_{BWS}$ (e), and $FS$ (f) as a function of the number of clusters from 2 to 10. Larger red hollow

circles indicate the solution selected for further analysis.

        In addition, we calculated the Bouguessa-Wang-Sun index ($V_{BWS}$) and the Fuzzy

Silhouette values ($FS$) for each FCM run. These two indices use slightly different definitions

of compactness and separation to measure the quality of clustering. The $V_{BWS}$ uses the fuzzy

covariance matrix as a measure of compactness, and thus $V_{BWS}$ takes cluster shape, density,

and orientation into account and has been proven to work well for largely overlapping clusters

(Bouguessa et al., 2006; Bouguessa and Wang, 2004). In general, the larger $V_{BWS}$ the better a

fuzzy partition will be, and hence the optimal number of clusters for our data is 3 and 4 based

on $V_{BWS}$ (Fig. 1e). Meanwhile, as depicted in Fig. 1e, $V_{BWS}$ shows that there is a local optimum

with $c = 7$. As for $FS$, it is an extension of the concept of Crisp Silhouette ($C$) that was originally

developed to assess non-fuzzy clustering (Rousseeuw, 1987). $FS$ is more appealing than $CS$

for fuzzy clustering since it makes explicit use of the fuzzy partition matrix. In $FS$, objects in

the near vicinity of cluster centers are given more importance than those located in the

boundary region (overlap). Consequently, it performs better than $CS$ for highly overlapping

data (Campello and Hruschka, 2006). In principle, a larger overall $FS$ suggests a better partition.





Therefore, the best number of clusters determined by $FS$ is 2 (Fig. 1f). Nevertheless, when $c =$ 2, the sum of the within-cluster variance for this solution is still quite high (Fig. 1a), which is not expected for a good partition. However, it looks reasonable to set the number of clusters to 5, which corresponds to the local maximum in terms of $FS$. It is worth noting that the silhouette

score can not only be used to assess the overall quality of partition but also to evaluate the quality of individual clusters and objects. The silhouette score ranges from -1 to +1, and a value close to +1 indicates that the object is correctly assigned. On the contrary, a silhouette value of -1 implies that the object is misclustered and should be assigned to a neighboring cluster. A silhouette value approaching 0 suggests that the object is in the overlapping region of clusters,

and thus the algorithm is unable to assign it to one cluster (Campello and Hruschka, 2006; Rawashdeh and Ralescu, 2012; Subbalakshmi et al., 2015).

In summary, different CVIs sometimes suggest a different optimal cluster number. However, by making use of information from multiple CVIs, the appropriate number of clusters in this study is determined to be 5. It should be noted that the main topic of this study is to offer

a proof of concept for the application of FCM in deconvolution of mass spectrometric data. Therefore, the depth of the discussion about the determination of the correct cluster number must suffice for our purpose, and the value of $c=5$ is selected here as one example for the chemical characterization and kinetic parameterization in the following sections. It should also be noted that the multiple CVIs method presented in this section provides a way to

automatically determine the optimal number of clusters for FCM.

**3.1.2 Distance metric**

Figure 2 shows four selected CVIs as a function of $c$ with different distance metrics. As a quick reminder, smaller $V_{FS}$ and $V_{Kwon}$ indicate better partitioning, whereas for $V_{BWS}$ and $FS$ the opposite applies. In terms of $V_{FS}$, the effects of using different distance metrics on the clustering

outcomes are negligible (Fig. 2a). However, different results arise when using $V_{BWS}$ (Fig. 2c). The $V_{BWS}$ values suggest that the cosine distance seems more appropriate for FCM regarding the data used in this study. Currently the reason for this is not clear. As for $V_{Kwon}$ and $FS$, there are no significant differences in the quality of partitioning when the number of clusters is small (e.g., $c = 2, 3, 4$) despite different distance metrics, as shown in Fig. 2b and Fig. 2d, but

discrepancies become more pronounced with increasing $c$. In general, the Euclidean distance is more appealing for our data, especially for $c$ with a value of 5, which is the appropriate cluster number determined in Sect. 2.3.1. Consequently, we conclude that among all the





distance metrics tested the Euclidean distance seems the most appropriate choice for fuzzy clustering regarding the data used in this study.

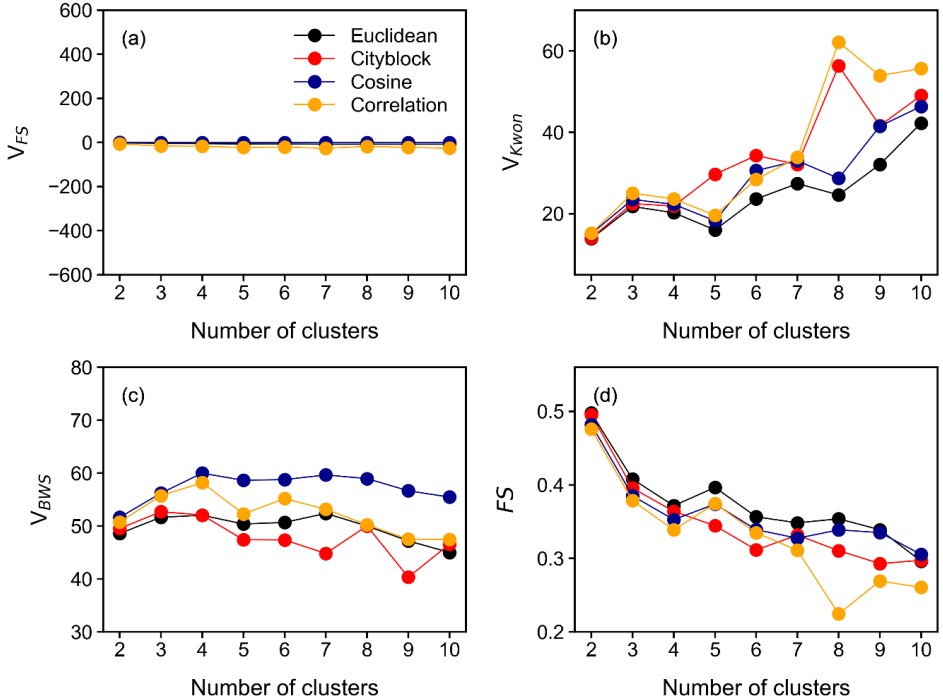


**Figure 2.** Values of selected clustering validity indices $V_{FS}$ (a), $V_{Kwon}$, (b), $V_{BWS}$ (c), and $FS$ (d) as a function of the number of clusters. Points in different color indicate results obtained with different distance or similarity metrics.

### 3.1.3 Fuzzifier value

Based on the fuzzy decision-making method introduced in Sect. 2.3.3, we searched $m^*$ in the range of [1.1, 9] with an increment of 0.1. The intersection of the fuzzy objective function, $G$, and the fuzzy constraint, $C$, as shown in Fig. 3a, indicates the optimal value of the fuzzifier for each run. To investigate whether $m^*$ is dependent on $c$ or on the initial values of the membership degree, the number of clusters was set to vary from 2 to 10. For each $c$ in this

range, FCM was performed 100 times with a randomly created initial fuzzy partition matrix $(U^0)$.

As shown in Fig. 3b, we do observe a relationship between $m^*$ and $c/U^0$. For smaller cluster numbers ($c = 2$ or 3) the determined optimal values of $m$ are slightly larger than those obtained for larger $c$ ($c \geq 4$). In addition, the results obtained with a smaller number of clusters



are more robust. Different repetitions always return identical $m^*$, which suggests that the initial

fuzzy partition matrix does not affect $m^*$ when the number of clusters is smaller than 4.

However, when $c$ increases to 4 or larger values, there is a variation in $m^*$ among different

repetitions, indicating that $U^0$ starts to affect the determined value of $m^*$, even though the

variation of the value of $m^*$ is small (between 1.42 and 1.52). It is not clear why different

numbers of clusters have such distinct effects on $m^*$, and answers for this question are outside

the scope of this work.

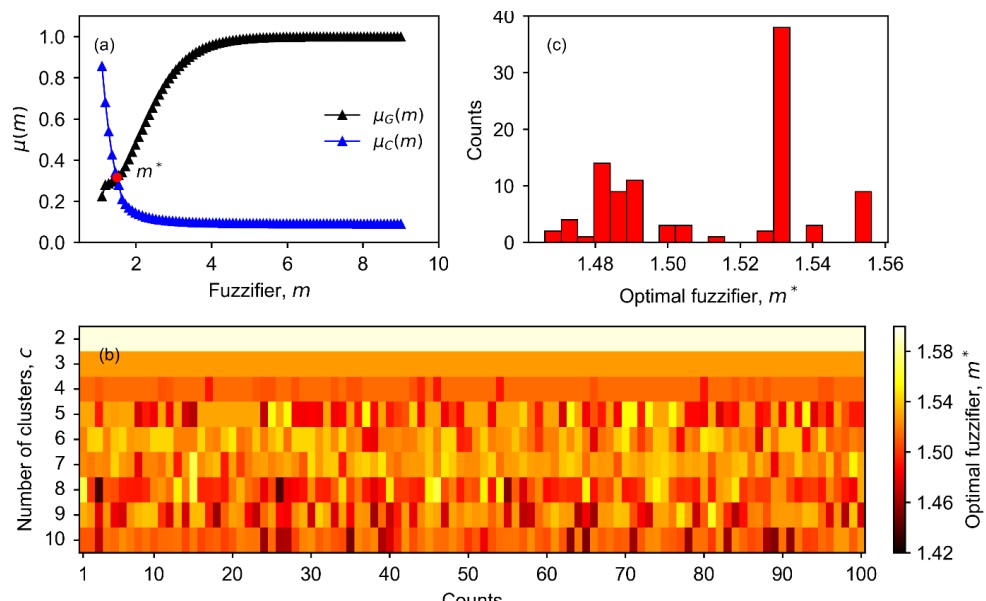

**Figure 3**. Determining the optimal value of the fuzzifier ($\boldsymbol{m^*}$) in FCM. In panel (a), the

intersection (red point) of the fuzzy objective function ($\boldsymbol{G}$) and constraint ($\boldsymbol{C}$) is determined as

$\boldsymbol{m^*}$. Panel (b) depicts the relationship between $\boldsymbol{m^*}$, the number of clusters ($\boldsymbol{c}$), and the initial

fuzzy partition matrix ($\boldsymbol{v^0}$). Panel (c) shows the frequency distribution of $\boldsymbol{m^*}$ for 100 repetitions

with $\boldsymbol{c = 5}$ (determined as the optimal number of clusters in this study).

Figure 3c displays the distribution of $m^*$ obtained from 100 repetitions with $c = 5$. The

histograms of the optimal value of $m$ with other numbers of clusters are provided in the

supplement (Fig. S3). For $c = 5$, the results suggest that the optimal value for $m$ is 1.53 in most

cases. Therefore, a value of $m = 1.53$ is used for the FCM in this study.



Overall, the number of clusters and the initial membership degree matrix do affect the optimal value of the fuzzifier that was determined based on the fuzzy decision-making method in this study, but the influence is not very strong.

**3.2 FCM clustering results**

**3.2.1 FCM of chamber data**

Using the appropriate clustering parameters determined in Sect. 2.3, we performed FCM on chamber data with the number of clusters varying from 2 to 10. For each case the algorithm was run 50 times. According to the results of these 50 repetitions, two- and three-cluster

solutions seem very robust. The repetition always gives identical outcomes despite different initial partition matrices. This is also true for the five-cluster case. However, the influence of the initial position of the cluster centers on the partition increases when the number of clusters is further increased, but in all cases at least half of the repetitions return the same results; thus, we select the most frequent outcomes as the final clustering results for each case. Here we will

not describe all solutions in detail, but instead try to formulate a synthesis of the results and present the common features shared by solutions with different numbers of clusters.

Figure 4 shows the FCM results with 2-5 clusters of the chamber data obtained during the isoprene-$NO_3$ experiment. Additional solutions with 6-10 clusters are shown in the Supplement (Fig. S4). Two distinct clusters emerge from the data for the two-cluster solution. According

to their relative formation rates, cluster 1 is regarded as first-generation cluster since species belonging to this cluster show a pronounced signal increase after addition of the reactants, while cluster 2 behaves more like second or later-generation products with its overall formation rate being much smaller compared to cluster 1. In addition to the time patterns, the mass profiles of cluster 1 and cluster 2 are clearly different (Fig. 4b).

When the cluster number is increased to 3, both, the time pattern and the mass profile of cluster 1, almost remain unchanged compared to those in the two-cluster case. It seems that mainly the former cluster 2 is separated into two new clusters (cluster 2 and 3) with different formation rates for each. Accordingly, cluster 2 is regarded as a representative of the second-generation processes, and cluster 3 represents third- or later-generation products since it

exhibits a smaller formation rate compared to cluster 2. However, the narrowing of the cluster members (with a membership degree over 0.5) of cluster 1 suggests that at least some of the former contributors of this cluster have been moved, most likely to the new cluster 2. The mass profiles of cluster 2 and cluster 3 display quite distinct features, as shown in Fig. 4b, but the





mass profiles of the two cluster 2 of the two- and the three-cluster solution match to a large

extent, even though their time patterns are somewhat different.

The effect of increasing the number of clusters from 3 to 4 can be best seen in the mass profiles (Fig. 4b). Part of the species from the former cluster 1 is separated out as a new cluster 2, dominated by molecule(s) from a very narrow mass range, where mass profile 1 also has its maximum. This migrates the former cluster 2 into cluster 3, and cluster 3 into cluster 4, shown

also by the according mass profiles 3 and 4. The time series of the new cluster 2 resembles that of cluster1, but with slowed down formation rates. In general, for all clusters the member traces seem to converge towards the time traces of the cluster centers, indicating that the system approaches the correct number of clusters.

When increasing the number of clusters from 4 to 5, the new cluster that emerges has very

small production in the early reaction stage, and its time trace shows that members in this cluster are destroyed very fast when there is abundant $NO_3$ in the system (Step IV in Fig. S1). This specific character in time seems to evolve already in cluster 4 of the four-cluster solution. As shown in Fig. 4b, the mass profiles of the first four clusters of the five-cluster solution are very similar to those of the four-cluster case, but the mass profile of cluster 5 shows distinct

differences from that of the others. It should be noted that the 5 clusters represent now also the loss rates at a time scale larger than 13h reasonably well, and that the members of most of the five clusters are well represented by the cluster centers.

When the number of clusters is further increased, more detailed and complicated clustering outcomes emerge, which is impelled by different formation and/or destruction

pathways of individual species (Fig. S4). However, the differences between the new and existing clusters become smaller. Since the major objective of this study is to demonstrate the applicability of FCM for analyzing mass spectrometric data, we will not discuss the detailed interpretation of these solutions here.

To better understand the chemical composition of clusters, the bulk chemical properties

such as hydrogen-to-carbon ratio (H:C), oxygen-to-carbon ratio (O:C), and average carbon oxidation state ($\overline{OS_C}$) of different clusters were calculated and compared. Figure 5 shows the distribution of clusters in the $\overline{OS_C}$ vs. $n_C$ space for solutions with 2 to 5 clusters. Additional results for solutions with 6 to 10 clusters can be found in the supplement (Fig. S5). The contribution of an individual species to a cluster is weighted by its nominal mass and signal

intensity in the cluster profile. Regardless of the number of clusters, different solutions cover similar chemical composition ranges in terms of average $\overline{OS_C}$ and $n_C$. However, there are





discrepancies in detail. For example, the $\overline{OS_C}$ of cluster 5 in the five-cluster solution slightly deviates from the trend the other four clusters are following. A similar behavior can be observed for cluster 1 in the six-cluster solution. This indicates that increasing the number of clusters

could help to find new groups of species with distinct chemical compositions. However, further increasing the number of clusters to 7 or more clusters does not point out new clusters with significantly different chemical composition, implying that $c = 5 \; or \; 6$ is the appropriate number of clusters in terms of separation by chemical composition. It is also shown in Fig. 5 that different clusters are well separated in the $\overline{OS_C}$ *vs.* $n_C$ space despite some overlaps,

indicating that different clusters have a distinct chemical composition. Even clusters with similar generation number, like cluster 1 and cluster 2 of the four-cluster solution, are grouped into different clusters due to their different chemical properties. In general, the early-generation clusters with lower oxidation degree fall in the corner of the plot with smaller $\overline{OS_C}$ but larger $n_C$, while the later-generation clusters with higher oxidation degree move towards the corner

with larger $\overline{OS_C}$ but smaller $n_C$, suggesting that the later-generation products are formed through further oxidation of early-generation species and undergo more fragmentation during oxidation.



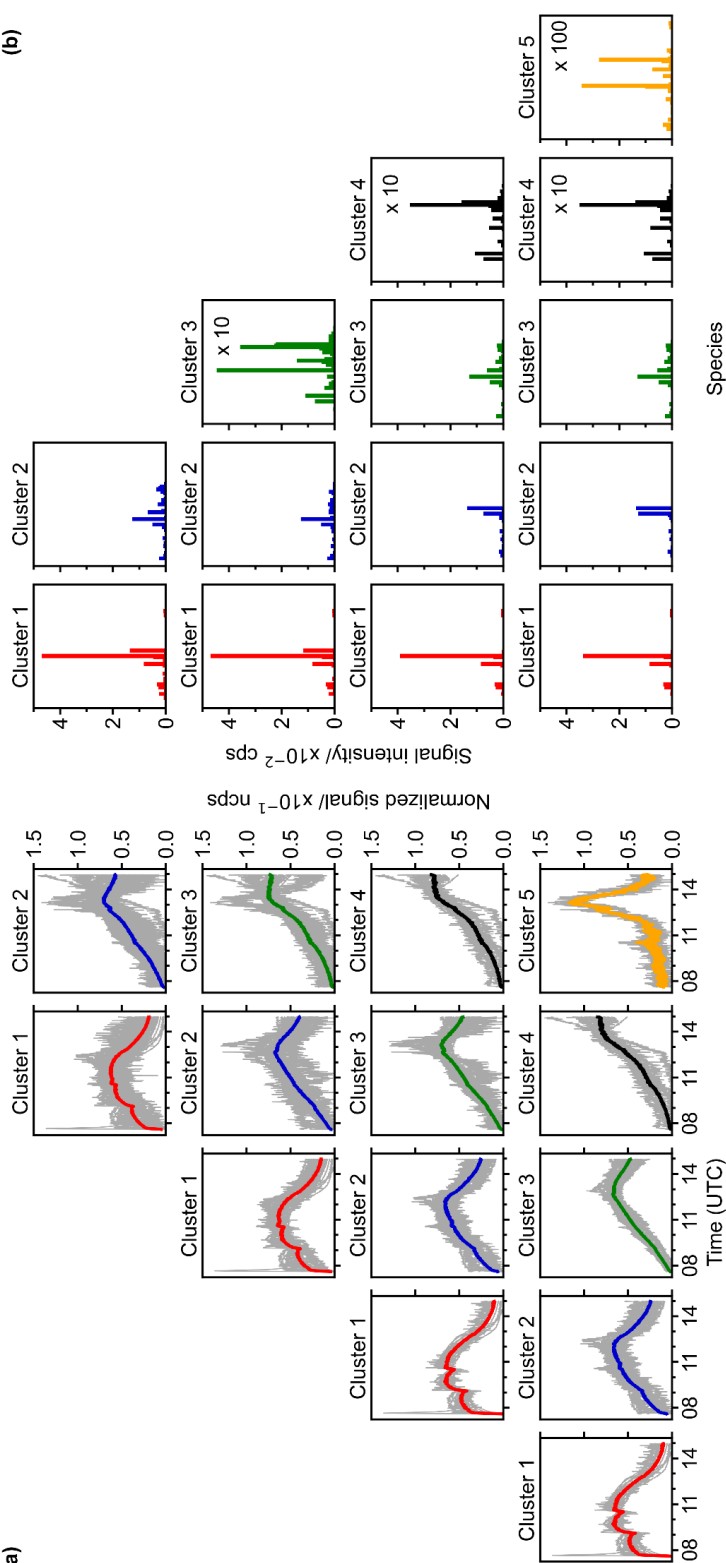

**Figure 4.** Results of fuzzy *c*-means clustering for chamber data with cluster numbers between 2 and 5: Time series (a) and mass profiles (b) of clusters for each solution (in row). The time series of cluster centers are shown as thick, colored solid lines, and the time series of species with the membership degree larger than 0.5 to the cluster are illustrated as thin, gray lines.






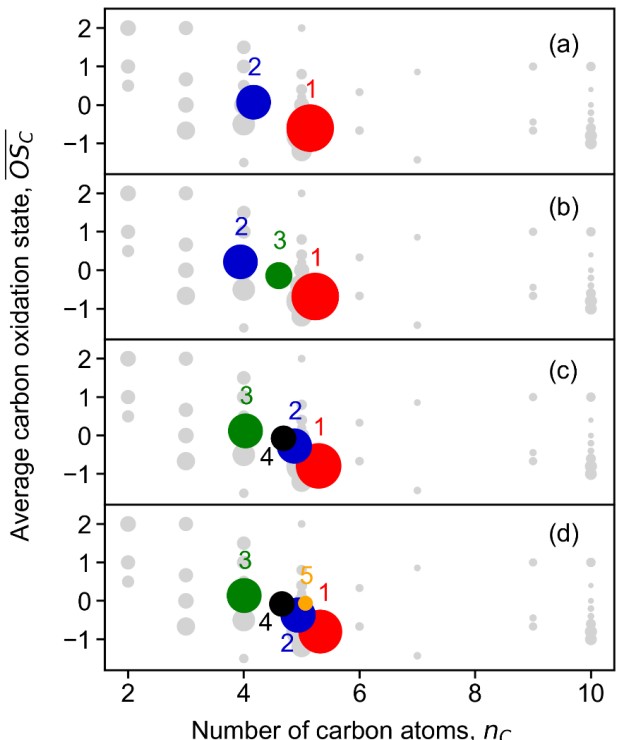

**Figure 5.** Average carbon oxidation state ($\overline{OS_C}$) of the obtained FCM clusters from chamber
data as a function of number of carbon atoms ($n_C$). Panel (a) to panel (d) show results for
solutions with 2 to 5 clusters, respectively. The color scheme follows that of the cluster centers
in Fig. 4. Individual species are shown as grey circles. Marker size is proportional to the square
root of the average intensity of the clusters.

### 3.2.2 FCM of model data

As mentioned previously, we also applied FCM onto data obtained from a box model with the

default gas-phase reaction schemes for isoprene-$NO_3$ taken from the MCM v3.3.1 (Jenkin et

al., 2015). For consistency, only closed-shell products from isoprene oxidation in MCM are

considered for the clustering. Since the reaction scheme of isoprene with $NO_3$ in the MCM

mechanism is semi-explicit, the clustering results of modelled data provides a way to evaluate

the applicability of fuzzy clustering for time series analysis. In turn, by comparing the cluster

centers derived from model data with those derived from mass spectrometric data, one can

check if the model can well reproduce the measurements, and thus investigate the

representativeness of reach mechanism coupled in the model.

Figure 6 shows the results of FCM applied to model data, again with the number of

clusters varying from 2 to 5. From the results it becomes clear that different species are sorted



according to their patterns of time behaviors, and that different clusters represent multi-generation products. Taking the 2-cluster solution as an example, the signals of most species in cluster 1 increase evidently as soon as the reaction is initiated, while those in cluster 2 grow considerably slower, indicating that cluster 1 is a surrogate of early-generation products, whereas cluster 2 corresponds to later-generation products. This is very similar to what we observe from the real measurements, even though the time behavior derived from those two cases are not the same. It seems that the fast-forming pathways are more important in the measured data than in the model data. Similarly, more later-generation clusters are screened out from the model data with increasing number of clusters, whilst the changes in early-generation clusters are not significant. Looking at clusters 3-5 in the five-cluster solution, it is evident that certain chemical loss processes are missing in the MCM mechanism, which are observed in the chamber data, however. It should be noted that autoxidation and related processes for the isoprene + $NO_3$ system are underrepresented the MCM, which is also true for the formation of accretions products.

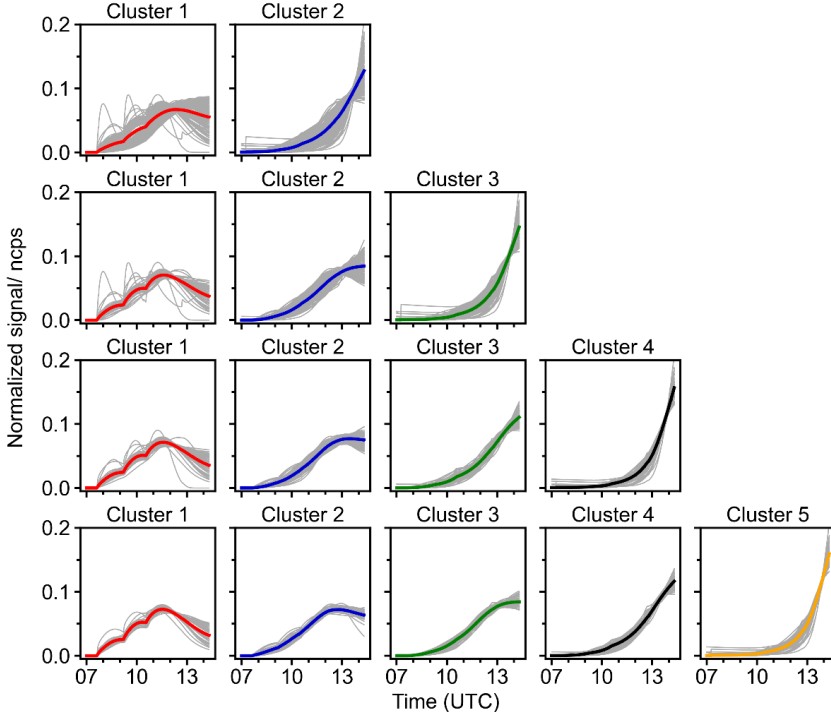

**Figure 6.** Results of FCM for model data with the number of clusters varying from 2 to 5. Each row represents one solution, with the time series of cluster centers shown in thick, colored solid lines, and species with the membership degree larger than 0.5 to the cluster illustrated as thin, gray solid lines.



As for the chemical properties, different clusters are discrete in the $\overline{OS_C}$ *vs.* $n_C$ space (Fig. S6), and thus we can conclude from the results of FCM that it will also classify product species in a reasonable way when applied to experimental data. Moreover, clusters in different solutions cover a similar chemical composition range of $\overline{OS_C}$ and $n_C$ despite increasing number of clusters (except for the two-cluster solution), well consistent with what we observed for the

chamber data. However, the $\overline{OS_C}$ of cluster decreases less prominently with increasing $n_C$ for the model data, probably due to the absence of accretion products in the MCM (mostly assigned to early-generation clusters with more carbon atoms in bulk). The MCM tends to produce more small species (with low $n_C$), which is not observed in the mass spectra data. This can be due to the detection limits of the Br⁻-CIMS for smaller compounds. Regarding the two-cluster solution,

the chemical range of clusters is much narrower, and they are overlapping in the chemical space to some extent, suggesting that the number of clusters is not enough.

       In general, according to the outcomes from the application of FCM to both measured and model data, we conclude that FCM can give interpretable and chemically meaningful results when applied to mass spectrometric data for time series analysis.

**3.3 Insights from clustering results**

**3.3.1 Chemical properties of different clusters**

In this section we will analyze the five-cluster solution to exemplify the functionality of FCM for extracting the chemical and kinetic information underlying in the mass spectrometric data. The five-cluster solution is chosen because $c = 5$ is the mathematically optimal cluster number

determined for our dataset in sect. 2.3. This does not mean that we claim it is superior to other solutions, e.g., the six-cluster solution. Besides, we confirmed in the previous sections that the FCM results exhibit general features regardless of the predefined number of clusters, so that findings based on the analysis of the five-cluster solution can hopefully also be generalized for other cases.

It can be clearly seen in Fig. 7a that different clusters are significantly different in composition. For example, cluster 1 representing the early-generation products is dominated by a single species (with the chemical formula $C_5H_9NO_5$), and its intensity is much higher than those of the other four clusters. Another characteristic of cluster 1 is that more than 80% of detected 2N-dimers (except one species with the formula $C_{10}H_{16}N_2O_{11}$) are assigned to this

cluster (Fig. S7). These compounds are obviously first-generation products formed through $RO_2 + RO_2$ reactions (Wu et al., 2021), and therefore it is reasonable to sort them into cluster 1,





which is representative for the early-generation products. Cluster 2 also behaves like early-generation products, but differs from cluster 1 in terms of reactivity, i.e., formation and destruction rates. The differences of cluster 1 and cluster 2 in chemical composition are even more perceptible. As shown in Fig. 7a, another 1N-monomer ($C_5H_9NO_6$) is present in cluster 2 with relatively high intensity besides $C_5H_9NO_5$. In addition, most of the detected small molecules ($C_{\leq 3}$) are assigned to this cluster (Fig. S7). Note that the formation rate of cluster 2 resembles that of cluster 1 in the five-cluster solution of the model data. In addition, the fractions of some 3N-dimers (e.g., $C_{10}H_{17}N_3O_{12-14}$) in cluster 2 are relatively high (Fig. S7). 3N-dimers are expected to be second or even later-generation products that are produced from the cross reaction of a first-generation nitrooxy peroxy radical and a secondary dinitrooxy peroxy radical, or from further oxidation of the corresponding 2N-dimers (Wu et al., 2021). This indicates that cluster 2 is very likely a mixture of the first- and second-generation products, which have not been resolved by FCM with the five-cluster solution. Increasing the number of clusters might help to separate the typical behavior of a minority of components. When the cluster number is increased to 6, it is indeed mainly the former cluster 2 in the five-cluster solution is further split into new clusters 2 and 3, in which the first-generation behavior of the new cluster 2 is more pronounced. From this point of view, the six-cluster solution seems better than the five one.

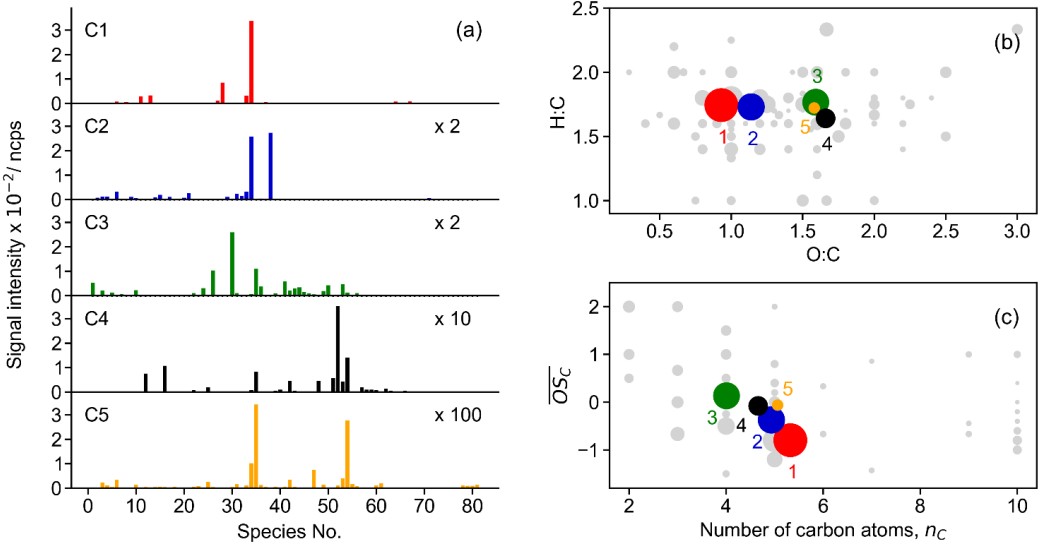

**Figure 7.** Chemical properties of clusters from the five-cluster solution. The subplots show mass profile of each cluster (a), van Krevelen plot (b), and average carbon oxidation state of clusters (c), respectively. Different clusters are distinguished by color, and the color scheme





follows that in Fig. 4. The species number in panel (a) corresponds to species listed in Fig. S7
in order. Grey circles in panel (b) and panel (c) denote species identified by CIMS. The marker
size is proportional to the square root of the average intensity of clusters/ species.

Regarding later-generation clusters, namely cluster 3, cluster 4 and cluster 5, in general
the second- or later-generation products such as C4 species, 2N- and 3N-monomers are
predominant in their composition. Nevertheless, the mass profiles of cluster 3, cluster 4, and
cluster 5 are quite distinct. For example, cluster 3 is dominated mainly by a C4 species
($C_4H_7NO_5$), while the major fingerprint of cluster 4 is constituted by two 2N-monomers
($C_5H_8N_2O_8$ and $C_5H_8N_2O_9$), a C4 species ($C_4H_7NO_6$), and a C2 species ($C_2H_3NO_5$). In addition,
3N-monomers are almost completely in cluster 4 (Fig. S7). Cluster 5 has a much lower intensity
compared to other clusters, and a distinctive characteristic of this cluster is a high attribution
of two 3N-dimers ($C_{10}H_{17}N_3O_{15}$ and $C_{10}H_{17}N_3O_{16}$) (Fig. S7).

Figure 7b and 7c show the chemical properties of clusters described by the bulk elemental
molar ratios (in the Van Krevelen space), and the average carbon oxidation state. The Van
Krevelen plot visualizes the chemical composition of organics by hydrogen-to-carbon (H:C)
*vs.* oxygen-to-carbon (O:C) ratio and is widely used to trace the origin and evolution of organic
compounds (Chhabra et al., 2011). The clusters cover a narrow range of chemical space of the
original dataset (grey spheres in Fig. 7b), but are located where most of the compounds fall in.
They lie almost along a line for H:C = 1.75 in the Van Krevelen plot, indicating that they have
gained on average one H atom compared to isoprene. A trajectory with slope zero is expected
in van Krevelen plots when only alcohol or hydroperoxide functionalities are introduced in the
molecule (Chhabra et al., 2011). This is a characteristic of autoxidation steps (-$O_2H$) or H-shifts
in alkoxy radicals (-OH, and thereafter –$O_2H$). Therefore, the distribution of the clusters in the
Van Krevelen space implies that autoxidation steps or intramolecular H-shifts were involved
in the reactions of isoprene with $NO_3$ studied in this work.

In terms of average oxidation state and carbon atom numbers, the early-generation
products which undergo less oxidation steps usually have much lower oxidation degree but
more carbon atoms per molecule. With the reaction proceeding, the early-stage products will
be further oxidized and fragmented, leading to the formation of later-generation products with
a higher oxidation state but less carbon atoms per molecule. Consequently, the trajectory of
chemical processes generally starts with the precursor in the right lower corner and moves
towards to the left upper area (products) in the $\overline{OS_C}$ *vs.* $n_C$ space through oxidation and
fragmentation. In this study, the early-generation clusters have a lower oxidation state but more





carbon atoms while the later-generation clusters are the other way around, well following the oxidation trajectory in chemical space.

Based on abovementioned results, we conclude that FCM is a feasible dimension-reduction technique for dealing with complex mass spectrometric data from an oxidation system of interest. The derived clusters show a chemical realistic time behavior and cover the major range of chemical properties of the original dataset. This suggests that FCM could be useful in simplification and analyzing mass spectra data and the chemical information underlying in the clusters and can be helpful to understand the system of interest.

### 3.3.2 Kinetic properties of different clusters

Our cluster analysis shows that the time series of the cluster centers indicate that they are formed by different (or a series of) reactions steps. By fitting the measurements to the GKP function (Eq. 12) we can extract underlying kinetic information (effective rate constant $k$ and generation number $m$) from time series data in terms of exposure to the oxidant. Generally, a larger value of $k$ implies a faster formation of a product class for a given oxidant exposure and vice versa. It should be noted that the $k$ obtained here is not a stepwise rate constant, and it has no direct relationship to the stepwise rate constants of the reaction sequence. However, this value offers a way to quantitatively measure the overall rate constant of all reactions along the pathway (Koss et al., 2020). Since the FCM cluster centers represent chemically realistic time patterns and thus retain the major chemical properties of the original dataset, they can be used as surrogates for various products formed in the isoprene-NO$_3$ system, and the GKP function can be fitted to the time series of cluster centers. This largely reduces the complexity of analysis and provides a way to get kinetic information directly from measurements.

Figure 8 shows the result of the fit of GKP to the FCM clusters derived from the chamber measurements for the five-cluster solution. All except cluster 5 fit with a coefficient of determination ($r^2$) of 0.96 or higher, indicating that the GKP model can well reproduce the kinetic behavior of the products formed from the isoprene-NO$_3$ oxidation system in this study. Cluster 5 is not well reproduced (with a $r^2$ of 0.41), probably due to its extremely low and noisy signal as a surrogate of later-generation products. The fitted values of $m$ for early-generation clusters are expected to be 1 in theory. As depicted in Fig. 8a, the generation number of cluster 1 is close to 1, and that of cluster 2 is between 1 and 2, coinciding with the expectation. As for the three later-generation clusters, their $m$ values are approximately 2 (cluster 3 and 4) or 3 (cluster 5), indicating that they undergo two or more NO$_3$ oxidation steps.





There are several possible reasons for non-integer values of $m$, including uncertainties
from signal noise, especially for low signal-to-noise data, and possible influences from physical
processes like vapor-wall interaction, which can lower the signal of species and thus lead to a
higher fitted $m$. (Koss et al., 2020). In addition, the value of $m$ can be distorted to some extent
if compounds are produced from isoprene reactions with oxidants other than $NO_3$, e.g., OH and
$O_3$ in this case. While $NO_3$ makes up the major fraction of consumption of isoprene and its
product, reactions with $O_3$ and OH still contribute for 10-15% of isoprene loss (Vereecken et
al., 2021, Carlson et al., 2022). Consequently, it is very likely that some species detected by
CIMS were oxidized by multiple oxidants. Such an effect will lower $m$ as unaccounted sources
increase the concentrations of species besides the $NO_3$ exposure, and the linear, first-order
kinetic assumption of the GKP model is no longer applicable. For example, the isoprene
hydroperoxy aldehyde ($C_5H_8O_3$), one of the major products from photooxidation, is also
observed from $NO_3$-initiated oxidation (Vereecken et al., 2021; Wennberg et al., 2018; Wu et
al., 2021). Furthermore, the deviation of $m$ from integer values can occur if isomers that were
formed by a different number of oxidation steps exist.

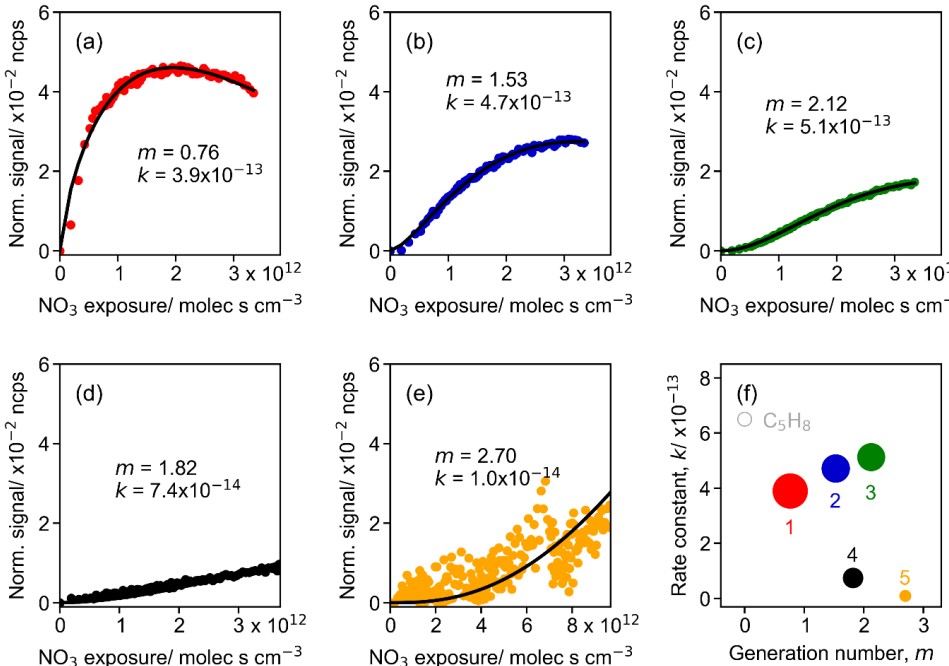

**Figure 8.** Parameterized effective rate constant ($k$) and generation number ($m$) for FCM
clusters (five-cluster case) derived from CIMS measurements of isoprene-$NO_3$ system. Panels
(a) to (e) show GKP fitting results for different clusters, with cluster 1 in red, cluster 2 in dark
blue, cluster 3 in green, cluster 4 in dark, and cluster 5 in orange, respectively. Colored dots in




each panel are time series of clusters, and black lines are GKP fits. Panel (f) shows the distribution of kinetic parameters. Marker size is proportional to the square root of the average intensity of clusters.

Since the generation number corresponds to the reaction steps with $NO_3$ to form the product, the later-generation species, which undergo more oxidation steps, should have larger $m$ values and higher nitrogen-to-carbon ratios (N:C) when considering only $NO_3$ as oxidant.

Figure 9 shows the relationship between generation number and chemical properties of clusters. In general, clusters with higher generation numbers have larger N:C ratios as expected, confirming that $NO_3$ is the predominate oxidant for isoprene oxidation in our system. Nonetheless, we find that species with larger N:C ratios are not necessarily later-generation products. As shown in Fig. 9a, cluster 4 has a larger N:C ratio than cluster 3 and cluster 5, but

turns out with a smaller $m$, which indicates that some of the nitrogen atoms of compounds in cluster 4 were gained through non-oxidative steps. On the other hand, cluster 5 has a larger $m$ value than cluster 3 and cluster 4, but its N:C ratio is relatively small. This is likely due to the species in cluster 5 being formed by reactions involving oxidants other than $NO_3$. Another possibility could be that the $NO_3$ oxidation reaction does not lead to an increase in nitrogen

content in the product molecules, e.g., through H-abstraction instead of addition to C=C double bonds (Wu et al., 2021).

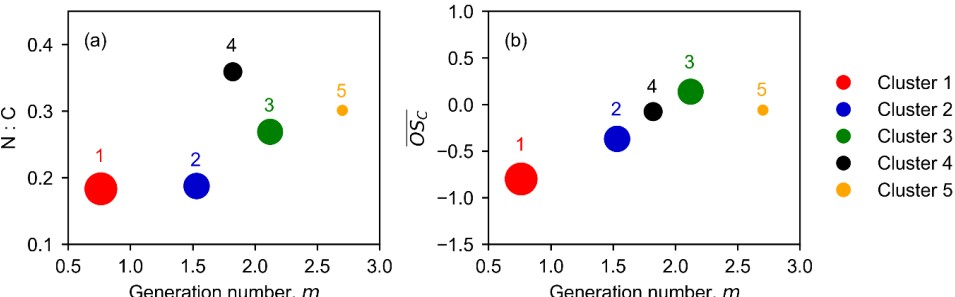

**Figure 9.** Relationship between generation number ($m$) and chemical properties of clusters: Nitrogen-to-carbon (N:C) ratio (a) and average carbon oxidation state ($\overline{OS_C}$) (b) as a function
of $m$. The marker size is proportional to the square root of the average intensity of the clusters.

There is a strong linear correlation between the generation number and the average oxidation state of the clusters apart from cluster 5, as illustrated in Fig. 9b. The early-generation clusters have smaller $m$ values than later-generation clusters, which corroborates that the generation number returned by the GKP model is reasonable. The linear regression result

shows that the value of $\overline{OS_C}$ increases by ~ 0.74 for each generation. For $m = 0$, the corresponding $\overline{OS_C}$ is −1.45, approximate to the average carbon oxidation state of isoprene





$(\overline{OS_C} = -1.6)$. For each addition of $NO_3$ functionality, the $\overline{OS_C}$ of the corresponding product increases by 0.2, and the following $O_2$ addition (if possible) results in the $\overline{OS_C}$ increasing by additional 0.8. Therefore, it involves at least one autooxidation step for each $NO_3$ addition considering an increase of about 0.8 in $\overline{OS_C}$ per generation.

Cluster 5 has a $m$ value approaching 3, suggesting that species belonging to this clusters roughly underwent three oxidation steps. However, its average oxidation rate is unexpectedly low, deviating from the linear line of $m$ and $\overline{OS_C}$. One plausible explanation is that such species are probably formed through unimolecular fragmentation. For example, if the H-abstraction (of $RO_2$) occurs at a carbon with an −OOH functionality attached, the reaction chain will be terminated by OH loss and carbonyl formation (Bianchi et al., 2019), which leads to resulting products with a lower average oxidation state.

In general, the effective rate constants of the clusters are limited by the reaction rate constant of isoprene, and the early-generation clusters have larger $k$ values than the later-generation ones. As shown in Fig. 8f, the returned $k$ values of the two early-generation clusters 1 and 2 are very close to the reaction rate constant of isoprene with $NO_3$ ($6.5 \times 10^{-13}$ $cm^3$ molecule$^{-1}$ s$^{-1}$ at 298K, IUPAC), while those of the later-generation clusters 4 and 5 are smaller. Cluster 3, which represents second-generation products with $m \approx 2$, has a similar effective rate constant as cluster 1 and cluster 2, indicating that the species belonging to this cluster form or react relatively fast. As shown in Fig. 7c, cluster 3 has a high oxidation degree, but less carbon atoms on average, suggesting that the species in cluster 3 are probably highly oxidized fragments. This is confirmed by its mass profile (Fig. 7a).

To conclude, the kinetic parameters derived from GKP fitting to the clusters are reasonable and well correlated to the chemical properties of corresponding clusters. Specifically, isoprene products formed in the early stage are larger molecules but less oxidized, with relatively high reactivity, while those formed in the later stage tend to be smaller but highly oxidized with relatively low reactivity. Fragmented species are exceptions that have a relatively high oxidation degree and reactivity simultaneously.

### 3.3.3 Characteristics of members in each cluster

Due to the fuzziness of FCM in belongingness of cluster members, only high-affiliation species (with a membership degree over 0.5) are considered as members of a cluster in the following discussion for simplicity. Figure 10 shows the chemical properties of the high-affiliation species described by their elemental molar ratios and average carbon oxidation state. In general, most of the high-affiliation species of the two early-generation clusters 1 and 2 fall in the



relatively low O:C area of the van Krevelen plot, while those from the three later-generation clusters 3, 4, and 5 are located in the higher O:C range. This confirms that species belonging to later-generation clusters are generally more oxidized than those from early-generation clusters, as expected. With respect to the average oxidation state, species of cluster 1 in general have lower $\overline{OS_C}$ than others, and they are mainly monomers ($n_c = 5$) and dimers ($n_c = 10$). The

$\overline{OS_C}$ of species from cluster 2 is slightly higher than that from those of cluster 1, and there are more fragments in this cluster, including both monomers with $n_c < 5$, and dimer species with $5 < n_c < 10$. The high-affiliation species of later-generation clusters generally have higher oxidation degree than that from early-generation clusters, but most of them are molecules with less than 6 carbon atoms.

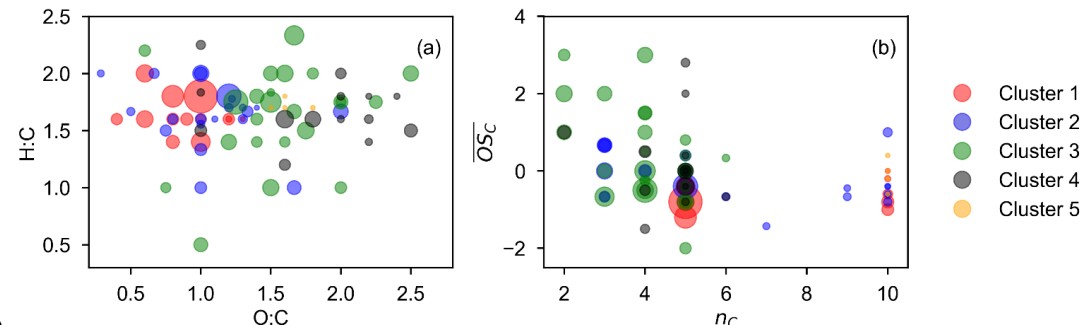


**Figure 10.** Chemical properties of high-affiliation species from each cluster (with a membership degree larger than 0.5) described by van Krevelen (a) and average carbon oxidation state ($\overline{OS_C}$) *vs.* carbon number ($n_c$) (b) plot. The marker size of species is proportional to the square root of the average signal intensity.

The gamma kinetic parameterization was also applied to individual species. Examples of fits for various species are shown in Fig. S8. Figure 11 depicts the fitted $k$ and $m$ values of all high-affiliation species from each cluster. For species from cluster 1, cluster 2, and cluster 3, most of the returned $k$ values fall in the same order of magnitude of the rate constant of isoprene with NO$_3$ ($k = 6.5 \times 10^{-13}$ cm$^{-3}$ molecule$^{-1}$ s$^{-1}$ at 298K). For those from the two later-generation

clusters 4 and 5, the returned $k$ values are about one, respectively, two order(s) of magnitude smaller. Most returned $m$ of species from cluster 1 are around 1, indicating that they are formed after one oxidation step (with NO$_3$), which is consistent with the expectation for early-generation-products. However, the returned $m$ of some species from cluster 1 are between 1 and 2, e.g., the compound(s) with the formula of $C_5H_9NO_5$ (the largest red marker in Fig. 11).

This suggests that such species may consist of isomers originating from more than one pathway, with different number of oxidation steps.




For species belonging to cluster 2 the generation numbers are mostly in a range from 1 to 2, but there are also some smaller molecules (mainly C3 and C4 species) with larger generation numbers, indicating that such fragmented compounds are formed after multiple oxidation steps. With regard to species from later-generation clusters, the returned $m$ values span a broader range, but there are no compounds with a generation number larger than 4. In general, most high-affiliation species (from both early- and later-generation) fall in the fast-reacting (large $k$) area, although a few can be observed with smaller $k$ and $m$. These two types are both kinetically realistic. However, there are individual species with large $m$ (around 3) but relatively small $k$, e.g., $C_{10}H_{17}N_3O_{15}$ and $C_{10}H_{17}N_3O_{16}$ from cluster 5. This suggests that they are slow-forming products that appear after several oxidation steps, which should be difficult to be formed and thus should be low in signal or even undetectable. In fact, the signals of $C_{10}H_{17}N_3O_{15}$ and $C_{10}H_{17}N_3O_{16}$ are extremely low and noisy at the beginning of reaction, as shown in Fig. S8(u) and Fig. S8(v). Detectable increases in signal for these masses are only observed when the $NO_3$ exposure is relatively high.

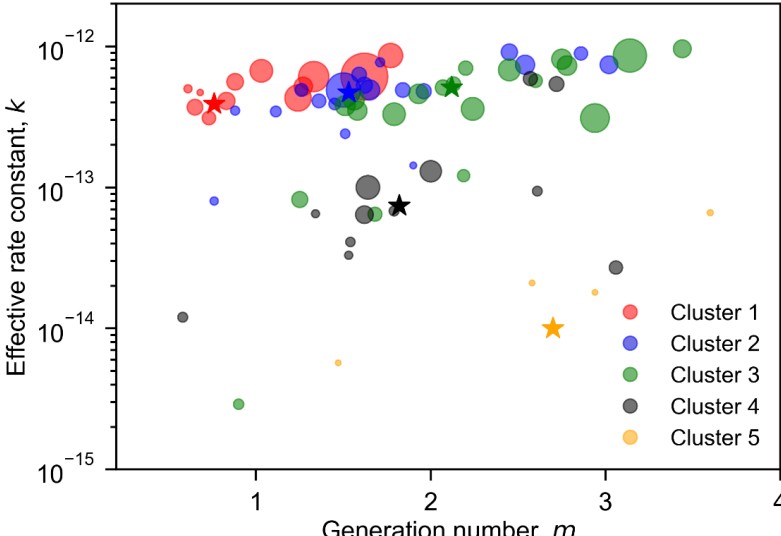

**Figure 11.** Fitted effective rate constant and generation number of the high-affiliation species of each FCM cluster. The cluster centers and members are denoted by color-coded circles and pentagrams, respectively. The marker size of individual species is proportional to the square root of the average signal intensity of species.

### 3.4 Implications to Isoprene-NO₃ chemistry

As noted previously, one big advantage of FCM is that variables can be affiliated to multiple clusters, which relates to many real-world problems in a more realistic and reasonable way. It



is elaborated in Sect. 3.3 that different FCM clusters show distinct differences in chemical and kinetic properties, potentially representing different chemical processes. Therefore, the clustering distribution of a species can give an insight into its formation mechanism.

Figure 12 shows the cluster apportionment of selected major products formed from isoprene oxidation by $NO_3$. Since each FCM cluster represents a type of chemical process or products, with distinct chemical and kinetic properties, a different distribution indicates different formation pathways of the respective species. According to the general reaction scheme of isoprene with $NO_3$ (Scheme S1), 1N- and 2N-monomers are expected to be the first- and second-generation products, respectively. The accretion products are supposed to be formed from $RO_2 + RO_2$ reaction (Berndt et al., 2018), and thus 2N-dimers are probably originating from self- or cross-reactions of two C5-nitroxy peroxy radicals, while 3N-dimers are most likely produced by cross-reactions of C5-nitroxy peroxy radicals with C5-dinitroxy peroxy radicals (Ng et al., 2008; Wu et al., 2021). Accordingly, 2N- and 3N-dimer should be first- and second-generation products, respectively. The FCM results affirm these suppositions to some extent. For example, 1N-monomer species like $C_5H_9NO_4$ and $C_5H_9NO_5$ are predominant in early-generation clusters (cluster 1 and cluster 2), while 2N-monomers are mostly found in the later-generation clusters (cluster 3 and cluster 4). However, there are some exceptions, such as $C_5H_7NO_6$ and $C_5H_7NO_7$. These two species have entirely different cluster distributions compared to $C_5H_7NO_4$ and $C_5H_7NO_5$ regardless of their similar formula composition, and the majority is apportioned to the second-generation cluster (cluster 3). This indicates that $C_5H_7NO_6$ and $C_5H_7NO_7$ should be second-generation products, while $C_5H_7NO_4$ and $C_5H_7NO_5$ are subsumed in early-generation products. A similar phenomenon is observed between $C_5H_9NO_7$ and $C_5H_9NO_{4,5}$. Another example is that of the 3N-dimers. By expectation, 3N-dimers are supposed to be second-generation products (Table S1), but the FCM outcomes show that different 3N-dimers are formed from different pathways with different generations. For example, $C_{10}H_{17}N_3O_{12}$, $C_{10}H_{17}N_3O_{13}$, and $C_{10}H_{17}N_3O_{14}$ are supposed to be early-generation products based on the FCM results, while $C_{10}H_{17}N_3O_{15}$ and $C_{10}H_{17}N_3O_{16}$ are formed at a slower rate compared to typical secondary compounds, suggesting them to be third- or even later-generation products. This implies that the formation mechanisms of 3N-dimers are more complicated than expected. Further investigation is needed to understand distinct behaviors of different 3N-dimers observed in this study. For 2N-monomers, the clustering results confirm that they are very likely second-generation products, but some species are probably originating from different formation pathways, even though they have the same generation number.



As shown in Fig. 12, most fraction of $C_5H_8N_2O_{8,10}$ fall into cluster 4, while $C_5H_8N_2O_7$ and $C_5H_{10}N_2O_{8,9}$ are preferably occupied by cluster 3. Cluster 3 and cluster 4 are different in chemical and kinetic properties, as noted in Sect. 3.3, most likely representing two chemical

processes. A similar phenomenon is observed for $C_{10}H_{16}N_2O_{11}$, which has a distinctive distribution compared to other 2N-dimers. This signifies the uniqueness of its formation mechanism.

Although a species can be apportioned to multiple clusters in FCM, most products in this study predominantly belong to one cluster, e.g., $C_5H_9NO_4$ and $C_5H_9NO_6$, suggesting that they

are dominated by a single pathway. In contrast, some species are primarily made up of two clusters, such as $C_5H_7NO_5$, $C_5H_9NO_5$, $C_5H_9NO_7$ and $C_{10}H_{17}N_3O_{12}$, which indicates that they are probably comprised of two structural isomers, or that they originate from two different reaction pathways.

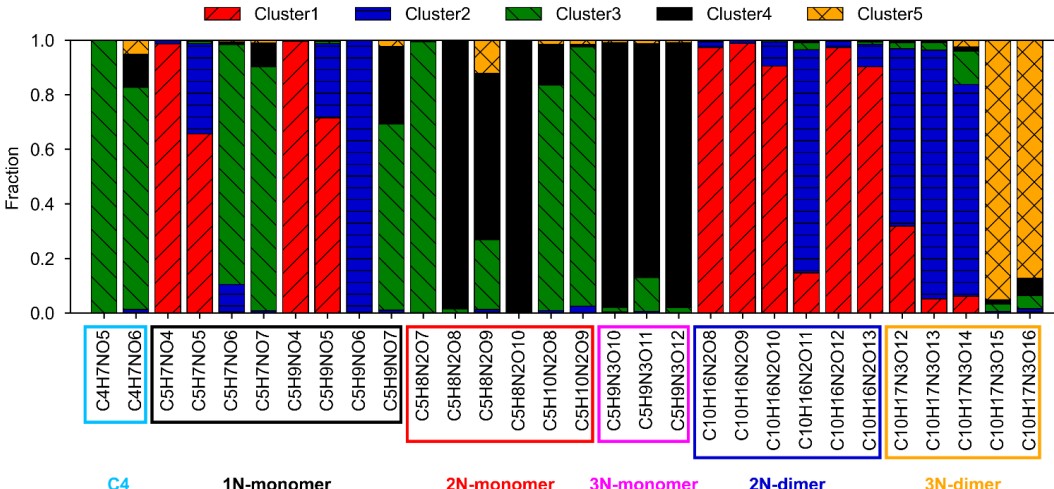

**Figure 12.** Cluster apportionment of selected major products from the isoprene-$NO_3$ oxidation system. The colored boxes correspond to different types of products.

All these findings from FCM are useful and can be used as constraints for mechanism development, especially for less-known species. For example, $C_4H_7NO_5$, a C4 species that contributes a significant fraction of the total isoprene organonitrates according to our

measurements in the SAPHIR chamber, is also ubiquitous in the real atmosphere (Tsiligiannis et al., 2022). However, it is not well-investigated, especially its formation mechanism in the nighttime (Tsiligiannis et al., 2022; Wu et al., 2021)). Only a few studies mentioned the formation processes of $C_4H_7NO_5$ in the daytime chemistry (Jenkin et al., 2015; Praske et al., 2015; Schwantes et al., 2015; Wennberg et al., 2018). According to the FCM outcomes,



$C_4H_7NO_5$ is exclusively assigned to cluster 3 (a second-generation cluster), suggesting that $C_4H_7NO_5$ is a second-generation product and is mainly originating from one pathway. Combining this information together with its molecular composition, we proposed that $C_4H_7NO_5$ is potentially formed via further oxidation of the hydroxy carbonyl ($C_5H_8O_2$) by $NO_3$, as shown in Scheme S2 in the Supplement (Wu et al., 2021). In a recent publication,

Tsiligiannis et al. (2022) have discussed the formation and fate of $C_4H_7NO_5$ in more detail based on both measurements and modelling results. They suggest that decomposition of $C_5H_8NO_7$ radicals, nitrated epoxides, or peroxides are also plausible formation pathways for nighttime $C_4H_7NO_5$. Nonetheless, the fuzzy clustering results in this study suggest that $C_4H_7NO_5$ should be formed only via one major reaction channel (or maybe an unknown

pathway) according to our chamber measurements.

## 4. Conclusions

While recent advances in mass spectrometry, especially the development of CIMS, empowers us to detect low-volatility vapors in the gas phase directly, which largely enhances our understanding of the mechanism of SOA formation, the complex, highly resolved mass spectra

introduce new difficulties for data processing and interpreting. Although different statistical analysis techniques, such as PMF, PCA, and HAC, were proposed and are widely used to analyze mass spectrometric data, the application of fuzzy clustering algorithms for CIMS data simplification and information extraction has not yet come into common view.

     In this study, we promote adopting the FCM method for the analysis of CIMS data

obtained from complex oxidation systems. Different from hard clustering algorithms, FCM allows variables to belong to multiple clusters, which is more suitable for overlapping data, and more reasonable for measurements in atmospheric science.

     Several parameters need to be defined before running FCM, some of which may have an important effect on clustering outcomes including the number of clusters, fuzzifier value, and

the distance metric used for measuring dissimilarity. By using multiple clustering validity indices, the effects of these parameters on partition were evaluated, and their optimal values were determined for our dataset. Furthermore, based on a practical case, we exemplified the functionalities of FCM in understanding the chemical and kinetic properties of the investigated system.

Overall, the FCM approach we presented in this work is an applicable and very useful tool to analyze mass spectrometric data, which can simplify the characterization of an oxidation



system by grouping numerous products into a much smaller number of clusters according to their different chemical and kinetic properties. The chemical and kinetic information retained from the clustering outcomes helps to understand the chemical processes involved in the

investigated system and can be useful for mechanism development.

**Data availability**

All data given in figures can be displayed in table or in digital form, including those given in the Supplement. Please send all requests for data to t.mentel@fz-juelich.de and r.wu@fz-juelich.de. The chamber data used in this work are available on the EUROCHAMP database,

(https://data.eurochamp.org/data-access/chamber-experiments/, EUROCHAMP, 2020) under https://doi.org/10.25326/JTYK-5V47 (Fuchs et al., 2020).

**Competing interests**

The authors declare that they have no conflict of interest.

**Author contributions**

TFM and SRZ designed the study. RW and SK collected CIMS data, and RW did the data analysis. RW and TFM wrote the paper. All co-authors discussed the results and commented on the paper.

**Acknowledgement**

The personnel of the ISOPNO$_3$ campaign is acknowledged for the help during the campaign.


**Financial support**

This research has been supported by the European Commission (EC) under the European Union's Horizon 2020 research and innovation program, (Eurochamp 2020 grant agreement no. 730997) and Horizon 2020 program societal challenges (FORCeS grant agreement

no. 821205), Vetenskapsrådet (VR, grant no. 2018-04430), and Svenska Forskningsrådet Formas (grant no. 2019-586).



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
