# Peer review of "Application of fuzzy *c*-means clustering for analysis of chemical ionization mass spectra: insights into the gas-phase chemistry of NO3-initiated oxidation of isoprene"

_EGUsphere, 2023_

## Referee Comment (RC2)

**Reviewer report on "Application of fuzzy c-means clustering for analysis of chemical ionization mass spectra: insights into the gas-phase chemistry of NO3-initated oxidation of isoprene" by Wu et al.**

In this manuscript, Wu et al. show how fuzzy c-means clustering (FCM) can be applied to mass spectrometric data. The FCM method is highly suitable for such data types where one variable/object may represent multiple different compounds (here isomers) or formation processes and thus should not be forced to belong to only a single cluster. The combination with gamma kinetics parametrisation links the clustering results to the chemical pathways and provides further insights into the reaction mechanisms relevant in the atmosphere.

The authors provide a good balance between the more technical investigation of the method to show its validity for this type of data and the scientific content with the application of the method to a case study.

The topic is highly relevant for the atmospheric science community as it provides an alternative dimension reduction technique for mass spectrometry and similar data. I recommend publication in this journal after my comments listed below are addressed.

**Major comments**

1) It is not clear how the authors treated the presence of nitrogen (N) when interpreting the elemental composition and oxidation state.
   For the interpretation of the scientific meaning of the clustering results, the authors use the average elemental composition (H:C and O:C) and the oxidation state of carbon (OSc). These are indeed important proxies for the composition of organic compounds. But in their case study, the authors use an experiment where a considerable amount of nitrogen containing compounds are formed. The presence of N in sum formulas complicates the interpretation of the elemental compositions as the O atoms can be bound either to C or to N. E.g., the two ions C5H10O3 and C5H11NO3 both have a O:C ratio of 0.6. For the second ion, it is reasonable to assume that there is a NO3 group, i.e., none of the O atoms is bound to a C. Thus, the formal value of O:C=0.6 becomes meaningless for the comparison of the degree of oxidation or the interpretation of trends in van-Krevelen diagrams between these two ions. Further, N does not necessarily occur as a nitrate (NO3) group. The authors do not clarify how they handled the presence of N when calculating OSc. Did they use Eq. 1 in Kroll et al. (2011)? What oxidation state did they assume for N? Priestley et al. (2021) encountered the same issue and suggested an algorithm to estimate the effect of N on OSc (Eq. 3 in this reference). The authors need to clarify how they handled the presence of N when calculating OSc and carefully check if their interpretations of the O:C trends are really valid when N is present in some of the ions.

2) It is not clear how the different distance metrics were used in reference to the investigated clustering validity index (CVI) metrics and how that impacts the conclusion about the usability of the different metrics.
   It is commendable that the authors investigated multiple distance metrics for their investigation (Eq. 4 - 7). But I could not derive how these different methods of determining the distance between clusters was then incorporated when using the CVIs, especially when determining which metric is the most meaningful. Eq. S1 - S4 seem to use the Euclidian distance ($\left\lVert x_j - v_j \right\rVert^2$). For Eq. S10, "a" and "b" are described as intra- and inter-cluster difference without stating how these are calculated.
   If the clustering algorithm applies any other than the Euclidian distance metric to assign the clusters, how can a CVI calculated with the Euclidian distance metric truly evaluate the quality of the clustering

result? Is the conclusion that the Euclidian metric is most suitable in FCM driven by the fact that the CVIs are calculated with that method and will then only give the "best" result if the clustering was performed using the same distance method?

The scaling of the input data was also selected to be beneficial for the Euclidean distance method. Could this have an effect of the concluded suitability of the other distance metric methods?

The authors need to clarify which distance metric was used when calculating the CVI values and how/if that impacts their conclusions.

3) The manuscript needs some reorganisation/rewriting for the methods part to make it more reader friendly.
   - Equations must appear when they are first broad up. It is extremely tedious to keep scrolling back and forth between the descriptive text and the equations at the end of the section (e.g., in section 2.3.2).
   - Equations cannot simply be dumped as a block like Eq. 1-3. They need introductory sentences identifying what these equations are about and linkage between them.
   - In other places, the reader has to wait for several sections before getting information that is only hinted about. E.g., it is mentioned multiple times that multiple runs were conducted (first in section 2.3.1). Why that was necessary, why sometimes 50 then 100, what was the difference between the runs, and how are these runs will be treated in the interpretation? Part of that information is presented much later, buried at the end of the "other parameter" section which at least for me was not an intuitive place to look for this information. The least that is needed here is a link/pointer to the section where this information will be provided.
   - Splitting section 3.3.1 and 3.3.3 feels forced. For me, the flow would make more sense to have the paragraph 875-889 with Fig 10 as part of 3.3.1. The rest would go to the end of 3.3.2

**Minor comments**

These comments are present in the order they appear in the text not by relevance.

1) They authors claim that FCM classifies the ions by their kinetic properties (e.g., in the abstract). But it is the fitting with the GKP approach that provides this classification. The authors need to emphasise that it is the combination of FCM with GKP that reveals the chemical pathway information. As an example, if there are fast 1st generation reactions and slow 1st generation reactions, the slow 1st generation reaction cluster could not be distinguished from a 2nd generation cluster by its time trace alone. Only the fit with GKP approach and the m>1 value allows that distinction and the mechanistical interpretation of the cluster results. They authors need to rephrase the related part in the manuscript to emphasize the combined effect of FCM and GKP.

2) "m" is used in Equations to mean the fuzzifier (e.g., Eq. 1) and the oxidation generation (Eq 12). The authors should consider renaming one of these parameters to avoid confusion.

3) Line 72f and 123f: The cluster or factor identification usually uses some type of analysis of the correlation of the variables. These correlations are not necessarily caused by physical or chemical reasons. We interpret them as such.

   In the studied case, the clusters represent typical chemical processes because of your measurement setup where the main reason for correlation/similarity is the chemical pathway. For ambient measurements, this may not be true. There, the source and not the reactions in the atmosphere may be the predominant driver of correlation.

4) line 88ff and 116f: Defining the "right" error matrix can indeed be a problem and cause for bias. But the error matrix intentionally added to the method to account for the uncertainty in measurements

(e.g., reduce the impact of outliers or less reliable data points). Thus, it can also be seen as an "advantage" of PMF over NMF. Also note that (Buchholz et al., 2020) is showing the error matrix bias for a specifically challenging data set (FIGAERO thermal desorption data). It is yet to be seen if the same issues can occur for "easier" time series data sets.

5) Line 76ff: I recommend splitting this paragraph into two: one for factorisation methods and one for clustering (starting at line 91).

6) Line 113ff: To put the "fuzziness" property of FCM into context, it would be helpful to specify which of the other mentioned methods allow variables/objects to participate in multiple factors/clusters /components.

7) The introduction does not contain any information about the studied VOC/SOA system. A brief paragraph about its relevance and the expected processes may be beneficial.

8) Line 336ff: The selected scaling method will shift the emphasis of the signals (maybe even as much as the error matrix can for PMF?). I guess that scaling would enhance the importance of signals with lower intensities. Is there any work how different scaling methods impact cluster identification?

9) Section 165 – 181: This section was more confusing then helpful to me. These are the questions I was left with. Most of them are not really relevant for the study as no UMR data was used (but that is not clear at that part of the manuscript yet).

   • The "500 peaks" (line 165) does not refer to the HR analysis result? I guess the authors mean UMR "peaks" in the first sentence? Why is that relevant if the study is about HR data? Rather than the "500 peaks" information, the authors should provide how many HR ions were identified (and how many were considered "product" ions).

   • Why where the UMR (?) peaks screened if they contain product ions? Shouldn't the FCM group the precursor/educt ions into separate cluster? Running FCM with all ions would then result in a higher number of clusters but the educt clusters should be distinct from the product ones. It would be yet another advantage of FCM if FCM indeed easily separates the educt ions as it would eliminate a step in the data cleaning process.

   • what is meant with pronounced changes in line 172? Increase/ decrease of signal intensity? How much would the change have to be to be pronounced?

   • Starting the sentence in line 175 with "Therefore" implies that the HR fitting was only applied because there was some uncertainty in separating the product ions when using the UMR data. What would have been the benefit to run FCM with UMR data?

   This section should be rewritten, omitting the UMR parts and only focusing on introducing and describing the HR data analysis.

10) Line 184f: What is "reasonable chemical meaning " in this context?
11) Line 189f: Is this normalisation now the same thing that is called "scaling" in lines 336-344?
12) Line 192ff: The Jenkin reference is for the Master Chemical Mechanism (MCM). Is there a reference for the actual box model? Is it the Carlsson et al 2022 reference at the end of the paragraph? If the model was "build" by the authors, should the code be made available?
13) Line 264: It is not clear what "with improvements" refers to in this sentence. Do the CVIs still need to be improved? Or they have been improved? But what is that improvement?
14) Line 369ff: This section is very difficult to follow. This is one of the examples where improvement is needed as pointed out in Major Comment 3.
    The sentence implies that $J_m$ is defined in Eq. 8 and $H_m$ is given in Eq 9. While $J_m$ is surely the Variable defined in Eq. 1, $H_m$ seem to not have a definition equation. Further, the text speaks of G and C while the equations are defining $\mu G$ and $\mu C$. The equations are given as a block instead of interlacing them with the relevant text. What are the alpha and beta constants in Eq 8 & 9?

15) Line 386ff: How does increasing the maximum number of iterations solve the issue of converging to early on a local minimum? The algorithm stops when the convergence criterium is reached. If that happens for a local minimum, increasing the maximum number of iterations should not have an effect because the criterium is already fulfilled at a low number of iterations. Increasing the number of iterations should only change the outcome if no convergence is reached within the number of allowed iterations.

16) Line 397ff: How frequent are invalid solutions? For one value of c (and m), how many of the 50 runs were disqualified. Also, was there a trend in number of invalid solutions? E.g., more valid solutions close to the "optimal" cluster number?

17) Section 2.3.4 The title "Other parameter" is misleading for this section. This section does not deal with other parameters used in FCM, but rather with the constraints and methods to improve the quality/reliability of the solutions.

18) Line 447: The authors went to the trouble of running ensembles of 50 or 100 runs. But now only one "representative" run is shown. Why not provide the average with standard deviation or interquartile range? Or using a heatmap of the distributions?
How small are considered "small variations"? E.g. compared to the dip for Vkwon and VXB for c=5.

19) Line 481: VBWS was shown to worked well for largely overlapping clusters: Is the investigated data set such a case where large overlap is expected?

20) Figure 1: I found it challenging to keep all the mentioned numbers for optimal cluster number for each CVI in my head while looking at Fig 1. It would be helpful to mark the optimal cluster number for each CVI in each panel (red circle for selected value, blue square for optimal number based on this CVI)

21) line 516f: why does this suggest that the cosine metric is more suited? Is there an assumption that the shape of VBWS should be smooth as a function of c?

22) Line 514: is it really true that the impact on clustering output is neglectable? Or isn't it rather that this metric is not sensitive to the differences? These metrics only tell us how well the solution was separated. But it does not tell us about the shape of the clusters. I.e., with the same degree of separation, the actual clusters might look different.

23) Why are the curves in Fig 1 not matching any of the lines in Fig 2?

24) Section 3.1.3 My take on what was described in this section: With small cluster number, we have too few clusters. Hence, we need to allow more overlap for variables. When we allow more clusters, we can be stricter with assigning the objects (m* gets smaller). As the solutions may be more "specific" we are now more sensitive to "local" minima. As those are driven by the starting point, we get more sensitive to the U0.

25) Section 3.1.3 How much does a difference of 0.1 in m really change the clustering results? I.e., how sensitive is the actual result to a slightly different m value? ~1.5 seems much closer to 1 (=non fuzzy) than to 9 (upper limited in this study)

26) Line 565ff: Do the authors have any idea why the 5 cluster case was super stable, but the 3, 4, and higher ones varied more? Could this be an indicator for the "perfect number of clusters"? Or just coincidence?

27) Line 592f "Part of the species from the former cluster 1 is separated out as a new cluster 2, dominated by molecule(s) from a very narrow mass range, where mass profile 1 also has its Maximum" – I do not understand this sentence.
However that sentence is meant, while the mass spectra of C1 and C2 may look similar, the time series are not. C2 is not directly linked to the injections (peak is later). Or are the authors implying that the C1 of the 3cluster solution had some reaction products grouped in which are now taken out?

28) Line 635ff: Could it also be that compounds which would retain the nC and get more functionalised are too low volatile to remain in the gas phase? Where particles formed in these experiments? If not, could low volatility compounds be lost to the chamber walls?

29) Fig 6: Cluster 1 has the most "outliers" while the other clusters seem to have the grey lines closer together. Do the authors have any explanation for this higher "spread" of members in Cluster 1?

30) Line 713f & 717f: These sentences seem contradictive. The first sentence says that C1 and C2 differ in their formation& production rate. Second sentence says that formation rate of C2 resembles that of C1. Are they different or similar?

31) Figure 5 & S5: The meaning of the grey circles is not clearly explained when the Figure is first mentioned. What are the "individual species"? What does their marker size refer to? Is it the average mass spectrum over the full experiment? Who do the size of the grey markers relate to the size of the cluster markers?

32) Line 750: The fact that the cluster markers cover a smaller space than the original data: 1) that is a logical consequence when doing clustering. The result will not be on the edge of the distribution. 2) this graph is not comparing the range of OSc vs nC for the clusters. Only the average of the cluster is given. But to check the range, one would have to see the position of all species contributing to each factor and evaluate those.

33) Line 766: assuming that the clusters are sorted by their oxidative "age", the statement is only partially true. 1 – 2 – 3 follow the trend . 4 is on the "line" but has higher nC than 3. Cluster 5 is off the line.

34) line 777: It is not the measurements that are fitted to the function, but the function is fitted to the data points. But which parameters in Eq 12 are "free fit"? k and m? Also a?

35) How are products of auto-oxidation classified in the GKP approach? The intermediate stage of one oxidation step with NO3 (=first generation) can lead to a range of products with varying number of autooxidation steps. Are such products still classified as "first generation"? How will varying degree of autooxidation affect the "m" parameter?

36) Line 784f "chemically realistic time patters" what is meant by that? That the time series looks like it could be from a combination of realist reactions? What is meant by chemical properties here?

37) Line 784f: What is the issue with Cluster 5? That it is so low intensity? Or that it is so noisy? Or is it the "unusual" shape? Are the authors not trust this "unchemical" shape?

38) isomer theory. If one ion represents isomers from different steps, shouldn't those isomers than be assigned to different clusters? I.e. the ion should show up in a first and second gen cluster
The example C5H9NO5 shows exactly that. It has a considerable contribution to C1 and to C2 (but there it falls under the 0.5 mark)

39) Fig 11: The position of the red star feels odd. The position seems to be at a "too low" m value. The 5 individual highest signals all have much higher m values. For the other clusters, the star falls more in the middle of the point distribution. If the position of the red star is correct, it should mean that the m and k values of the red cluster are not represented well by the high affiliation species but rather dominated by the species with lower affiliation.

40) The existence of C5 N1 monomers (e.g. C5H7NO6) as second-generation products confirms the importance of H abstraction by NO3 radicals as oxidation mechanism. Or are the authors suggesting a different mechanism that does not result in addition of the NO3 radical to the molecule? Anyhow, the reaction scheme in Scheme S1 does not consider any such products, only dual NO3 addition products are listed.

41) Line 986 & Scheme S2: In my book, there are two pathways to form C4H7NO5: one via 1,4 h shift and one via +RO2 reaction. The split ratio will depend on the RO2 conc. But for the formation kinetics, the rate limiting step is relevant. Since that is most likely the first NO3 addition, no differentiation between the two paths is possible (i.e., it is impossible to determine the split ratio). This shows the limitation of this approach. If a product is formed through different pathways, but the rate limiting

step is common, FCM coupled with GKP will only provide the sum over those paths. This comment also relates to the claim in Line 970 that those compounds are formed by a single pathway.

42) SI section (1) What is meant by the "knee"? How is it determined? Is it the turning point of the curve? In Fig 1 a) c=5 is chosen? But why? What is the math behind that? F it is "by eye" I could also take c=6 as "knee" or elbow or other bent body part.

43) SI section (2) The equation of $V_{FS}$ is of the shape $V_{FS} = A-B$. A smaller value of VFS would only indicate that A and B become more similar. That means that a bad compactness value (high A) could be compensated by a larger difference between the clusters (high B)

44) Equation S2 would be much easier to read if they introduce a variable for cluster compactness and separation of cluster. Especially since these variables are used again for VXB and later. The similarities/differences between the CVIs will be much easier to see using variables instead of the lengthy double sums.

45) SI section (2): What are recommended values to identify a good solution for this CVI? From Fig 1b, VFS seems to always go down with cluster number until reaching a minimum at ~-8. If smaller is better, why is c=5 chosen?

46) SI section (3): " the smaller the numerator…" this sentence is talking about the individual clusters. But the formula is summing over all clusters. I think this should be changed to plural. So, the more compact the clusters are. The more the clusters are separated.

47) SI section (4): Eq S4 will become much more readable if the "punishing function" is defined in its own equation.

48) SI section (4): If c approaches n, the key point about using a clustering algorithm (dimension reduction) is not achieved. Why is a metric needed that works at c approaching n? In Fig 1 c and d are identical in shape. What is the added values? TO enhance clarity of the already complicated manuscript $V_{kwon}$ or $V_{XB}$ should be omitted.

49) SI section (5) This section implies that overlapping clusters are not treated well with all previous metrics. Is that indeed the intention?

50) Higher $V_{BWS}$ values indicate a better solution. Fig 1e looks like there is a "maximum" in the curve. Is that a feature of this CVI?

51) SI section (6) "The average cluster silhouette score can tell if the cluster is appropriately configurated or not. " Isn't it a problem if there are an equal number of bad assigned and well assigned ones? Because the positive and negative cancel each other out?

52) how are a and b calculated for equation S10? Section 2.3.2 introduces 4 ways of calculating the difference. Which one is used here?

**Language and technical comments**

General: The authors should carefully check their manuscript for adverbial constructions/inserts at the start of a sentence and decide if they follow the recommendation of separating them by a comma from the main clause. E.g., the sentence in line 57ff ("Benefitting from this it has….") should have a comma after "this". Personally, I like using the comma for this grammatical structure as it enhances readability.

Line 26: "an approach by using FCM" -> omit the "by"

Line 32 "system investigated" -> investigated system

Line 32ff "chemical properties were characterised… ": characterised and parameterised can be used in this sentence, but the term "described" may be better in this context.

Line 44: "…and convert to condensable vapors" Not all products of atmospheric VOC oxidation are condensable. In most cases, the majority will be still too volatile. -> rephrase

Line 46: add comma before "and thereby" to indicate that that is referring to both condensation and nucleation.

Line 47: SOA was already introduced in the sentence before.

Line 55: "propagation": is that the right word here?

Line 67: "nonwithstanding the apparatus of high resolution": this insert is not clear. What is the apparatus of high resolution? Do the authors mean an instrument with high resolution?

line 73f: "thus simplify the chemistry of the investigated system" what is meant by that? Clearly, the actual chemistry does nt change? It is just our representation of the chemical processes that is simplified?

Line 77f: "best-known approach": in my opinion "most commonly used approach" seems more appropriate here.

Line 74 and later: (Buchholz et al., 2019) is the wrong reference. The authors most likely mean (Buchholz et al., 2020)

Line 148: "the injection was repeated…": Using the term "injection" can be easily misunderstood in this sentence. The previous sentence only calls the addition of isoprene an injection and nothing about the delivery of NO2 and O3. -> rephrase

Line 154: "particle-phase" no hyphen in this case as it is a noun (but gas- needs the hyphen)

Line 169 "related to ion source" -> related to the ion source

Line 273: "There're" -> there are

Line 278: "…among all the alternatives following…" -> alternatives the following

Line 322: "are called as distance" -> omit "as"

Line 331: "Since it is difficult…." sentence: comma after "data"

Line 379: "Where b a constant that is usually set to be 10 in practice": The grammar of this sentence seems broken.

Line 393: "The clustering results of FCM is…" -> "The results are…"

Line 409: "produce target compound" -> produce the target(ed) compound

Line 409: "The kinetic information…" sentence: comma after "species" to close the insert

Line 411: "involve" should be include

Line 485 "Crisp Silhouette (C)": in the next sentences CS is used as an abbreviation

Line 533 "is dependent" should be "depends"

Line 599: Which one is meant by "new " cluster? C5 (yellow)?

Line 630f "Even clusters with similar generation number […] are grouped into different clusters due to their different chemical properties." I do not understand this sentence. Clusters are grouped into clusters?

Line 668f: What is meant by "screened out"? Do you mean identified? Selected?

Line 685f: The trend description in this sentence (for the model data) may be easier to understand if presented with in the opposite way. I.e., speaking of the less pronounced increase of OSc and decrease of nC with increasing chemical age of the clusters.

Line 698: "information underlying in the mass spectrometric data.": this sounds incorrect. It probably should be "underlying information in the mass spectrometric data"

Line 744: Is "attribution" the correct word here or should it be "contribution"?

Line 726f "it is indeed mainly the former cluster 2 in the five-cluster solution is further split into new clusters 2 and 3" -> it is indeed … which is further split…"

Line 826 "larger N:C values as expected" the "as expected" sounds a bit weird in this context. Using a comma to indicate the "as expected" as a grammatical insert will help. Or putting the "as expected" at the start of the sentence.

Line 856f "resulting products" -> omit resulting

Line 875 "fuzziness of FCM in belongingness of cluster members": the term "belongingness" does not work here. Maybe change to "assignment of cluster members"

Line 895 "gamma kinetic parametrisation" was already introduced -> use GKP here

Line 951 & 962 Using the comma separated values for oxygen in the sum formulas confused me at first. (C5H9NO4,5). Since there are only two cases where this nomenclature is used, consider writing both formulas out.

Line 963: "preferably occupied by cluster 3" this sounds incorrect. I am not sure what is meant by occupied in this context

Line 1001: the acronym "HAC" was not introduced

Figure 2a has a y scale up to +/-600? FS value is -2 - -8 in Fig 1f -> reduce scale in Fig 2a

Fig S1: left y axis labels are cut off

What is Table S1 for? I did not notice any reference to this table in the main manuscript or the SI.

SI paragraph (6) "…was first proposed by Rousseeuw (1987), which can be used… " it is not clear where the "which" is pointing to. Better link the two parts with "and" -> "…was first proposed by Rousseeuw (1987) and can be used… "

SI paragraph (6) "With different cluster number…" sentence is difficult to understand. -> consider rephrasing

Fig S7: it was difficult for me to line up the markers for, e.g., cluster 1 with the names on the x axis. For me, some vertical grid lines every X ions would help to make this figure more readable.

Fig7 and S7: What is the sorting criterium for the species in these figures? Ion mass? Consider if that is the optimal way of presenting the data or if another order (e.g. by C number) would be beneficial.

**References**

Buchholz, A., Lambe, A. T., Ylisirniö, A., Li, Z., Tikkanen, O.-P., Faiola, C., Kari, E., Hao, L., Luoma, O., Huang, W., Mohr, C., Worsnop, D. R., Nizkorodov, S. A., Yli-Juuti, T., Schobesberger, S., and Virtanen, A.: Insights into the O:C dependent mechanisms controlling the evaporation of a-pinene secondary organic aerosol

particles, Atmos. Chem. Phys. Discuss., 1–21, https://doi.org/10.5194/acp-2018-1305, 2019.

Buchholz, A., Ylisirniö, A., Huang, W., Mohr, C., Canagaratna, M., Worsnop, D. R., Schobesberger, S., and Virtanen, A.: Deconvolution of FIGAERO–CIMS thermal desorption profiles using positive matrix factorisation to identify chemical and physical processes during particle evaporation, Atmos. Chem. Phys., 20, 7693–7716, https://doi.org/10.5194/acp-20-7693-2020, 2020.

Kroll, J. H., Donahue, N. M., Jimenez, J. L., Kessler, S. H., Canagaratna, M. R., Wilson, K. R., Altieri, K. E., Mazzoleni, L. R., Wozniak, A. S., Bluhm, H., Mysak, E. R., Smith, J. D., Kolb, C. E., and Worsnop, D. R.: Carbon oxidation state as a metric for describing the chemistry of atmospheric organic aerosol, Nat. Chem., 3, 133–139, https://doi.org/10.1038/nchem.948, 2011.

Priestley, M., Bannan, T. J., Le Breton, M., Worrall, S. D., Kang, S., Pullinen, I., Schmitt, S., Tillmann, R., Kleist, E., Zhao, D., Wildt, J., Garmash, O., Mehra, A., Bacak, A., Shallcross, D. E., Kiendler-Scharr, A., Hallquist, Å. M., Ehn, M., Coe, H., Percival, C. J., Hallquist, M., Mentel, T. F., and McFiggans, G.: Chemical characterisation of benzene oxidation products under high- And low-NOx conditions using chemical ionisation mass spectrometry, Atmos. Chem. Phys., 21, 3473–3490, https://doi.org/10.5194/acp-21-3473-2021, 2021.

---

## Author Comment (AC1)

The reviews of our manuscript are thorough and well-considered. We would like to thank the reviewers for their careful reading and valuable comments, as well as the improvements they help us make.

All the suggestions and comments from Referee 2 are addressed below point by point in bold text, followed by our responses in non-bold text. The corresponding revisions to the manuscript are marked in blue. All updates to the original submission are tracked in the revised manuscript.

**In this manuscript, Wu et al. show how fuzzy c-means clustering (FCM) can be applied to mass spectrometric data. The FCM method is highly suitable for such data types where one variable/object may represent multiple different compounds (here isomers) or formation processes and thus should not be forced to belong to only a single cluster. The combination with gamma kinetics parametrisation links the clustering results to the chemical pathways and provides further insights into the reaction mechanisms relevant in the atmosphere.**

**The authors provide a good balance between the more technical investigation of the method to show its validity for this type of data and the scientific content with the application of the method to a case study.**

**The topic is highly relevant for the atmospheric science community as it provides an alternative dimension reduction technique for mass spectrometry and similar data. I recommend publication in this journal after my comments listed below are addressed.**

We appreciate the positive feedback from the referee.

**Major comments**

1) **It is not clear how the authors treated the presence of nitrogen (N) when interpreting the elemental composition and oxidation state.**

**For the interpretation of the scientific meaning of the clustering results, the authors use the average elemental composition (H:C and O:C) and the oxidation state of carbon (OSc). These are indeed important proxies for the composition of organic compounds. But in their case study, the authors use an experiment where a considerable amount of nitrogen containing compounds are formed. The presence of N in sum formulas complicates the interpretation of the elemental compositions as the O atoms can be bound either to C or to N. E.g., the two ions C5H10O3 and C5H11NO3 both have a O:C ratio of 0.6. For the second ion, it is reasonable to assume that there is a NO3 group, i.e., none of the O atoms is bound to a C. Thus, the formal value of O:C= 0.6 becomes meaningless for the comparison of the degree of oxidation or the interpretation of trends in van-Krevelen diagrams between these two ions. Further, N does not necessarily occur as a nitrate (NO3) group. The authors do not clarify how they handled the presence of N when calculating OSc. Did they use Eq. 1 in Kroll et al. (2011)? What oxidation state did they assume for N? Priestley et al. (2021) encountered the same issue and suggested an algorithm to estimate the effect of N on OSc (Eq. 3 in this reference). The authors need to clarify how they handled the presence of N when calculating OSc and carefully check if their interpretations of the O:C trends are really valid when N is present in some of the ions.**

**Response:** Thanks for the valuable comment! We fully agree with the referee. The presence of N complicates the evaluation of elemental composition and the oxidation state of carbon, because it is difficult to tell the oxidation state of N in each detected molecule.

In this study, we assume that all the N atoms exist as nitrate groups and thus the oxidation state of N is +5. The $\overline{OS_C}$ is indeed estimated following the method proposed by Kroll et al. (2011), as the referee suspected. This has been clarified in our previous study (Wu et al., 2021). We have added a reference to this in the revised manuscript. The assumption of $OS_N = +5$ is reasonable since the oxidation of isoprene in our system was mainly initiated by $NO_3$ radicals (Wu et al., 2021). Therefore, for the N-containing species, like the 1N-monomers, the N atom is most likely present in a nitrate group, in which one of the O is bonded to the C atom while another two are bonded to N. As a

result, the oxidation state of nitrate group is -1. Of course, we cannot completely exclude the occurrences where the N atom exists as a cyanate (with $OS_N = 0$) or nitro group (with $OS_N = +3$), but these are less possible in our case.

Priestley et al. (2021) investigated benzene oxidation by OH radicals under different $NO_x$ conditions, which is however different from our system. In their system, it's also probable that N presents in a nitro group, such as nitrophenol and nitrocatechol, one of the major types of oxidation products of benzene under high $NO_x$ conditions (Cai et al., 2022; Wang et al., 2019). In addition, the Priestley method can only provide a limit to the calculated $\overline{OS_C}$, so the relative distribution of each cluster in terms of $\overline{OS_C}$ would not change. Further, in this study, we are more interested in the differences of clusters in chemical composition, rather than the exact composition of each cluster.

Taking all this into consideration, and also to keep consistent with our previous study (Wu et al., 2021), we prefer to keep the method we used for $\overline{OS_C}$ estimation, that is, assuming all N atoms present in nitrate groups with the $OS_N = +5$.

As for the O:C ratio, we agree that the O:C ratios of $C_5H_{10}O_3$ and $C_5H_{11}NO_3$ should be different if assuming the N exists in a nitrate group (one of O atom is bound to C, not none). It is more appropriate to use the concept of effective oxygen number (two of the O atoms bonded to the N atom in a nitrate group are subtracted, that is, $n_{O\_eff} = n_O - 2 * n_N$) when calculating the O:C ratio of N-containing compounds (Xu et al., 2021). If using $O_{eff}$:C instead of O:C, the positions of different cluster centers are mainly horizontally shifted (to the left side) and they are much closer to each other in the van-Krevelen plot (see the updated plots below). However, the conclusions we made according to the distribution of clusters and species in the van-Krevelen plot do not change.

To sum up, in calculation of $\overline{OS_C}$ in this study, we assume that all N atoms of N-containing compounds present in nitrate groups, and follow the method proposed by Kroll et al. (2011) ($OS_N = +5$). When exploring the relative distribution of species/ clusters in the van-Krevelen diagram, $O_{eff}$:C ratio is used instead of O:C ratio, where in the case of a nitrate group, only one of the O atoms bonded to C atom is considered.

To clarify these, the following sentence "The $\overline{OS_C}$ of each cluster was calculated following the method proposed by Kroll et al (2011), in which all the N atoms of N-containing compounds were assumed to be present in nitrate groups (and thus $OS_N =$

+5), as descried in our previous study (Wu et al., 2021)." has been added in the revised manuscript in L768-771.

And the sentence "When calculating the O:C ratios of N-containing compounds, the concept of effective oxygen number ($n_{O\_eff}$, $n_{O\_eff} = n_O - 2 * n_N$) was employed, where in the case of a nitrate group, only one of the O atoms bonded to C atom was considered in the calculation (Xu et al., 2021)." has been added in the revised manuscript in L933-936.

Figure 7 and Fig. 10 have been updated accordingly, and the revised plots are shown below:

[Figure]

**Figure 7.** Chemical properties of clusters from the five-cluster solution. The subplots show mass profile of each cluster (a), van Krevelen plot (b), and average carbon oxidation state of clusters (c), respectively. Different clusters are distinguished by color, and the color scheme follows that in Fig. 4. The marker area of clusters is proportional to the sum of the average signal intensity of all species in the cluster weighted by their membership degrees. The species number in panel (a) corresponds to species listed in Fig. S7 (in order of molecular mass). Grey hexagons in panel (b) and panel (c) denote species identified by Br⁻ CIMS, and the marker area is proportional to the average intensity of species over the whole experiment.

[Figure]

**Figure 10.** Chemical properties of high-affiliation species from each cluster (with a membership degree larger than 0.5) described by van Krevelen (a) and average carbon oxidation state ($\overline{OS_C}$) vs. carbon number ($n_C$) (b) plot. The marker area is proportional to the average signal intensity of species over the whole experiment.

2) **It is not clear how the different distance metrics were used in reference to the investigated clustering validity index (CVI) metrics and how that impacts the conclusion about the usability of the different metrics.**

   **It is commendable that the authors investigated multiple distance metrics for their investigation (Eq. 4 - 7). But I could not derive how these different methods of determining the distance between clusters was then incorporated when using the CVIs, especially when determining which metric is the most meaningful. Eq. S1 – S4 seem to use the Euclidian distance ($\|x_j - v_j\|^2$). For Eq. S10, "a" and "b" are described as intra- and inter-cluster difference without stating how these are calculated.**

   **If the clustering algorithm applies any other than the Euclidian distance metric to assign the clusters, how can a CVI calculated with the Euclidian distance metric truly evaluate the quality of the clustering result? Is the conclusion that the Euclidian metric is most suitable in FCM driven by the fact that the CVIs are calculated with that method and will then only give the "best" result if the clustering was performed using the same distance method?**

**The scaling of the input data was also selected to be beneficial for the Euclidean distance method. Could this have an effect of the concluded suitability of the other distance metric methods?**

**The authors need to clarify which distance metric was used when calculating the CVI values and how/ if that impacts their conclusions.**

**Response:** The distances of related CVIs were all calculated based on the Euclidean distance method. The reason for this is that we aim to evaluate the impacts of different distance metrics on clustering outcomes by using the CVIs. For a specific clustering result, one would get different CVIs values if applying different distance metrics in the calculation. Therefore, if we use the same distance metric is CVIs calculation as that employed in clustering, we cannot tell where the differences in CVIs values come from. It's challenging to differentiate whether the differences in CVIs are attributed to different clustering outcomes (with using different distance metrics), or they are simply due to the distance metrics selected in CVIs calculation. Therefore, Euclidean distance was used in the calculation of various CVIs.

However, we agree that it also makes sense to keep the distance metric used in CVIs calculation consistent with that utilized in clustering. So we have also calculated the CVIs using identical distance metric to clustering. The results are shown in the plot below. Compared to Fig. 2, we can see from the figure below that some of the calculated values of CVIs, such as $V_{FS}$ and $V_{Kwon}$, are extremely small or large, indicating a more significant impact introduced by using different distance method in the calculation of CVIs, than the impact of different distance metrics on clustering results. The extreme results of $V_{FS}$ and $V_{Kwon}$ obtained using cityblock and correlation method suggest that these two metrics might be inappropriate for the calculation. This indirectly reflects that employing Euclidean distance in the calculation of all CVIs is justified to some extent.

To clarify the distance method used in the calculation of CVIs, a sentence "Additionally, Euclidean distance was used in the calculation of various CVIs." has been added in L653-654. This is also clarified in the caption of Fig. 2.

[Figure]

**Figure .** Values of selected clustering validity indices $V_{FS}$ (a), $V_{Kwon}$ (b), $V_{BWS}$ (c), and $FS$ (d) as a function of the number of clusters. Points in different colors are results obtained with different distance or similarity metrics. The averages of results from 50 repetitions are shown in the plot, and the error bars denote the standard deviations. The distance method used in the calculation of CVIs was consistent with that used for clustering.

For "a" and "b" in Eq. S10, as described in the text below the equation, $a_j$ is the average distance of object $x_j$ (belonging to cluster $i$) to all other objects in the same cluster, while $b_j$ is the average distance of object $x_j$ to all objects in its closet neighboring cluster $r$ ($r \neq i$). According to the definitions, $a_j$ is indeed a compactness distance, while $b_j$ is a separation distance.

To clarify how they are calculated, following content has been added in the revised supplement:

"In a hard partition, each object is exclusively partitioned to one cluster, and it is easier to determine the intra- (within-cluster) and inter- (between-cluster) distances. With regard to a fuzzy partition, however, an object could belong to multiple clusters simultaneously, and its affiliation to each cluster is measured by the membership degree. In order to determine the intra- and inter-distance of an object in a fuzzy partition, the original definition of silhouette is reformed by introducing a concept of intra-inter scores. The intra-score matrix is defined by

$$IntraDist_i = [intra_i(d_{jk})], \quad 1 \leq j \neq k \leq n, 1 \leq i \leq c \tag{S13}$$

where $intra_i(d_{jk}) = (u_{ij} \wedge u_{ik})$.

And the inter-score matrix is given by

$$InterDist_{ir} = [inter_{ir}(d_{jk})], \quad 1 \leq j \neq k \leq n, 1 \leq i < r \leq c \tag{S14}$$

where $inter_{ir}(d_{jk}) = (u_{ij} \wedge u_{rk}) \vee (u_{rj} \wedge u_{ik})$.

$u_{ij}$ and $u_{ik}$ are the membership degree of object $x_j$ and $x_k$ to cluster $i$, and $u_{rj}$ and $u_{rk}$ are the membership degree of object $x_j$ and $x_k$ to cluster $r$, respectively.

With the intra- and inter-distance scores defined above, we can calculate the intra-distance $a_j$ and inter-distance $b_j$ as follows:

$$a_j = \min\left\{\frac{\sum_{k=1}^n IntraDist_i(j,k)d(x_j,x_k)}{\sum_{k=1}^n IntraDist_i(j,k)}\right\}, \quad 1 \leq j \neq k \leq n, 1 \leq i \leq c \tag{S15}$$

$$b_j = \min\left\{\frac{\sum_{k=1}^n InterDist_{ir}(j,k)d(x_j,x_k)}{\sum_{k=1}^n InterDist_{ir}(j,k)}\right\}, \quad 1 \leq j \neq k \leq n, 1 \leq i < r \leq c \tag{S16}$$

where $IntraDist_i(j,k)$ and $InterDist_{ir}(j,k)$ are the intra-, and inter-distance score of the object $x_j$, respectively, as defined in Eq. S13 and Eq. S14, and $d(x_j,x_k)$ represents the distance between the object $x_j$ and $x_k$.".

For the last point, we would say, yes, normalization could affect the clustering outcomes, since this treats all compounds equally and thus raises the importance of molecules with low signal intensities. However, one of the objectives of this study is to better understand the multigeneration chemistry of isoprene-NO$_3$ reaction. Therefore, we intend to extract as much chemical and kinetic information as possible from the time behaviors of different compounds. As we are interested in the patterns of products, species with small signals can also help. Without normalization, using Euclidean distance would not make sense in this case. For instance, two species share identical time behavior,

but one has large signal, whereas the other has small signal. They would be different in Euclidean distance though they are the same in terms of time pattern. Under such circumstances, we must determine the angle between two variables to decide whether they are identical or not in terms of pattens, which makes the thing more complicated. In addition, normalization should not impact the clustering results obtained when using cosine and correlation distance as similarity metrics for clustering.

3) **The manuscript needs some reorganisation/rewriting for the methods part to make it more reader friendly.**
   - **Equations must appear when they are first broad up. It is extremely tedious to keep scrolling back and forth between the descriptive text and the equations at the end of the section (e.g., in section 2.3.2).**
   - **Equations cannot simply be dumped as a block like Eq. 1-3. They need introductory sentences identifying what these equations are about and linkage between them.**
   - **In other places, the reader has to wait for several sections before getting information that is only hinted about. E.g., it is mentioned multiple times that multiple runs were conducted (first in section 2.3.1). Why that was necessary, why sometimes 50 then 100, what was the difference between the runs, and how are these runs will be treated in the interpretation? Part of that information is presented much later, buried at the end of the "other parameter" section which at least for me was not an intuitive place to look for this information. The least that is needed here is a link/pointer to the section where this information will be provided.**
   - **Splitting section 3.3.1 and 3.3.3 feels forced. For me, the flow would make more sense to have the paragraph 875-889 with Fig 10 as part of 3.3.1. The rest would go to the end of 3.3.2.**

**Response:** Thanks for the comments, and we apologize for the inconvenience in reading.

We have reconstructed the manuscript to enhance the readability by placing descriptive text closer to, or directly followed by equations. Additionally, equation blocks have been split. Detailed revisions can be found in the revised manuscript.

Regarding the third point, explanations have been added in the same sentence, or right after that. The clustering algorithm was run multiple times to ensure a solid result, rather than a coincidence. And the repetition times have been fixed to 50 times through the whole manuscript. An average of results from these runs is taken for further analysis, accompanied by the standard deviation to indicate variations among the runs. To make these information clear to readers, following sentences have been refined/ added in the revised manuscript:

"To obtain a robust result, for each $c$ in this range, the FCM algorithm is performed 50 times with the default settings ($m = 2$, metric = Euclidean distance, $\varepsilon = 1\times10^{-5}$). The selected CVIs are calculated for each repetition, and the averages of results from 50 repetitions are used for further analysis." at the end of Sect. 2.3.1.

"Again, for each distance metric under scrutiny, the FCM algorithm was repeated 50 times to ensure reliable outcomes. The averages of results from these runs are then utilized for subsequent analysis." Has been added at the end of Sect. 2.3.2.

"For each setting, the algorithm is run 50 times for dependable results. By evaluating the variations of $m^*$ with $c$ and the initial values of membership degree …" at the end of Sect. 2.3.3.

For the last point, we have divided Sect. 3.3.3 into two parts and put them to Sect. 3.3.1 and 3.3.2, respectively, as the referee suggested, incorporating some minor revisions to improve the readability.

**Minor comments**

**These comments are present in the order they appear in the text not by relevance.**

1) **They authors claim that FCM classifies the ions by their kinetic properties (e.g., in the abstract). But it is the fitting with the GKP approach that provides this classification. They authors need to emphasize that it is the combination of FCM**

with GKP that reveals the chemical pathway information. As an example, if there are fast $1^{st}$ generation reactions and slow $1^{st}$ generation reactions, the slow $1^{st}$ generation reaction cluster could not be distinguished from a $2^{nd}$ generation cluster by its time trace alone. Only the fit with GKP approach and the m>1 value allows that distinction and the mechanistical interpretation of the cluster results. The authors need to rephrase the related part in the manuscript to emphasize the combined effect of FCM and GKP.

**Response:** Here must be a misunderstanding; the GKP does nothing to the classification. Various products are grouped into different clusters given their different kinetic properties (time behaviors). The GKP was fit to FCM results to extract kinetic information, i.e., generation number ($m_G$) and effective rate constant ($k$), from time series of each cluster center/ species. As far as two types of reactions exhibit distinct time patterns, FCM can differentiate them (in some circumstances, the number of clusters needs to be large enough sometimes). The function of GKP method is only to ascertain the specific values of $m_G$ and $k$. Of course, we can only name them to fast first-generation, slow first-generation, or second-generation cluster when their $m_G$ values are known. But it doesn't hurt if we don't know these values. They can be distinguished as early-generation I, early-generation II, later-generation I, ..., according to their relative ranking (in terms of formation/loss rate), and this will not change the conclusions we made.

2) **"m" is used in Equations to mean the fuzzifier (e.g., Eq. 1) and the oxidation generation (Eq 12). The authors should consider renaming one of these parameters to avoid confusion.**

**Response:** Accepted. The *m* referring to generation number in Eq. 12 has been replaced by $m_G$, as well as those in the text and plots.

3) **Line 72f and 123f: The cluster or factor identification usually uses some type of analysis of the correlation of the variables. These correlations are not necessarily caused by physical or chemical reasons. We interpret them as such.**

   **In the studied case, the clusters represent typical chemical processes because of your measurement setup where the main reason for correlation/similarity is the chemical pathway. For ambient measurements, this may not be true. There, the source and not the reactions in the atmosphere may be the predominant driver of correlation.**

   **Response:** Fully agree! Since chamber data was used for the case study, compound evolutions were closely associated with chemical reactions/ processes. But the referee is correct. For ambient measurements, different sources might be the main drivers of correlation.

   To make our description of FCM clustering universally relevant, the original sentence in Line 71-74 has been revised to "To reduce this complexity, dimension-reduction techniques are necessary, which compress the information in a dataset into a few to a dozen of factors/ clusters based on the underlying correlation/ similarity of different variables, in terms of their sources or physicochemical properties, and thus simplify the chemistry of investigated systems" and "each of which represents a typical chemical process with unique kinetic behavior" has been revised to "each of which represents a typical chemical process/ source with unique time behavior".

4) **line 88ff and 116f: Defining the "right" error matrix can indeed be a problem and cause for bias. But the error matrix intentionally added to the method to account for the uncertainty in measurements (e.g., reduce the impact of outliers or less reliable data points). Thus, it can also be seen as an "advantage" of PMF over NMF. Also note that (Buchholz et al., 2020) is showing the error matrix bias for a specifically challenging data set (FIGAERO thermal desorption data). It is yet to be seen if the same issues can occur for "easier" time series data sets.**

**Response:** It's true that outliers or less reliable data points (with low S/N) can be down-weighted by artificially increasing their errors/ uncertainties, and thus reducing their impact on FMP results. From this perspective, the error matrix could be treated as an "advantage" of PMF. However, the referee also acknowledged the difficulty in properly estimating the error matrix (in most cases analytical uncertainties are missing), which would affect PMF outcomes. Yes, except for the study by Buchholz et al. (2020), there are few literatures address this point.

To avoid contradictions/confusions, sentence in Line 88-91 has been revised to "Despite the similarities of mathematical formulation and constraints to PMF, the NMF algorithm does not need an error matrix as input This eliminates the potential impact of error estimation on outcomes and makes it more user-friendly."

And sentence in Line 115-118 has been revised to "Secondly, FCM is more user-friendly since only the data matrix is needed as input, whereas additional information is required for factor analysis methods, such as the error matrix needed in PMF. Furthermore, PMF assumes that the factor profiles remain constant over time and that the chemical species do not react with each other during the sampling period (Chen et al., 2011; Reff et al., 2007; Xie et al., 2022), which is not the case for chamber measurements".

references added:

Chen, L.-W. A., Watson, J. G., Chow, J. C., DuBois, D. W., and Herschberger, L.: PM2. 5 source apportionment: reconciling receptor models for US nonurban and urban long-term networks, Journal of the Air & Waste Management Association, 61, 1204-1217, 2011.

Reff, A., Eberly, S. I., and Bhave, P. V.: Receptor modeling of ambient particulate matter data using positive matrix factorization: review of existing methods, Journal of the Air & Waste Management Association, 57, 146-154, 2007.

Xie, M., Lu, X., Ding, F., Cui, W., Zhang, Y., and Feng, W.: Evaluating the influence of constant source profile presumption on PMF analysis of PM2. 5 by comparing long-and short-term hourly observation-based modeling, Environmental Pollution, 314, 120273, 2022.

5) **Line 76ff: I recommend splitting this paragraph into two: one for factorisation methods and one for clustering (starting at line 91).**

   **Response:** Accepted.

6) **Line 113ff: To put the "fuzziness" property of FCM into context, it would be helpful to specify which of the other mentioned methods allow variables/ objects to participate in multiple factors/clusters/components.**

   **Response:** The sentence in L109-111 has already mentioned that FCM, as well as PMF, allow variables to be associated with multiple clusters/ factors, whereas hard clustering algorithms assign each variables exclusively into one cluster.

   In reality, variables can always be assigned to multiple factors when using factorization methods, such as PMF, NMF, and PCA mentioned in the manuscript. But for clustering algorithms, to our knowledge, FCM is the only method that has such a function.

   Therefore, the sentence in Line 109-111 has been revised to "Firstly, FCM allows variables to be affiliated with multiple clusters, as PMF dose, similar to factorization methods like PMF, NMF, and PCA. Conversely, hard clustering methods, such as the most popular $k$-means clustering, assign each variable exclusively into one cluster."

7) **The introduction does not contain any information about the studied VOC/ SOA system. A brief paragraph about its relevance and the expected processes may be beneficial.**

   **Response:** Accepted. Following sentences have been added in 122-128 in the revised manuscript to give a brief introduction about the isoprene-$NO_3$ system:

"Isoprene is the most abundant BVOC on earth, and is highly reactive in the atmosphere, which is an important precursor of $O_3$ and SOA and thus imposes detrimental effects on climate and health (Carlton et al., 2009; Surratt et al., 2019). The reaction of isoprene with $NO_3$ is an important source of SOA, but its gas-phase reaction mechanism, especially the multi-generation chemistry and the contribution of the corresponding oxidation products to SOA formation remain ambiguous so far (Carlton et al., 2009; Fry et al., 2018; Ng et al., 2008; Rollins et al., 2009; Wu et al., 2021)."

references added:

Carlton, A. G., Wiedinmyer, C., and Kroll, J. H.: A review of Secondary Organic Aerosol (SOA) formation from isoprene, Atmos. Chem. Phys., 9, 4987-5005, 10.5194/acp-9-4987-2009, 2009.

Fry, J. L., Brown, S. S., Middlebrook, A. M., Edwards, P. M., Campuzano-Jost, P., Day, D. A., Jimenez, J. L., Allen, H. M., Ryerson, T. B., Pollack, I., Graus, M., Warneke, C., de Gouw, J. A., Brock, C. A., Gilman, J., Lerner, B. M., Dubé, W. P., Liao, J., and Welti, A.: Secondary organic aerosol (SOA) yields from NO3 radical + isoprene based on nighttime aircraft power plant plume transects, Atmos. Chem. Phys., 18, 11663-11682, 10.5194/acp-18-11663-2018, 2018.

Rollins, A. W., Kiendler-Scharr, A., Fry, J., Brauers, T., Brown, S. S., Dorn, H.-P., Dubé, W. P., Fuchs, H., Mensah, A., and Mentel, T.: Isoprene oxidation by nitrate radical: alkyl nitrate and secondary organic aerosol yields, Atmos. Chem. Phys., 9, 6685–6703, 10.5194/acp-9-6685-2009, 2009.

Surratt, J. D., Lin, Y.-H., Arashiro, M., Vizuete, W. G., Zhang, Z., Gold, A., Jaspers, I., and Fry, R. C.: Understanding the early biological effects of isoprene-derived particulate matter enhanced by anthropogenic pollutants, Research Reports: Health Effects Institute, 2019, 2019.

Wu, R., Vereecken, L., Tsiligiannis, E., Kang, S., Albrecht, S. R., Hantschke, L., Zhao, D., Novelli, A., Fuchs, H., Tillmann, R., Hohaus, T., Carlsson, P. T. M., Shenolikar, J., Bernard, F., Crowley, J. N., Fry, J. L., Brownwood, B., Thornton, J. A., Brown, S. S., Kiendler-Scharr, A., Wahner, A., Hallquist, M., and Mentel, T. F.:

Molecular composition and volatility of multi-generation products formed from isoprene oxidation by nitrate radical, Atmospheric Chemistry and Physics, 21, 10799-10824, 10.5194/acp-21-10799-2021, 2021.

**8) Line 336ff: The selected scaling method will shift the emphasis of the signals (maybe even as much as the error matrix can for PMF?). I guess that scaling would enhance the importance of signals with lower intensities. Is there any work how different scaling methods impact cluster identification?**

**Response:** Yes, normalizing/ scaling the data raises the importance of low-intensity signals. But as already mentioned in the response to the second major comment, one of the purposes of this study is to better understand the multigeneration chemistry of isoprene-$NO_3$ reaction, i.e. we are after time patterns of different products. Therefore, we intend to extract as many different time patterns/ processes of oxidation products as possible from measurements, rather than only obtaining the important reaction patterns/ processes. If using non-normalized data for clustering, certain small groups of low-intensity species, like 3N-dimers in our case, which have significantly different time series from others, would be neglected when the number of clusters is not large enough.

It is well recognized that the data preprocessing methods could affect clustering results. However, few studies address the details about impacts of different scaling methods on cluster identification. It is also out of the scope of this study. But we have tried to perform FCM to non-normalized data. There would be many one-component clusters (variables with membership degree larger than 0.5) and the cluster representative for 3N-dimers cannot be distinguished even when the number of clusters is set to 15.

**9) Section 165-181: This section was more confusing then helpful to me. These are the questions I was left with. Most of them are not really relevant for the study as no UMR data was used (but that is not clear at the part of the manuscript yet).**

- **The "500 peaks" (line 165) does not refer to the HR analysis result? I guess the authors mean UMR "peaks" in the first sentence? Why is that relevant if the study is about HR data? Rather than the "500 peaks" information, the authors should provide how many HR ions were identified (and how many were considered "product" ions).**
- **Why where the UMR (?) peaks screened if they contain product ions? Shouldn't the FCM group the precursor/educt ions into separate cluster? Running FCM with all ions would then result in a higher number of clusters but the educt clusters should be distinct from the product ones. It would be yet another advantage of FCM if FCM indeed easily separates the educt ions as it would eliminate a step in the data cleaning process.**
- **what is meant with pronounced changes in line 172? Increase/ decrease of signal intensity? How much would the change have to be to be pronounced?**
- **Starting the sentence in line 175 with "Therefore" implies that the HR fitting was only applied because there was some uncertainty in separating the product ions when using the UMR data. What would have been the benefit to run FCM with UMR data?**

**This section should be rewritten, omitting the UMR parts and only focusing on introducing and describing the HR data analysis.**

**Response:** We apologize for the unclear expression. Yes, "500 peaks" in the beginning of the sentence refers to the UMR peaks. These sentences were added to provide a general description of the Br- CIMS data.

It's a wrong expression here. The product ions were selected out not screened out.

In principle, yes, the precursor related ions would be grouped into a separate cluster since their time patterns are different from products. However, the precursor (isoprene here) cannot be detected by Br- CIMS, so the corresponding time series was not included in the data set.

The sentence in Line 172 describes how we identified product ions. The "pronounced changes" in this sentence just mean visible changes in the ion signals, which could be either an increase or decrease, when $NO_2$, $O_3$, and/ or isoprene were added.

There is no quantitative measure for "pronounced" change. The signal changes before and after the oxidant and/ or VOC precursor addition were only visually compared.

We will delete "therefore" in the sentence in Line 175. When applying FCM to UMR data, we can get the kinetic information of different clusters, but no chemical properties. This helps us to quickly learn the chemical processes of the investigated system without putting in a lot of effort in the data analysis (to obtain HR results).

According to comments from the referee, UMR data related description in this section has been deleted. The revised paragraphs are shown below:

"The raw mass spectrometric data were processed using the Tofware toolkit ... following the routines described by Stark et al. (2015). High-resolution peak fitting was conducted in the mass range of *m/z* 60 – 600 to identify the chemical composition of detected ions. For high-resolution peak assignment, we fitted the observed peaks using predefined instrument functions (including peak shape, peak width as a function of *m/z*, and baseline). If necessary, contributions of more than one component were considered for the fit, in order to reduce the residuals of the fitting. Once the peak numbers and peak positions were fixed, the chemical formula (consisting of C, H, O, and N atoms) of each peak was assigned manually by selecting from a formula list generated by the software. During the peak fitting, isotopes were constrained, and only plausible formulas with relative *m/z* deviations smaller than 10 ppm were considered. In addition, only molecule formulas with a time behavior commensurable with expectations for the specific chemical system were assigned (Pullinen et al., 2020). For example, it is illogical if large amounts of organonitrates are observed under low $NO_x$ conditions.

Overall, around 160 ions were identified by the Br⁻-CIMS. The background signal of each ion was determined from measurements prior to precursor injection and was subtracted from the signal measured in the chamber. These ions consist of species related to real isoprene oxidation products, as well as other signals related to ion source, internal standard, and interferences from chamber and tubing. The product ions are those produced by isoprene oxidation, and they should have visible changes (either increase or decrease) when the chemistry is initiated or modified. A simple way to select out the product ions from other chemically irrelevant signals is to examine the time evolution of each ion. By comparing the signals before and after each injection we can easily

distinguish the product ions from others. Among all the identified ions, a total of 91 ions were recognized as product signals. Since we intend to investigate the underlying chemical relationships of different products through their time behavior, not the absolute concentration, normalized signals were finally used for further analysis. Calibration procedures are described in more detail elsewhere (Wu et al., 2021)."

10) **Line 184f: What is "reasonable chemical meaning" in this context?**

**Response:** Here "reasonable chemical meaning" means that the formula assigned to a peak should be reasonable in terms of the DBE and elemental composition, as well as be potentially possible to exist in the system. For example, there is a large peak at $m/z$ 183 and the software suggests it should be "C8H8*Br⁻". This is not a "chemically reasonable" compound because the DBE of this compound (= 4) increases compared to isoprene (DBE = 2) and it does not consist of any oxygen atoms, which is unlikely to happen during an oxidation process.

To avoid confusion, the original sentence has been rephrased to "During the peak fitting, isotopes were constrained, and only plausible formulas with relative m/z deviations smaller than 10 ppm  were considered.".

11) **Line 189f: Is this normalisation now the same thing that is called "scaling" in lines 336-344?**

**Response:** No, the normalization here is different from the "scaling" in Lines 336-344. Here, the data was normalized to the sum of total ion counts to minimize the influence of instrument performance drift on data. A more detailed description of this could be found in our previous study (Wu et al., 2021). The scaling in Lines 336-344 means applying the Euclidian norm to the data (which are normalized to total ions) to scale the signal intensity of each product.

To make this clear, the original sentence has been revised to "…, normalized (to the sum of total ion counts) signals were finally used for further analysis.".

12) **Line 192ff: The Jenkin reference is for the Master Chemical Mechanism (MCM). Is there a reference for the actual box model? Is it the Carlsson et al 2022 reference at the end of the paragraph? If the model was "build" by the authors, should the code be made available?**

**Response:** Yes, the Jenkin reference is for the updated isoprene mechanism in MCM. The model was resolved by the EASY complier, a program designed to create an interface for the modeling of zero- and one-dimensional chemical reactions and transport system. It is a program used in our institute (IEK-8, Research Center Juelich) based on IDL and FACSIMILE. Carlsson et al. (2022) also did their simulation using EASY.

13) **Line 264: It is not clear what "with improvements" refers to in this sentence. Do the CVIs still need to be improved? Or they have been improved? But what is that improvement?**

**Response:** In this sentence, "with improvements" corresponds to the disadvantage of the first type of CVIs mentioned in last sentence. The geometry structure of data is not considered in the first type of CVIs but considered in the second type of CVIs. Therefore, the second type of CVIs were "improved" (compared to the first type).

To make this clearer, the original sentences have been rephrased to "Considering this, another type of CVIs, such as Fukuyama-Sugeno index (Fukuyama, 1989), Xie-Beni index (Xie and Beni, 1991), Kwon index (Kwon, 1998) and Bouguessa-Wang-Sun index (Bouguessa et al., 2006), were proposed, which takes both membership degree and the geometry structure of data set into consideration.".

14) **Line 369ff: This section is very difficult to follow. This is one of the examples where improvement is needed as pointed out in Major Comment 3.**
**The sentence implies that Jm is defined in Eq. 8 and Hm is given in Eq. 9. While Jm is surely the Variable defined in Eq. 1, Hm seem to not have a definition**

**equation. Further, the text speaks of G and C while the equations are defining μG and μC. The equations are given as a block instead of interlacing them with the relevant text. What are the alpha and beta constants in Eq 8 & 9?**

**Response:** As denoted in the descriptive text below the equations, $\alpha$ and $\beta$ are two constants defined in the function of $\mu_G(m)$ and $\mu_C(m)$, which are set to the value of 1.5 and 10 in the calculation (Gao et al. 2000).

To make it clear and for better understanding, this section has been rephrased to "In this study, we adopted the method proposed by Gao et al. (2000) to determine the optimal fuzzifier value $m^*$. Based on their method, a fuzzy objective function ($\mu_G$) and a fuzzy constraint function ($\mu_C$) have been defined, and the intersection of $\mu_G$ and $\mu_C$ is supposed to be the value of $m^*$, as defined by Eq. 8:

$$m^* = \arg_{\forall m}\left\{max\{min\{\mu_G(m), \mu_C(m)\}\}\right\} \tag{8}$$

where $\mu_G$ is a fuzzy objective function, as calculated by Eq. 9:

$$\mu_G(m) = exp\left\{-\alpha \times \frac{J_m(U,V)}{\max_{\forall m}(J_m(U,V))}\right\} \tag{9}$$

where $\alpha$ is a constant larger than 1, and generally set to be 1.5 in practice, and $J_m(U,V)$ is the objective function of fuzzy clustering as shown in Eq. 1.

And $\mu_C$ is a fuzzy constraint function as defined by

$$\mu_C(m) = \left\{1 + \beta \times \left(\frac{H_m(U,c)}{\max_{\forall m}(H_m(U,c))}\right)\right\}^{-1} \tag{10}$$

where $\beta$ is a constant that is usually set to be 10 in practice, and $H_m(U,c)$ is the fuzzy partition entropy calculated by

$$H_m(U,c) = -\frac{1}{n}\sum_{i=1}^{c}\sum_{j=1}^{n} u_{ij} \cdot log_a(u_{ij}) \tag{11}$$

where $u_{ij}$ is the membership degree of object $j$ to the $i^{th}$ cluster, and $a$ is a constant $\in (1, \infty)$ which is usually set to the mathematical constant."

15) **Line 386: How does increasing the maximum number of iterations solve the issue of converging to early on a local minimum? The algorithm stops when the convergence criterium is reached. If that happens for a local minimum, increasing the maximum number of iterations should not have an effect because the criterium is already fulfilled at a low number of iterations. Increasing the number of iterations should only change the outcome if no convergence is reached within the number of allowed iterations.**

    **Response:** Agree. If the algorithm already converges, increasing the maximum number of iterations would not change the outcome. The paragraph has been rewritten to "We find that when using a small number of iterations FCM does not always return the same result for each run, and sometimes not even a valid solution.  This is probably because the limit of iterations is reached before the algorithm converges. To avoid this , the maximum number of iterations was set to be 10000 in this study. In our case, however, hundreds of iterations can already ensure a valid solution and reproducible results for our data."

16) **Line 397ff: How frequent are invalid solutions? For one value of c (and m), how many of the 50 runs were disqualified. Also, was there a trend in number of invalid solutions? E.g., more valid solutions close to the "optimal" cluster number?**

    **Response:** For $c$ in the range from 2 to 10 and $m$ in the range from 1.1 to 2.3, all of the 50 repetitions return valid solution. There is a trend in the number of valid solutions. It is observed that with increasing $c$, the range of $m$ that returns valid solution became narrower. For example, when $c = 2$, for $m$ in the whole tested range (from 1.1 to 9.0), FCM always returns valid solutions. But when $c = 10$, if $m$ is set with a value larger than 2.3, FCM always returns invalid solutions.

17) **Section 2.3.4: The title "Other parameter" is misleading for this section. This section does not deal with other parameters used in FCM, but rather with the constraints and methods to improve the quality/reliability of the solutions.**

**Response:** Not really. The maximum number of iterations, the initial fuzzy partition matrix ($U_0$), and the stop criterion ($\varepsilon$) are all parameters of FCM that should be defined by the user. However, the definition of a valid cluster/ solution is made by us, and it is not a parameter of FCM.

In the revised manuscript, the first and the last paragraphs, which are both related to the FCM parameters, are put together, and the paragraph that describes the definition of a valid cluster/ solution has been moved to the end of this section. In addition, the title of this section has been changed to "Other parameters and constraints".

18) **Line 447: The authors went to the trouble of running ensembles of 50 or 100 runs. But now only one "representative" run is shown. Why not provide the average with standard deviation or interquartile range? Or using a heatmap of the distributions?**

**Response:** Accepted. The repetition times have been fixed to 50 throughout the manuscript, and an average of results from these runs with standard deviation is used for further analysis.

19) **Line 481: $V_{BWS}$ was shown to work well for largely overlapping clusters: Is the investigated data set such a case where large overlap is expected?**

**Response:** Yes, we would say that there is an overlap in our data, but the overlap is not very large. As illustrated in Fig. 12, for the selected major products, most of them are dominated by one cluster, indicating that they were formed mainly from one pathway. However, some species were primarily assigned to two or more clusters, such as $C_5H_7NO_5$, $C_5H_9NO_5$, $C_5H_9NO_7$, $C_5H_9N_2O_9$ and $C_{10}H_{17}N_3O_{12}$, to which the

second cluster contributed 20-30%, indicating that they were probably comprised of two or more isomers, or they were produced from two or more different pathways.

**20)** **Figure 1: I found it challenging to keep all the mentioned numbers for optimal cluster number for each CVI in my head while looking at Fig 1. It would be helpful to mark the optimal cluster number for each CVI in each panel (red circle for selected value, blue square for optimal number based on this CVI).**

**Response:** Accepted. Figure 1 has been changed based on the referee's comment. The updated plot is shown in below:

[Figure]

**Figure 1.** Values of selected clustering validity indices $V_{SWCV}$ (a), $V_{FS}$ (b), $V_{XB}$ (c), $V_{Kwon}$ (d), $V_{BWS}$ (e), and $FS$ (f) as a function of the number of clusters from 2 to 10. The averages of results from 50 repetitions are shown in the plot, and the error bars show the standard deviations. Blue points denote the optimal values of c according to each CVI, and the solution selected for further analysis is marked by red circles.

21) **line 516f: why does this suggest that the cosine metric is more suited? Is there an assumption that the shape of V$_{BWS}$ should be smooth as a function of c?**

**Response:** This is based on the definition of $V_{BWS}$. As mentioned in Lines 479-483 and in SI (5), $V_{BWS}$ uses the ratio of separation and compactness to evaluate the quality of a clustering outcome. Therefore, the greater the isolation of the cluster centers from each other, and the lower the dispersion of the objects within each cluster, a larger value of $V_{BWS}$ would be, which indicates a better partition. When choosing the cosine distance as the similarity metric for FCM, the calculated $V_{BWS}$ is always larger than those using other similarity metrics, suggesting that the cosine distance is better than other metrics in terms of $V_{BWS}$.

There is no assumption that the shape of $V_{BWS}$ should be smooth as a function of $c$. In theory, there're no constraints on the shape of $V_{BWS}$. The shape of $V_{BWS}$ depends inherently on the structure of data.

22) **Line 514: is it really true that the impact on clustering output is neglectable? Or isn't it rather that this metric is not sensitive to the differences? These metrics only tell us how well the solution was separated. But it does not tell us about the shape of the clusters. I.e., with the same degree of separation, the actual clusters might look different.**

**Response:** Agreed. It is more appropriate to say that $V_{FS}$ is insensitive to the differences in partition caused by different similarity metrics. As the referee also pointed out in the minor comments, the scale of the y axis in Fig. 2a is not proper. After reducing the y scale in Fig. 2a, we can see that the $V_{FS}$ value suggests that the cosine distance is slightly better than others, though differences caused by different distance metrics are small.

Therefore, the original sentence has been rewritten to "In terms of $V_{FS}$,  it

indicates that the cosine distance is more suitable for FCM in our case, although the differences caused by different distance metrics are minimal (Fig. 2a)".

[Figure]

**Figure 2.** Values of selected clustering validity indices $V_{FS}$ (a), $V_{Kwon}$ (b), $V_{BWS}$ (c), and $FS$ (d) as a function of the number of clusters. Points in different colors are results obtained with different distance or similarity metrics. The averages of results from 50 repetitions are shown in the plot, and the error bars denote the standard deviations. Euclidean distance was used in the calculation of various CVIs.

**23) Why are the curves in Fig 1 not matching any of the lines in Fig 2?**

**Response:** The curves in Fig. 1 actually match lines for the Euclidean case in Fig. 2. Because the sizes of panels in Fig. 1 and Fig. 2 are different, the shape of the curves in the two plots look different.

24) **Section 3.1.3 My take on what was described in this section: With small cluster number, we have too few clusters. Hence, we need to allow more overlap for variables. When we allow more clusters, we can be stricter with assigning the objects (m\* gets smaller). As the solutions may be more "specific" we are now more sensitive to "local" minima. As those are driven by the starting point, we get more sensitive to the U0.**

**Response:** This explanation makes sense! We have added this in L679-686 in the revised manuscript, to provide a plausible explanation for the relationship between $m^*$ and $c/U^0$. The revised paragraph is shown below:

"As shown in Fig. 3b, we do observe a relationship between $m^*$ and $c/U^0$…  One plausible explanation for the dependency of $m^*$ on $c/U^0$ is shown as follows. When $c$ is small, there are more overlaps between clusters and thus $m^*$ can be relatively large. When $c$ becomes larger, the assignment becomes "stricter" and the overlaps between clusters are reduced. Therefore, $m^*$ gets smaller, and the clustering outcomes become more specific, which are likely to be more sensitive to local minima. Since the local minima largely depends on $U^0$, consequently, the results become more sensitive to $U^0$."

25) **Section 3.1.3 How much does a difference of 0.1 in m really change the clustering results? I.e., how sensitive is the actual result to a slightly different m value? ~1.5 seems much closer to 1 (= non fuzzy) than to 9 (upper limit in this study).**

**Response:** We have checked the clustering outcomes with $m$ = 1.4, 1.5, 1.6, 1.7, 1.8, 1.9, and 2.0. The patterns of cluster centers are almost identical for different $m$. However, with different $m$, some of the membership degrees of objects to cluster centers changed to some degree, but less than 50% at most.

According to previous studies, the value of $m^*$ usually falls in the range between 1 and 5, and $m^*$= 2 was used by default in most cases (Hathaway and Bezdek, 2001; Huang et al., 2012; Ozkan and Turksen, 2007; Pal and Bezdek, 1995; Wu, 2012). In

this study, we set the examining range of $m^*$ between 1.1 and 9.0, and the clustering results show that the values of $m^*$ under different conditions (with different $c$) vary in a narrow range around 1.5, indicating that the FCM outcomes in our case are relatively crisp.

We have added a sentence "The values of $m^*$ determined for our data set vary around 1.5 despite different $c$ and $U^0$, indicating that the FCM results in this study are relatively crisp." In L699-701 in the revised manuscript to clarify this.

26) **Line 565ff: Do the authors have any idea why the 5 cluster case was super stable, but the 3, 4, and higher ones varied more? Could this be an indicator for the "perfect number of clusters"? Or just coincidence?**

**Response:** To be honest, it is not clear for us why the FCM results with smaller cluster numbers are quite robust, but more variations can be observed when the number of clusters becomes larger. We have expanded the repetitions to 200 times, and similar phenomena have been observed. Therefore, this is not a coincidence, but rather an unknown cause or rule behind it. Perhaps this is an indicator of the optimal number of clusters.

27) **Line 592f: "Part of the species from the former cluster 1 is separated out as a new cluster 2, dominated by molecule(s) from a very narrow mass range, where mass profile 1 also has its Maximum" – I do not understand this sentence. However that sentence is meant, while the mass spectra of C1 and C2 may look similar, the time series are not. C2 is not directly linked to the injections (peak is later). Or are the authors implying that the C1 of the 3 cluster solution had some reaction products grouped in which are now taken out?**

**Response:** We apologize for the confusing explanation. In this sentence, we intended to express that some of the products in C1 of the 3-cluster solution are partially assigned to C2 of the 4-cluster solution.

To make it clearer, the original sentences have been rephrased to " As shown in Fig. 4b, part of the species from cluster 1 in the three-cluster solution is separated out to a new cluster (cluster 2 in four-cluster solution) when increasing the number of clusters from 3 to 4.  The newly formed cluster shares the same fingerprint molecules, i.e., $C_5H_9NO_5$ and $C_5H_9NO_6$ (corresponding to species no. 34 and no. 38 in Fig 4b), in the mass profile with cluster 1 in three-cluster case."

28) **Line 635: Could it also be that compounds which would retain the nC and get more functionalised are too low volatile to remain in the gas phase? Where particles formed in these experiments? If not, could low volatility compounds be lost to the chamber walls?**

**Response:** Yes, it is possible that there are later-generation products with larger $n_C$. However, as they become highly functionalized through multiple oxidation steps, they would have very/extremely low volatility and thus only exist in the particle phase. Therefore, we cannot detect such compounds in the gas phase.

The chamber data used in this study is from an experiment without seed addition. No SOA was formed according to SMPS and AMS measurements. Except for the fraction remaining in the gas phase, the low-volatility products produced from oxidation could condense onto the chamber wall.

To make it more comprehensive, we have revised the original sentence to "In general, the early-generation …, suggesting that the later-generation products detected in the gas phase in this study are formed through further oxidation of early-generation species and undergo more fragmentation during oxidation. Of course, it is very likely that there are later-generation products with larger $n_C$. However, as they become highly functionalized through multiple oxidation steps, they would have a very or extremely low volatility and thus only exist in the particle phase, undetectable in the gas phase."

29) **Fig 6: Cluster 1 has the most "outliers" while the other clusters seem to have the grey lines closer together. Do the authors have any explanation for this higher "spread" of members in Cluster 1?**

**Response:** According to their time patterns, Cluster 1 of model data is supposed to represent first-generation products, whereas other clusters stand for later-generation products which have significantly different time behaviors from Cluster 1. Because there are various first-generation products in MCM, their formation/ loss rates vary, and thus their time patterns also differ, leading to a more dispersed cluster. Nevertheless, they were mainly allocated to Cluster 1 since their time behaviors are more similar to that of Cluster 1 than to other later-generation clusters.

The most distinct outlier members in Cluster 1 (in two-cluster case) are $CH_3O_2NO_2$, $ME_3BU_3ECHO$, and $ISOPCNO_3$, among which $CH_3O_2NO_2$ has much larger formation and loss rates, making its time series significantly different from the others. With the number of clusters increasing (to 6), Cluster 1 is further split into two sub-clusters according to their different formation/ loss rates. But $CH_3O_2NO_2$ cannot be separated out even when increasing the cluster number to 10. Instead, it tends to be evenly distributed to each cluster (with slightly higher membership degrees to two first-generation type clusters), which indicates that this compound somehow becomes an outlier of all clusters. This is actually a limitation of FCM clustering. It has trouble clustering outliers or objects with a very small group size since it assumes a similar member size for all clusters. They would be ignored instead of getting their own cluster if the cluster number is not large enough.

30) **Line 713f & 717f: These sentences seem contradictive. The first sentence says that C1 and C2 differ in their formation& production rate. Second sentence says that formation rate of C2 resembles that of C1. Are they different or similar?**

**Response:** We are sorry for causing the confusion, but in the first sentence, cluster 1 and cluster 2 are both from chamber data, while in the second sentence, it is pointed out that the formation rate of cluster 2 resembles that of cluster 1 of the model data.

To emphasize this point, the second sentence in Lines 717-718 has been revised to "Note that the formation rate of cluster 2 (from FCM analysis of the chamber data) resembles that of cluster 1 (in the five-cluster solution) from FCM analysis of the model data.".

31) **Figure 5 & S5: The meaning of the grey circles is not clearly explained when the Figure is first mentioned. What are the "individual species"? What does their marker size refer to? Is it the average mass spectrum over the full experiment? Who do the size of the grey markers relate to the size of the cluster markers?**

**Response:** The "individual species" here refers to closed-shell oxidation products detected by $Br^-$ CIMS. The marker size (area) of species is proportional to their average signal intensity over the whole experiment. The marker size of clusters is proportional to the sum of average signal intensity of all species in the cluster weighted by their membership degrees.

To make it clearer, the captions of Fig. 5 and Fig. S5 have been rephrased, as shown below. And to distinguish from cluster centers, species are depicted as grey hexagons instead of circles. Other related plots in the manuscript have also been updated accordingly.

[Figure]

**Figure 5.** Average carbon oxidation state ($\overline{OS_C}$) of the obtained FCM clusters from chamber data as a function of number of carbon atoms ($n_C$). Panel (a) to panel (d) show results for solutions with 2 to 5 clusters, respectively. Cluster centers are depicted by circles in different colors. The color scheme follows that in Fig. 4. The marker area of clusters is proportional to the sum of average signal intensity of all species in the cluster weighted by their membership degrees. Closed-shell products detected by Br⁻ CIMS are shown as grey circles, and the marker area is proportional to the average intensity of species over the whole experiment.

[Figure]

**Figure S5.** Average oxidation state $(\overline{OS_C})$ of FCM clusters of chamber data as a function of number of carbon atoms $(n_C)$. Panel (a) to panel (e) show results for solutions with 6 to 10 clusters, respectively. Cluster centers are depicted by circles in different colors. The color scheme follows that in Fig. 4. The marker area of clusters is proportional to the sum of average signal intensity of all species in the cluster weighted by their membership degrees. Closed-shell products detected by Br- CIMS are shown as grey circles, and the marker area is proportional to the average intensity of species over the whole experiment.

32) **Line 750: The fact that the cluster markers cover a smaller space than the original data: 1) that is a logical consequence when doing clustering. The result will not be on the edge of the distribution. 2) this graph is not comparing the**

range of OSc vs nC for the clusters. Only the average of the cluster is given. But to check the range, one would have to see the position of all species contributing to each factor and evaluate those.

**Response:** Yes, the cluster centers represent the (weighted) centroids of objects belonging to the cluster, and thus always cover a narrower space than the original data. In this paragraph, we aim to characterize the chemical properties of different clusters by analyzing each cluster center. The characteristics of members in each cluster were analyzed in Sect. 3.3.3 in the original manuscript. Now this part has been put directly after this.

33) **Line 766: assuming that the clusters are sorted by their oxidative "age", the statement is only partially true. 1-2-3 follow the trend. 4 is on the "line" but has higher nC than 3. Cluster 5 is off the line.**

**Response:** Different clusters are separated out by their distinct kinetic properties (oxidation steps/generation number and effective rate constant in this study). However, in reality reactions taking place in an oxidation system are far more complicated than expected. For instance, molecules can also undergo autooxidation, fragmentation, as well as dimerization during oxidation processes, which could significantly change the chemical properties of products. This could probably explain why cluster 4 and cluster 5 did not strictly follow the oxidation trajectory in chemical space. But the conclusion that the early-generation clusters (taking cluster 1 and cluster 2 as a whole) have a lower oxidation state but larger $n_C$ while the later-generation clusters (taking cluster 3, cluster 4 and cluster 5 as a whole) have a higher oxidation but smaller $n_C$, is true in general.

34) **line 777: It is not the measurements that are fitted to the function, but the function is fitted to the data points. But which parameters in Eq 12 are "free fit"? k and m? Also a?**

**Response:** Accepted. The original sentence "By fitting the measurements to the GKP function (Eq. 12) we can …" in Line 777 has been revised to "By fitting the GKP function (Eq. 13) to the measurements, we can …."

$k$, $m$ (replaced by $m_G$ in the revised version), and $a$ are all free-fitted parameters in Eq. 13. But because $a$ does not have a specific kinetic meaning, its value was not listed in the manuscript.

35) **How are products of auto-oxidation classified in the GKP approach? The intermediate stage of one oxidation step with NO3 (= first generation) can lead to a range of products with varying number of autooxidation steps. Are such products still classified as "first generation"? How will varying degree of autooxidation affect the "m" parameter?**

**Response:** As mentioned in Sect. 2.4, the GKP model describes a multi-step reaction system as a linear system, and only the first-order reactions of precursor and products with the oxidant ($NO_3$ in this study) are considered. Therefore, the generation number ($m_G$) in the model only reflects the reaction steps (with $NO_3$) needed to form a certain product. It has nothing to do with autooxidation. If the formation of a product involves several autooxidation steps after adding $NO_3$ to one of the double bonds of isoprene, it should still be recognized as first-generation product by the GKP model. Autooxidation steps should not change the value of $m_G$. However, autooxidation will affect the chemical properties of products.

36) **Line 784f: "chemically realistic time patters" what is meant by that? That the time series looks like it could be from a combination of realist reactions? What is meant by chemical properties here?**

**Response:** The time series of clusters dominated by early-generation products show that they were formed immediately after the reaction triggered and their formation rates were relatively large. In contrast, the signals of later-generation clusters were very low and increased slowly at the beginning of the reaction. These are well

consistent with our understanding of early- and later-generation products, and make sense in chemistry. That's why we call them "chemically realistic". "chemical properties" is a typo in this sentence. It should be "kinetic properties" instead, which mainly refers to the generation number and effective rate constant mentioned in previous sentences.

37) **Line 784f: What is the issue with Cluster 5? That it is so low intensity? Or that it is so noisy? Or is it the "unusual" shape? Are the authors not trust this "unchemical" shape?**

**Response:** Cluster 5 is dominated by several 3N- and 4N-dimers, such as $C_{10}H_{17}N_3O_{15}$, $C_{10}H_{17}N_3O_{16}$, and $C_{10}H_{17}N_3O_{18}$, which have very low signals. Such compounds are large and highly oxidized, resulting in low or extremely low volatility, and therefore predominantly exist in the particle phase. So their gas-phase signals are low and noisy. However, these compounds are real as there are several species detected by $Br^-$ CIMS and they share similar time behaviors. More information about the characteristics of these compounds can be found in our previous study (Wu et al., 2021).

38) **isomer theory. If one ion represents isomers from different steps, shouldn't those isomers than be assigned to different clusters? I.e., the ion should show up in a first and second gen cluster. The example C5H9NO5 shows exactly that. It has a considerable contribution to C1 and to C2 (but there it falls under the 0.5 mark).**

**Response:** Definitely. If a species consists of isomers formed from different pathways, it is expected to separate them into different clusters, such as $C_5H_9NO_5$, which is mainly attributed to cluster 1, but a certain fraction (~ 25%) to cluster 2. However, it's important to note that only isomers from kinetically different pathways can be differentiated by FCM in this study. In other words, if two isomers are both formed through one oxidation step despite different reaction channels, they cannot be separated by FCM in theory. This is the limitation of this method.

**39) Fig 11: The position of the red star feels odd. The position seems to be at a "too low" m value. The 5 individual highest signals all have much higher m values. For the other clusters, the star falls more in the middle of the point distribution. If the position of the red star is correct, it should mean that the m and k values of the read cluster are not represented well by the high affiliation species but rather dominated by species with lower affiliation.**

**Response:** We have double checked the fitting program and the returned results. No artificial errors have been identified. In this work, we aim to cluster different time patterns of various products, regardless of their absolute signal intensities. Therefore, variables were normalized by their Euclidean norms before clustering, in order to eliminate the impact of different signal intensities of species.

In fact, the value of $m_G$ depends on the shape, or more specifically, the curvatures of the time series curves (Koss et al., 2020). As shown in the plot below, the time pattern of the cluster center of C1 is more similar to that of 2N-dimers (panel a). As a consequence, the derived generation number of the cluster center aligns closely with the $m_G$ values of 2N-dimers, as shown in panel b.

[Figure]

**40) The existence of C5 N1 monomers (e.g., C5H7NO6) as second-generation products confirms the importance of H-abstraction by NO3 radicals as oxidation mechanism. Or are the authors suggesting a different mechanism that does not**

**result in addition of the NO3 radical to the molecule? Anyhow, the reaction scheme in Scheme S1 does not consider any such products, only dual NO3 addition products are listed.**

**Response:** Besides H abstraction by $NO_3$, second-generation 1N-monomers can be formed through $NO_3$ addition. For example, a $NO_3$ radical attacks the remaining double bond in $C_5H_7NO_4$, forming a $RO_2$ radical with chemical formula of $C_5H_7N_2O_9$. This $RO_2$ radical can undergo intramolecular H-shift, and if H-shift occurs at the carbon with a $-ONO_2$ group attached, it leads to the formation of a carbonyl compound with $NO_2$ loss (with formula of $C_5H_7NO_7$). Without additional information, it is challenging to tell which mechanism is closer to the truth. Further, providing detailed formation mechanisms of products is out of the scope of this work.

Scheme S1 is a general framework developed to trace the reaction mechanism of isoprene and $NO_3$, and to predict the most likely products formed during this process, based on our knowledge about isoprene, peroxy and alkoxy radical chemistry. It is not guaranteed to be comprehensive and/ or correct in real situations. As shown in Scheme S1, C5 1N-monomers are generally supposed to be first-generation products from isoprene-$NO_3$ reaction. But the FCM results indicate that some of the C5 1N-monomers detected in our system, such as $C_5H_7NO_6$, $C_5H_7NO_7$, and $C_5H_9NO_7$, were primarily originated through later-generation reactions. This is indeed one of the big advantages of FCM analysis, which gives insights into the possible formation processes of compounds.

41) **Line 986 & Scheme S2: In my book, there are two pathways to form C4H7NO5: one via 1,4-H shift and one via +RO2 reaction. The split ratio will depend on the RO2 conc. But for the formation kinetics, the rate limiting step is relevant. Since that is most likely the first NO3 addition, no differentiation between the two paths is possible (i.e., it is impossible to determine the split ratio). This shows the limitation of this approach. If a product is formed through different pathways, but the rate limiting step is common, FCM coupled with GKP will only provide**

**the sum over these paths. This comment also relates to the claim in Line 970 that those compounds are formed by a single pathway.**

**Response:** Fully agree. As already mentioned in the response to comment no. 38, FCM analysis can only differentiate kinetically different pathways. For $C_4H_7NO_5$ in this case, due to different subsequent $RO_2$ reactions, it can be formed from two different channels. However, there are no oxidation step(s) in subsequent reactions, and reaction with $NO_3$ (forming $RO_2$) seems to be the rate limiting step for both channels. From this perspective, these two channels are kinetically identical, and FCM cannot separate them. That's why we didn't discuss the details of its formation mechanism in the manuscript, but only stated that $C_4H_7NO_5$ was probably formed from further oxidation of $C_5H_8O_2$ by $NO_3$. But as both intramolecular isomerization and bimolecular reaction are possible, it is difficult to filter out either of them without additional information. Therefore, we include both of them in the proposed scheme.

42) **SI section (1) What is meant by the "knee"? How is it determined? Is it the tuning point of the curve? In Fig 1 a) c=5 is chosen? But why? What is the math behind that? F it is "by eye" I could also take c=6 as "knee" or elbow or other bent body part.**

**Response:** "knee" here means the "elbow" point of the sum of within-cluster variance (SWCV). In other words, the elbow point is where the SWCV stops to drop as rapidly as before, namely the point of maximum curvature. In this study, we used the function *KneedLocator()* of Kneed package in Python to find the elbow point of the SWCV. According to the calculations, $c = 5$ is the elbow point of SWCV curve, which has the maximum curvature.

In the revised manuscript, following sentences have been added at the end of SI section (1) to explain how we determined the elbow point: "The elbow point is where the $V_{SWCV}$ stops to drop as rapidly as before, namely the point of maximum curvature. In this study, the KneedLocator function of Kneed package in Python was used to find the elbow point.".

43) **SI section (2) The equation of V$_{FS}$ is of the shape V$_{FS}$ = A-B. A smaller value of V$_{FS}$ would only indicate that A and B become more similar. That means that a bad compactness value (high A) could be compensated by a larger difference between the clusters (high B).**

**Response:** Mathematically, it could be, but the scenario exemplified by the referee is unlikely to occur in reality. A partition with bad compactness indicates a bad clustering. For a bad partition, differences between clusters should be small instead of being large.

44) **Equation S2 would be much easier to read if they introduce a variable for cluster compactness and separation of cluster. Especially since these variables are used again for V$_{XB}$ and later. The similarities/differences between the CVI s will be much easier to see using variables instead of the lengthy double sums.**

**Response:** Accepted. A compactness function, $Compact(c)$, and a separation function, $Separation(c)$ are introduced, and the related CVIs have been updated accordingly. Detailed revisions can be found in the revised SI.

45) **SI section (2): What are recommended values to identify a good solution for this CVI? From Fig 1b, V$_{FS}$ seems to always go down with cluster number until reaching a minimum at ~ -8. If smaller is better, why is c= 5 chosen?**

**Response:** There is no recommended value to identify a good partition for $V_{FS}$. According to its definition, the smaller $V_{FS}$, the better the partition. In our case, the recommended optimal number of clusters is 8 rather than 5 in terms of $V_{FS}$. However, $c = 5$ is chosen considering all CVIs. Detailed discussion about this can be found in Sect. 3.1.1. In the revised version, both the optimal value of $c$ suggested by each CVI and the adopted value of $c$ are displayed in Fig. 1, as shown below:

[Figure]

**Figure 1.** Values of selected clustering validity indices $V_{SWCV}$ (a), $V_{FS}$ (b), $V_{XB}$ (c), $V_{Kwon}$ (d), $V_{BWS}$ (e), and $FS$ (f) as a function of the number of clusters from 2 to 10. The averages of results from 50 repetitions are shown in the plot, and the error bars show the standard deviations. Blue points denote the optimal values of c according to each CVI, and the solution selected for further analysis is marked by red circles.

46) **SI section (3): "the smaller the numerator…" this sentence is talking about the individual clusters. But the formula is summing over all clusters. I think this should be changed to plural. So, the more compact the clusters are. The more the clusters are separated.**

**Response:** Accepted. The original sentence has been rephrased to "The smaller the numerator, the more compact  the clusters are, whereas the larger the denominator, the more  dispersed the clusters are from each other.".

47) **SI section (4): Eq S4 will become much more readable if the "punishing function" is defined in its own equation.**

**Response:** Accepted.

48) **SI section (4): If c approaches n, the key point about using a clustering algorithm (dimension reduction) is not achieved. Why is a metric needed that works at c approaching n? In Fig 1 c and d are identical in shape. What is the added values? To enhance clarify of the already complicated manuscript V$_{Kwon}$ or V$_{XB}$ should be omitted.**

**Response:** It is an inherent property of $V_{XB}$, that when the number of clusters ($c$)is very large, e.g., approaching the number of objects ($n$), $V_{XB}$ decreases monotonically and loses its function in determining the optimal $c$. This doesn't mean that the real $c$ approaches $n$. In our study, the optimal $c$ is far less than $n$, so the extreme case didn't occur. Consequently, the penalty function added in $V_{Kwon}$ did not change the trend of $V_{Kwon}$ in our case. As shown in Fig. 1, the trend of $V_{XB}$ and $V_{Kwon}$ are identical. $V_{Kwon}$ only has larger values than $V_{XB}$.

However, this does not mean that these two indices always share an identical behavior. Since they were both utilized in the analysis, we prefer to include both of them in Fig. 1 to maintain the integrity. Also, excluding one of the indices will not truly simplify the analysis.

49) **SI section (5): This section implies that overlapping clusters are not treated well with all previous metrics. Is that indeed the intention?**

**Response:** Not exactly. This section only provides a description of $V_{BWS}$, including its definition and background information. Bouguessa and Wang (2004) reviewed several most widely used CVIs in the literature and found that one of the major sources of failure encountered with these indices is overlapping of clusters. They

hence proposed a new validity index, namely $V_{BWS}$, to overcome this problem. They also demonstrated that it works well when there is a large overlap between clusters. This does not necessarily mean that other indices mentioned in this study have a bad performance.

50) **Higher V$_{BWS}$ values indicate a better solution. Fig 1e looks like there is a "maximum" in the curve. Is that a feature of this CVI?**

**Response:** According to its definition, a larger $V_{BWS}$ value indicates a better partition. Therefore, theoretically, there should be (at least) a maximum point in the curve of $V_{BWS}$, which corresponds to the solution with optimal $c$.

51) **SI section (6) "The average cluster silhouette score can tell if the cluster is appropriately configurated or not." Isn't it a problem if there are an equal number of bad assigned and well assigned ones? Because the positive and negative cancel each other out?**

**Response:** In theory, the example provided by the referee is possible. However, as already mentioned in this section, one of the advantages of *FS* is that it not only calculates the overall silhouette score of a clustering solution, but also the average silhouette score of each cluster and the silhouette score of each object. Therefore, we can not only judge the overall quality of the partition, but also check every object and tell if it is misclassified or not. Situations exemplified by the referee were screened out. We only considered solutions with the silhouette scores of objects and clusters being larger than 0.

52) **How are a and b calculated for equation S10? Section 2.3.2 introduces 4 ways of calculating the difference. Which one is used here?**

**Response:** As mentioned in the response to the second major comment, a description about how a and b were calculated has been added to the revised supplement. Details can be found in the response to the second major comment, or in the revised SI.

We have compared the results by using four different distance metrics, and finally Euclidean distance was chosen and used in the calculation of the distance in all CVIs. More information about this can be found in the response to the second major comment.

At the end of Sect. 3.1.2, the following sentence has been added to clarify the distance method used for CVIs calculation: "Additionally, the Euclidean distance was used in the calculation of various CVIs.". This is also clarified in the caption of Fig. 2.

**Language and technical comments**

**General: The authors should carefully check their manuscript for adverbial constructions/ inserts at the start of a sentence and decide if they follow the recommendation of separating them by a comma from the main clause. E.g., the sentence in line 57ff ("Benefitting from this it has …") should have a comma after "this". Personally, I like using the comma for this grammatical structure as it enhances readability.**

**Response:** Accepted. Thanks for the comment. We have thoroughly read and checked the entire manuscript, and a comma has been added for such grammatical structures.

**Line 26: "an approach by using FCM" --> omit the "by"**
**Response:** Accepted.

**Line 32: "system investigated" --> investigated system**
**Response:** Accepted.

**Line 32 "chemical properties were characterized…": characterised and parameterised can be used in this sentence, but the term "described" may be better in this context.**
**Response:** Accepted.

**Line 44: "… and convert to condensable vapors" Not all products of atmospheric VOC oxidation are condensable. In most cases, the majority will be still too volatile. -->rephrase**
**Response:** Accepted. The original sentence has been rephrased to "Volatile organic compounds (VOCs) in the atmosphere …, leading to the formation of  condensable vapors such as low- and extremely low-volatility organic compounds ….".

**Line 46: add comma before "and thereby" to indicate that that is referring to both condensation and nucleation.**
**Response:** Accepted.

**Line 47: SOA was already introduced in the sentence before.**
**Response:** Accepted.

**Line 55: "propagation": is that the right word here?**
**Response:** It has been replaced by "availability" in the revised version.

**Line 67: "nonwithstanding the apparatus of high resolution": this insert is not clear. What is the apparatus of high resolution? Do the authors mean an instrument with high resolution?**
**Response:** We apologize for the awkward expression. Yes, it means instruments with high resolution.

To make it concise and clear, the original sentence has been revised to "In addition, the mass spectrometers are unable to detect structures of molecules despite modern instruments with high resolution  ….".

**Line 73: "thus simplify the chemistry of the investigated system" what is meant by that? Clearly, the actual chemistry does not change? It is just our representation of the chemical processes that is simplified?**

**Response:** We are sorry for the confusing expression. We intended to clarify that data dimension-reduction techniques group hundreds to thousands of compounds into several to dozens of clusters, which retain the general chemical and kinetic information of the original data set, and thus one can extract this information in an easier and effective way, instead of systematically exploring the whole data set.

we have rephrased the original sentence to "…, and thus  obtain the chemical and kinetic information of investigated systems in an easier and more effective way….".

**Line 77: "best-known approach": in my opinion "most commonly used approach" seems more appropriate here.**

**Response:** Accepted.

**Line 74 and later: (Buchholz et al., 2019) is the wrong reference. The authors most likely mean (Buchholz et al., 2020)**

**Response:** Yes, Buchholz et al. (2020) is the right one. It has been replaced and the correct reference has been added.

**Line 148: "the injection was repeated …" Using the term "injection" can be easily misunderstood in this sentence. The previous sentence only calls the addition of isoprene an injection and nothing about the delivery of NO2 and O3. --> rephrase**

**Response:** To avoid misunderstanding, the previous sentence has been revised to " $O_3$ and $NO_2$ were added in sequence to produce $NO_3$, followed by the addition of ~ 10 ppbv of isoprene to initiate the reaction. The injections were ".

**Line 154: "particle-phase" no hyphen in this case as it is a noun (but gas- needs the hyphen)**
**Response:** Accepted.

**Line 169: "related to ion source" --> related to the ion source**
**Response:** Done.

**Line 273: "There're" --> there are**
**Response:** Done.

**Line 278: "… among all the alternatives following…" --> alternatives the following**
**Response:** Done.

**Line 322: "are called as distance" --> omit "as"**
**Response:** Done.

**Line 331: "Since it is difficult…." sentence: comma after "data"**
**Response:** Done.

**Line 379: "Where b a constant that is usually set to be 10 in practice": the grammar of this sentence seems broken.**

**Response:** To make its grammar correct, we have added "is" to this sentence "where $\beta$ is a constant that is usually set to be 10 in practice".

**Line 393: "The clustering results of FCM is …" --> "The results are …"**
**Response:** Done.

**Line 409: "produce target compound" --> produce the target(ed) compound**
**Response:** Done.

**Line 409: "The kinetic information …" sentence: comma after "species" to close the insert**
**Response:** Done.

**Line 411: "involve" should be include**
**Response:** Done.

**Line 485: "Crisp Silhouette (C)": in the next sentences CS is used as an abbreviation**
**Response:** "Crisp Silhouette (C)" has been replaced by "Crisp Silhouette (CS)".

**Line 533: "is dependent" should be "depends"**
**Response:** Done.

**Line 599: Which one is meant by "new" cluster? C5 (yellow)?**

**Response:** Yes, it's cluster 5 (the orange one in Fig. 4a). We have added "(cluster 5)" right after "new cluster" to explain which one it refers to.

**Line 630f: "Even clusters with similar generation number […] are grouped into different clusters due to their different chemical properties" I do not understand this sentence. Clusters are grouped into clusters?**

**Response:** We are sorry for the awkward expression. The original sentence has been revised to " For instance, the two early-generation clusters, like cluster 1 and cluster 2 in the four-cluster solution, are  differentiated in chemical properties from each other.".

**Line 668f: What is meant by "screened out"? Do you mean identified? Selected?**

**Response:** Yes, it means "identified/ selected" here. We have replaced "screened" to "selected" in the revised manuscript.

**Line 685f: The trend description in this sentence (for the model data) may be easier to understand if presented with in the opposite way. I.e., speaking of the less pronounced increase of OSc and decrease of nC with increasing chemical age of the clusters.**

**Response:** To enhance the legibility, the original sentences have been written to "However,  the increase in the $\overline{OS_C}$ of clusters for the model data is less pronounced during the oxidation processes, probably due to the absence of autooxidation steps in the MCM. Moreover, the MCM lacks accretion products (mostly assigned to early-generation clusters with more carbon atoms in bulk), but tends to have more small species …."

**Line 698: "information underlying in the mass spectrometric data." This sounds incorrect. It probably should be "underlying information in the mass spectrometric data"**

**Response:** "underlying in" has been replaced by "from", and this paragraph has been rewritten to enhance the readability. Detailed revisions can be found in the revised manuscript.

**Line 744: Is "attribution" the correct word here or should it be "contribution"?**

**Response:** Yes, "contribution" sounds more appropriate. It has been revised.

**Line 726f: "it is indeed mainly the former cluster 2 in the five-cluster solution is further split into new clusters 2 and 3" --> it is indeed …which is further split…**

**Response:** Done.

**Line 826: "larger N:C values as expected" the "as expected" sounds a bit weird in this context. Using a comma to indicate the "as expected" as a grammatical insert will help. Or putting the "as expected" at the start of the sentence.**

**Response:** Accepted. A comma has been added in front of "as expected".

**Line 856: "resulting products" --> omit resulting**

**Response:** Done.

**Line 875: "fuzziness of FCM in belongingness of cluster members": the term "belongingness" does not work here. Maybe change to "assignment of cluster members"**

**Response:** Sorry for the awkward expression. To make it concise and more readable, the original sentence has been revised to "Due to the fuzziness of FCM in belongingness of cluster members, When considering the characteristics of members in each cluster, we focus solely on only high-affiliation species (with a membership degree over 0.5) are considered as members of a cluster in the following discussion for simplicity to simplify the discussion.".

**Line 895: "gamma kinetic parameterization" was already introduced --> use GKP here**

**Response:** Done.

**Line 951: & 962 Using the comma separated values for oxygen in the sum formulas confused me at first (C5H9NO4,5). Since there are only two cases where this nomenclature is used, consider writing both formulas out.**

**Response:** Done.

**Line 963: "preferably occupied by cluster 3" this sounds incorrect. I am not sure what is meant by occupied in this context**

**Response:** We apologize for the awkward expression. The original sentence has been rephrased to "As shown in Fig. 12, …, while $C_5H_8N_2O_7$,  $C_5H_{10}N_2O_8$ and $C_5H_{10}N_2O_9$ are  primarily assigned to cluster 3.".

**Line 1001: the acronym "HAC" was not introduced**

**Response:** We are sorry for the typo. It should be HCA, which stands for hierarchical clustering analysis and was introduced in Line 99 in the introduction.

**Figure 2a has a y scale up to +/-600? FS value is -2 - -8 in Fig 1f --> reduce scale in Fig 2a**

**Response:** Accepted. Figure 2 have been updated, as shown below:

[Figure]

**Figure 2.** Values of selected clustering validity indices $V_{FS}$ (a), $V_{Kwon}$ (b), $V_{BWS}$ (c), and $FS$ (d) as a function of the number of clusters. Points in different colors are results obtained with different distance or similarity metrics. The averages of results from 50 repetitions are shown in the plot, and the error bars denote the standard deviations. Euclidean distance was used in the calculation of various CVIs.

**Fig S1: left axis labels are cut off**

**Response:** The size of this plot has been adjusted to ensure that it can be displayed completely. The updated version is shown below:

[Figure]

**Figure S1.** Concentrations of trace gases ($NO_x$, $NO_y$, and isoprene) and conditions of the chamber experiment selected for FCM analysis in this study. Adapted from Wu et al. (2021).

**What is Table S1 for? I did not notice any reference to this table in the main manuscript or the SI.**

**Response:** Table S1 lists the possible permutation scheme for the formation of 2N-, 3N-, and 4N-dimers. The sentence in Line 952 refers to this table. And we have also added a sentence "Possible permutation scheme for the formation of 2N- and 3N-dimers can be found in Table S1 in the supplement." in L1173-1174 to explain the usage of this table.

**SI paragraph (6) "… was first proposed by Rousseeuw (1987), which can be used …" it is not clear where the "which" is pointing to. Better link the two parts with "and" -> "… was first proposed by Rousseeuw (1987) and can be used …"**

**Response:** Accepted.

**SI paragraph (6) "With different cluster number …" sentence is difficult to understand. --> consider rephrasing**

**Response:** The original sentence has been revised to " When plotting the overall silhouette score as a function of cluster number, the maximum point of the curve indicates the optimal value of $c$, where the clustering solution has a minimum intra-cluster distance ($a_j$) and a maximum inter-cluster distance ($b_j$).".

**Fig S7: it was difficult for me to line up the markers for, e.g., cluster 1 with the names on the x axis. For me, some vertical grid lines every X ions would help to maker this figure more readable.**

**Response:** Accepted. The redrawn plot is show below:

[Figure]

**Figure S7.** Cluster apportionment of species for the five-cluster solution. The sum of fractions of a compound in each cluster adds up to 1. Different clusters are distinguished by color, and the color scheme follows that in Fig. 4. Species are listed in the same order (in order of molecular mass) to those in Fig. 7.

**Fig 7 and S7: What is the sorting criterium for the species in these figures? Ion mass? Consider if that is the optimal way of presenting the data or if another order (e.g., by C number) would be beneficial.**

**Response:** Yes, they are shown in an ascending order of molecular mass in the plots, and we have clarified this in the captions of figures. In essence, there is no difference whether they are shown in order of molecule mass, or in another order like C number. We therefore leave these figures unchanged.

**Reference**

Bouguessa, M. and Wang, S.-R.: A new efficient validity index for fuzzy clustering, Proceedings of 2004 International Conference on Machine Learning and Cybernetics (IEEE Cat. No. 04EX826), 1914-1919.

Cai, D., Wang, X., George, C., Cheng, T., Herrmann, H., Li, X., and Chen, J.: Formation of secondary nitroaromatic compounds in polluted urban environments, Journal of Geophysical Research: Atmospheres, 127, e2021JD036167, 2022.

Gao, X.-B., PEI, J.-h., and XIE, W.-x.: A study of weighting exponent m in a fuzzy c-means algorithm, ACTA ELECTONICA SINICA, 28, 80, 2000.

Hathaway, R. J. and Bezdek, J. C.: Fuzzy c-means clustering of incomplete data, IEEE Transactions on Systems, Man, and Cybernetics, Part B (Cybernetics), 31, 735-744, 2001.

Huang, M., Xia, Z., Wang, H., Zeng, Q., and Wang, Q.: The range of the value for the fuzzifier of the fuzzy c-means algorithm, Pattern Recognition Letters, 33, 2280-2284, 2012.

Koss, A. R., Canagaratna, M. R., Zaytsev, A., Krechmer, J. E., Breitenlechner, M., Nihill, K. J., Lim, C. Y., Rowe, J. C., Roscioli, J. R., and Keutsch, F. N.: Dimensionality-reduction techniques for complex mass spectrometric datasets: application to laboratory atmospheric organic oxidation experiments, Atmospheric chemistry and physics, 20, 1021-1041, 2020.

Kroll, J. H., Ng, N. L., Murphy, S. M., Flagan, R. C., and Seinfeld, J. H.: Secondary organic aerosol formation from isoprene photooxidation, Environ. Sci. Technol., 40, 1869–1877, https://doi.org/10.1021/es0524301, 2006.

Ng, N., Kwan, A., Surratt, J., Chan, A., Chhabra, P., Sorooshian, A., Pye, H. O., Crounse, J., Wennberg, P., and Flagan, R.: Secondary organic aerosol (SOA) formation from reaction of isoprene with nitrate radicals (NO3), Atmospheric Chemistry and Physics, 8, 4117–4140, 2008.

Ozkan, I. and Turksen, I.: Upper and lower values for the level of fuzziness in FCM, in: Fuzzy Logic, Springer, 99-112, 2007.

Pal, N. R. and Bezdek, J. C.: On cluster validity for the fuzzy c-means model, IEEE Transactions on Fuzzy systems, 3, 370-379, 1995.

Wang, Y., Hu, M., Wang, Y., Zheng, J., Shang, D., Yang, Y., Liu, Y., Li, X., Tang, R., and Zhu, W.: The formation of nitro-aromatic compounds under high NO x and anthropogenic VOC conditions in urban Beijing, China, Atmospheric Chemistry and Physics, 19, 7649-7665, 2019.

Wu, K.-L.: Analysis of parameter selections for fuzzy c-means, Pattern Recognition, 45, 407-415, 2012.

Wu, R., Vereecken, L., Tsiligiannis, E., Kang, S., Albrecht, S. R., Hantschke, L., Zhao, D., Novelli, A., Fuchs, H., Tillmann, R., Hohaus, T., Carlsson, P. T. M., Shenolikar, J., Bernard, F., Crowley, J. N., Fry, J. L., Brownwood, B., Thornton, J. A., Brown, S. S., Kiendler-Scharr, A., Wahner, A., Hallquist, M., and Mentel, T. F.: Molecular composition and volatility of multi-generation products formed from isoprene oxidation by nitrate radical, Atmospheric Chemistry and Physics, 21, 10799-10824, 10.5194/acp-21-10799-2021, 2021.

Xu, Z., Nie, W., Liu, Y., Sun, P., Huang, D., Yan, C., Krechmer, J., Ye, P., Xu, Z., and Qi, X.: Multifunctional products of isoprene oxidation in polluted atmosphere and their contribution to SOA, Geophysical Research Letters, 48, e2020GL089276, 2021.

---

## Author Comment (AC2)

The reviews of our manuscript are thorough and well-considered. We would like to thank the reviewers for their careful reading and valuable comments, as well as for the improvements they help us make.

All the suggestions and comments from Referee 1 are addressed below point by point in bold text, followed by our responses in non-bold text. The corresponding revisions to the manuscript are marked in blue. All updates to the original submission are tracked in the revised manuscript.

**Wu and colleagues present the novel application of a data treatment technique (fuzzy *c*-means clustering) that reduces the complexity of a data set for the interpretation of atmospheric mass spectra. I consider the exploration of novel techniques for the extraction of information from increasingly more complex and information rich mass spectra to be of interest to the atmospheric science community and the readership of AMT. The manuscript strikes a balance between being an introduction to the technique and the demonstration of the technique on an established chemical system. I consider the chosen format and presentation as appropriate for the presentation of a new approach, and can see how this manuscript could serve as a reference for future studies using the technique.**

**I enjoyed reading this manuscript and would welcome a publication after some minor points have been addressed.**

Thanks for the positive comments. It's appreciated.

**Could you touch briefly on the practical implementation of the algorithm? I do not readily find what software you used. What is the code availability? What are typical run times of the algorithm?**

**Response:** The fuzzy c-means clustering was implemented using the open-source scikit-fuzzy (v 0.4.2) package (https://pypi.org/project/scikit-fuzzy/) in Python. For each case, the algorithm was

repeated 50 times to get a robust result, and each run took several to tens of seconds depending on the number of clusters.

In the revised manuscript in L293-294, a sentence "In this study, the FCM clustering was implemented using the open-source scikit-fuzzy (v 0.4.2) package (https://pypi.org/project/scikit-fuzzy/) in Python." has been added to address this point.

**I did not check the correctness of the equations, but encourage the authors to double (or even triple) check the manuscript for typos in the equations before the final publications.**

**Response:** Thanks for this comment. We have double checked all the equations in both the main text and the supplement. Typos have been revised.

**Else, I found few typographical errors, only, which will be taken care of in copy-editing, and limit my comments to content observations only.**

**Response:** Thanks for this.

**Paragraph 3.1.1: unclear on what data set you "ran the FCM algorithm 50 times", please clarify.**

**Response:** We apologize for the unclear expression. In this study, except for results shown in Sect. 3.2.2, all others are results from the FCM analysis of chamber data. In Sect. 3.1.1, the FCM algorithm was applied to chamber data to find out the optimal number of clusters.

In order to make this point clearer, the original sentence has been revised to "To explore the effect of cluster number on partition results,  we applied the FCM algorithm to the chamber data with $c$ varying from 2 to 10. For each $c$ in this range, the algorithm was run 50 times and the selected CVIs were calculated accordingly for each repetition.".

**Paragraph 3.1.3: expand on the accuracy of $m^*$. Is 1.42 and 1.52 significantly different or essentially the same? Generally, what can be considered as different?**

**Response:** That is a good point! The value of $m$ defines the fuzziness degree of a clustering, which affects the convergency of the algorithm and the separation of clusters. Generally, the larger $m$ is, the fuzzier the FCM clustering results would be. We have checked the clustering outcomes with $m = 1.4, 1.5, 1.6, 1.7, 1.8, 1.9$, and $2.0$. The patterns of cluster centers are almost identical with different $m$. However, for different $m$, some of the membership degrees of objects to cluster centers changed to some degree (but less than 50% at most), which implies that $m$ does affect the clustering outcomes. In this regard, we would say selecting $m = 1.4$ or $m = 1.5$ makes a difference.

In addition, the major purpose of this section is to provide a method to determine the "optimal" fuzzifier value for a given data set, rather than using the default value of $m = 2$, which has been proven to be inappropriate in many conditions (Huang et al., 2012; Hwang and Rhee, 2007; Schwämmle and Jensen, 2010; Yu et al., 2004; Zhou et al., 2014). In this study, we intend to provide a method to determine the optimal value of $m$. The exact value of $m^*$ is not what we really care about.

**Fig. 4: I was a little thrown off by the missing x-axis label on the mass profile panels. Can you add a label, and/or label the dominant species in the individual panels? Also, consider changing the time axis label to elapsed time since start of experiment.**

**Response:** Accepted. Due to limited space, species No. is added in the x-axis of panel (b) in Fig. 4, instead of chemical formulas. Specific chemical formulas of each species are listed in Fig. S7. The updated plot is shown below.

In addition, Fig. S4 has been revised accordingly. The time axis of Fig. 6 has also been changed to time elapsed since the start of experiment.

[Figure]

**Figure 4.** Results of fuzzy c-means clustering for chamber data with cluster numbers between 2 and 5: Time series (a) and mass profiles (b) of clusters for each solution (in row). The time series of cluster centers are shown as thick, colored solid lines, and the time series of species with the membership degree larger than 0.5 to the cluster are illustrated as thin, gray lines. The species number in panel (b) corresponds to species listed in Fig. S7 (in order of molecular mass).

**L184: Consider rewording "and only formulas within an accuracy tolerance of 10 ppm and with reasonable chemical meanings were considered." to "and only plausible formulas with relative m/z deviations smaller than 10 ppm were considered"**

**Response:** Done.

**L290: Incomplete sentence. Missing "be"?**

**Response:** Done. To make it complete and concise, the original sentence has been revised to "…, what we believe to the expected optimal number of clusters is determined.".

**L300: Consider rewording "the right choice" to "may not always be appropriate".**

**Response:** Done.

**L428: "mathematically unsolvable". Probably better to say that there is no simple analytical solution.**

**Response:** Done. The original sentence has been reworded to "…, and there is no simple analytical solution for the differential equations that describe Eq. 11 are mathematically unsolvable.".

**L467: Punishing function? While punishing seems to be used in some literature, it should maybe rather be penalty instead?**

**Response:** Totally agreed on. Penalty function should be the right word. The "punishing function" in this sentence has been replaced by "penalty function". In addition, the improper expression in S1 has also been revised.

**L493: reword "looks reasonable"**

**Response:** Done. The original sentence has been changed as follows: " It seems more sensible to set the number of clusters to 5, as this is where $FS$ reaches its local maximum and $V_{SWCV}$ is significantly reduced and has the maximum curvature .".

**L512: Drop "As a quick reminder"**

**Response:** Done. "As a quick reminder" in the original sentence has been replaced by "As mentioned before".

**L699: "mathematically"? Consider making the paragraph (especially lines 699-701) more concise.**

**Response:** Done.

The original paragraph has been revised to "In this section, we utilize  the five-cluster solution, identified as the optimal cluster number for our dataset (Sect. 2.3), to illustrate how to extract  chemical and kinetic information from  the mass spectrometric data based on the FCM analysis.  This does not necessarily mean that the five-cluster solution  is superior over  other solutions. However, as demonstrated  in  previous sections,  the FCM results exhibit consistent  features regardless of the  number of clusters predefined . Therefore,  findings derived from  the five-cluster solution could potentially apply to  other cases.".

**L736: marker size: I appreciate the attempt to be specific about what the marker size represents. Please be fully specific by referring to the marker area or the diameter. This comment applies to a couple more figures in the main text and SI.**

**Response:** The marker area of clusters is proportional to the sum of average signal intensity of all species in the cluster weighted by their membership degrees, and that of species is proportional to the average intensity of species over the whole experiment.

To make this clear, the original caption of Fig. 7 has therefore been rewritten to "Chemical properties of clusters from the five-cluster solution. The subplots show mass profile of each cluster (a), van Krevelen plot (b), and average carbon oxidation state of clusters (c), respectively. Different clusters are distinguished by color, and the color scheme follows that in Fig. 4. The marker area of clusters is proportional to the sum of average signal intensity of all species in the cluster weighted by their membership degrees. The species number in panel (a) corresponds to species listed in Fig. S7 (in order of molecular mass). Grey  hexagons in panel (b) and panel (c) denote species identified by Br⁻ CIMS, and  the marker  area is proportional to  the average intensity of  species over the whole experiment."

The captions of Fig. 5, Fig. 8 - 11, and Fig. S5 - S6 have been revised accordingly.

**Fig. 11: Units for k**

**Response:** $k$ is a second-order rate constant in unit of $cm^3$ molecule$^{-1}$ s$^{-1}$. Figure 11 has been updated to include the unit of $k$. The updated plot is shown below:

[Figure]

**Figure 11.** Fitted effective rate constant ($k$) and generation number ($m_G$) of the high-affiliation species of each FCM cluster. The cluster centers and members are denoted by color-coded

circles and pentagrams, respectively. The circle area is proportional to the average signal intensity of species over the whole experiment.

**Supplementary information**

**Page 2: proposed >the< Kwon index.**

**Response:** Done.

**Page2: punishing function, same as main text.**

**Response:** Done. "punishing function" has been replaced by "penalty function".

**Page 4: it's --> it is**

**Response:** Done.

**Fig S1: axis labels cut off**

**Response:** The size of this plot has been adjusted to ensure that it can be displayed completely. The updated version is shown below:

[Figure]

**Figure S1.** Concentrations of trace gases (NO$_x$, NO$_y$, and isoprene) and conditions of the chamber experiment selected for FCM analysis in this study. Adapted from Wu et al. (2021).

**Fig S2: subscripts in O3, NO2, NO3**

**Response:** Done. The caption of Fig. S2 has been updated accordingly.

**Fig S4: same comments as main text figure**

**Response:** Done. Figure S4 has been updated according to referee's comments, as shown in the following:

[Figure]

**(a)**

[Figure]

**Figure S4.** Fuzzy c-means clustering results of chamber data with 7-10 clusters. Time series (a) and profiles (b) of clusters for each solution. The cluster centers are shown as colored thick lines, and species with the membership degree larger than 0.5 to the cluster are illustrated as thin lines in gray. The species number in panel (b) corresponds to species listed in Fig. S7 (in order of molecular mass).

**References**

Huang, M., Xia, Z., Wang, H., Zeng, Q., and Wang, Q.: The range of the value for the fuzzifier of the fuzzy c-means algorithm, Pattern Recognition Letters, 33, 2280-2284, 2012.

Hwang, C. and Rhee, F. C.-H.: Uncertain fuzzy clustering: Interval type-2 fuzzy approach to c-means, IEEE Transactions on fuzzy systems, 15, 107-120, 2007.

Schwämmle, V. and Jensen, O. N.: A simple and fast method to determine the parameters for fuzzy c–means cluster analysis, Bioinformatics, 26, 2841-2848, 2010.

Yu, J., Cheng, Q., and Huang, H.: Analysis of the weighting exponent in the FCM, IEEE Transactions on Systems, Man, and Cybernetics, Part B (Cybernetics), 34, 634-639, 2004.

Zhou, K., Fu, C., and Yang, S.: Fuzziness parameter selection in fuzzy c-means: the perspective of cluster validation, Science China Information Sciences, 57, 1-8, 2014.